# Reaction Graph: Towards Reaction-Level Modeling for Chemical Reactions with 3D Structures

Yingzhao Jian [1]   Yue Zhang [1]   Ying Wei [1]   Hehe Fan [1] [*]   Yi Yang [1]

## Abstract

Accurately modeling chemical reactions using Artificial Intelligence (AI) can accelerate discovery and development, especially in fields like drug design and material science. Although AI has made remarkable advancements in single molecule recognition, such as predicting molecular properties, the study of interactions between molecules, particularly chemical reactions, has been relatively overlooked. In this paper, we introduce Reaction Graph (RG), a unified graph representation that encapsulates the 3D molecular structures within chemical reactions. RG integrates the molecular graphs of reactants and products into a cohesive framework, effectively capturing the interatomic relationships pertinent to the reaction process. Additionally, it incorporates the 3D structure information of molecules in a simple yet effective manner. We conduct experiments on a range of tasks, including chemical reaction classification, condition prediction, and yield prediction. RG achieves the highest accuracy across six datasets, demonstrating its effectiveness. The code is available at https://github.com/Shadow-Dream/Reaction-Graph.

## 1. Introduction

In recent years, data-driven Artificial Intelligence (AI) methods have made significant strides in chemistry (De Almeida et al., 2019), bioinformatics (Senior et al., 2020; Jumper et al., 2021; Abramson et al., 2024), pharmaceutical (Wang et al., 2023a; Mak et al., 2023), and materials science (Butler et al., 2018), considerably enhancing research efficiency and accuracy, reducing costs and accelerating discovery cycles. In the field of chemistry (Cheng et al., 2024), AI enables

precise spectral analysis (Young et al., 2024) and quantum chemical simulation (Gilmer et al., 2017), improves inverse design of molecular structure (Jin et al., 2018) and retrosynthesis planning (Dong et al., 2022). However, most related methods primarily concentrate on recognizing and understanding single molecules, such as predicting their properties or functions (Yang et al., 2019; Zhou et al., 2023). The study of interactions between molecules, particularly chemical reactions, has not garnered as much attention.

Learning accurate representation of chemical reactions is essential for reaction recognition and understanding, benefiting various tasks such as predicting reaction conditions (Wang et al., 2023b), types (Schwaller et al., 2021a), and yields (Kwon et al., 2022b). As shown in Fig. 1, early works typically employ bit vector representations of reactions, i.e., fingerprints, to predict relevant reaction properties (Gao et al., 2018). With the advent of the Transformer in natural language processing, the string-based Simplified Molecular Input Line Entry System (SMILES) has gained widespread popularity (Wang et al., 2023b; Yin et al., 2024).

Among various representation methods, molecular graphs have proven inherently advantageous for various chemical tasks (Fang et al., 2022; Zhou et al., 2023). However, as shown in Fig. 1, most graph-based methods first employ single-molecule modeling to extract individual molecule-level representations for reactants and products, and then combine these representations to form an ensemble reaction representation for the prediction (Kwon et al., 2022a;b; Zhang et al., 2022). These methods largely overlook the reaction information itself, relying solely on molecule-level representations, which inevitably complicates reaction recognition and understanding. Rxn Hypergraph (Tavakoli et al., 2022) may have potential to mitigate this issue. It first learns a hypernode for reactants and another for products, and then merges these nodes as the reaction representation. However, this method still separates reactions, which also causes loss of reaction information. Moreover, in single-molecule modeling, 3D structures are widely used due to their intrinsic connection to molecular properties. Yet, the utilization of 3D structures has remained largely unexplored in reaction modeling. This oversight prompts the question of whether incorporating 3D molecular structures could enhance reaction prediction.

---

[1] College of Computer Science and Technology, Zhejiang University, Hangzhou, China. Correspondence to: Hehe Fan <hehefan@zju.edu.cn>.

*Proceedings of the 42nd International Conference on Machine Learning*, Vancouver, Canada. PMLR 267, 2025. Copyright 2025 by the author(s).

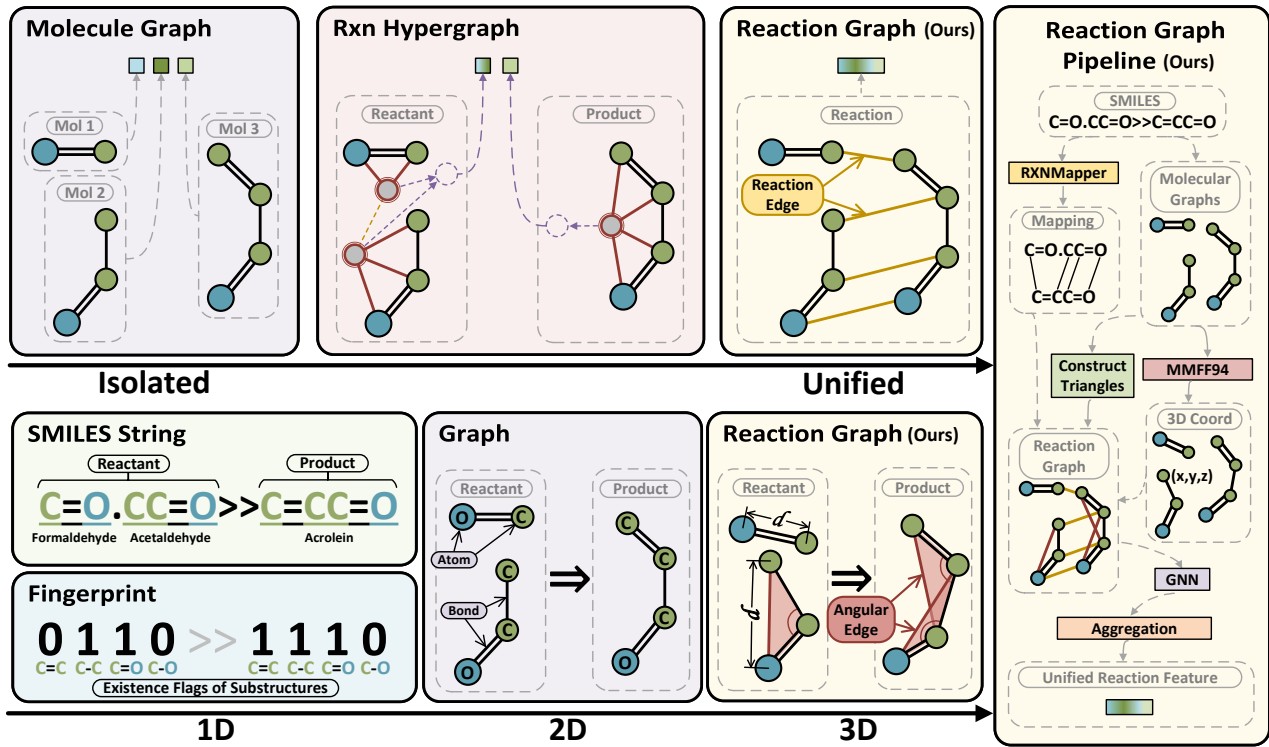

*Figure 1.* Illustration of Reaction Graph (RG). (1) Existing methods extract isolated representations for reactants and products, and then combine them for prediction, which may fail to effectively model reaction relationship. In contrast, RG unifies the modeling for reactants, products and reactions. (2) Existing 1D- or 2D-based methods may not adequately capture the complexity of molecular structures. RG exploits edge length and an angular edge to implicitly model the 3D structure information. (3) Our method first constructs molecular graph based on SMILES and predicts atomic mapping for creating reaction edges using RXNMapper (Schwaller et al., 2020). Then, 3D atom coordinates are calculated using MMFF94 (Halgren, 1996) and angular edges are constructed for each bond angle. Finally, a GNN is used to extract the unified reaction feature vector based on RG.

In this paper, we propose Reaction Graph (RG) to effectively model chemical reactions. To model the chemical reaction as an entirety and capture the molecular transformations occurring during reaction, we integrate a reaction edge into graph. This edge connects nodes representing the same atom in both reactants and products, based on atomic mapping relationships. It discerns molecular independence, while allowing graph neural networks (GNNs) to exchange information between reactants and products during the message-passing phase, thereby assimilating changes in chemical reactions. Furthermore, we enhance the graph's capability by embedding 3D spatial information through a new rotationally and translationally invariant approach. Specifically, we utilize edge length and introduce an angular edge to implicitly convey bond angle information by forming shape-stable triangles within the molecular graph. We conduct extensive experiments on a range of reaction-related tasks, including chemical reaction condition prediction, reaction yield prediction and reaction classification. Experimental results indicate that the proposed method is efficient and effective, outperforming existing methods on six datasets. The contributions of this paper are three-fold:

- We propose Reaction Graph, a novel unified graph representation for chemical reactions that allows GNNs to extract reaction transformation related features during the message passing stage.

- We integrate 3D molecular information into reaction modeling. Additionally, we develop a new method to implicitly convey invariant features of bond angles.

- We achieved state-of-the-art accuracy in several tasks, demonstrating the effectiveness of our methods.

## 2. Proposed Method

In this section, we first briefly review the Molecular Graph (MG) representation. Then, we discuss the potential limitations of MG in reaction modeling and describe the proposed Reaction Graph (RG) in detail. Finally, we incorporate RG into deep neural networks to address multiple chemistry tasks, including reaction condition prediction, yield prediction, and reaction classification.

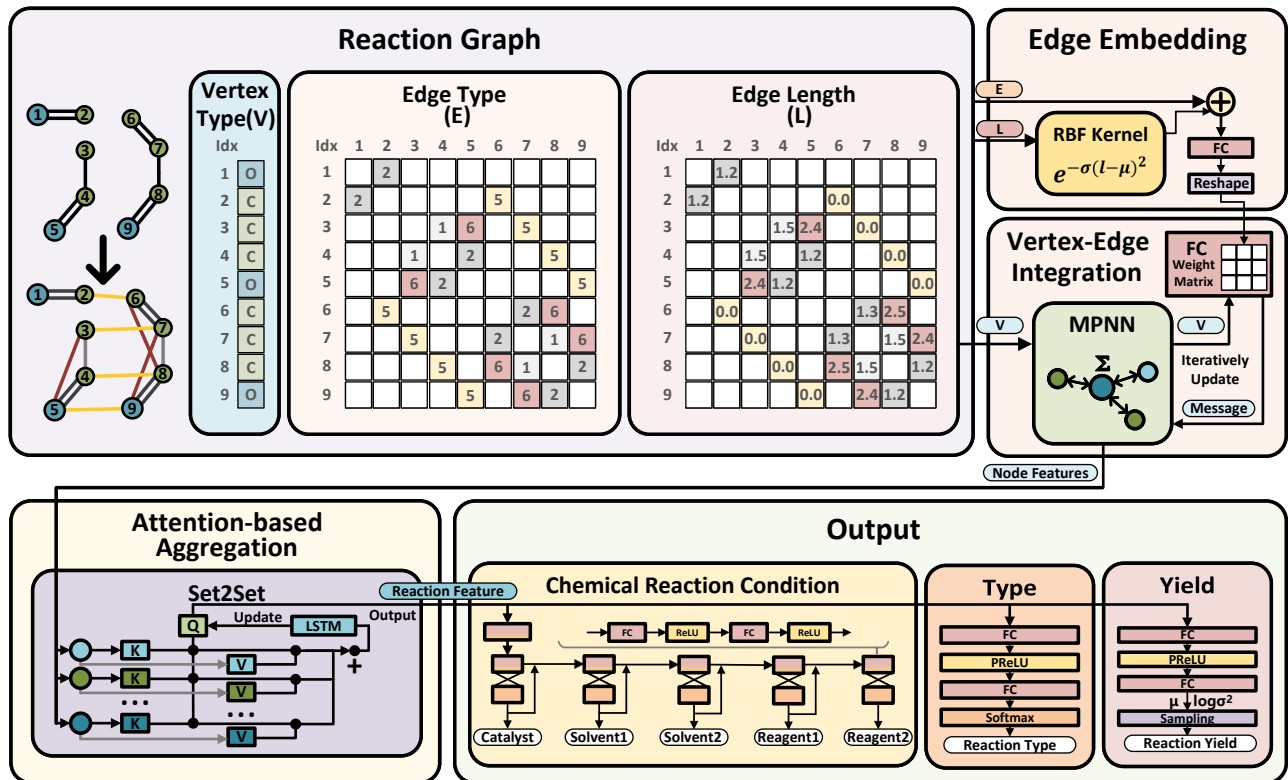

*Figure 2.* Illustration of the proposed Reaction Graph and the associated model architecture. The input contains the vertex type matrix $\boldsymbol{V}$, the edge type matrix $\boldsymbol{E}$ and the edge length matrix $\boldsymbol{L}$ of Reaction Graph. Model first computes 3D-aware edge embeddings, and then iteratively integrates edge and vertex information into vertex features. Vertex features are aggregated into a unified reaction feature using attention-based method. Finally, task-specific output modules generate prediction results based on reaction features.

## 2.1. Preliminary: Molecular Graph

In computational and mathematical chemistry, a molecular graph is a representation of a chemical compound's structural formula using graph theory. It is a labeled graph where the vertices represent the compound's atoms and the edges represent chemical bonds. The vertices are labeled with the types of corresponding atoms, while the edges are labeled with the types of bonds.

Specifically, a molecular graph can be represented as $\mathcal{G} = (\boldsymbol{V}, \boldsymbol{E})$, where $\boldsymbol{V} \in \mathbb{R}^{N \times 1}$ denotes the vertices and $\boldsymbol{E} \in \mathbb{R}^{N \times N}$ denotes the edges. Here, $N$ represents the number of vertices. The edge between the $i$-th atom and $j$-th atom is denoted as $e_{ij} \in \{0, 1, 2, 3, 4\}$, with each number corresponding to a specific type of chemical bond:

> 0 : no edge, 1 : single bond, 2 : double bond,
> 3 : triple bond, 4 : aromatic bond.

Molecular graph provides direct access to the graph underpinning all molecule objects, allowing seamless integration with existing graph functionality.

## 2.2. Reaction Graph

When using molecular graphs to model chemical reactions, existing methods typically begin by extracting individual representations for each reactant and product, then combine these representations to form an ensemble reaction representation for prediction. In doing so, these approaches often overlook the reaction information itself, relying exclusively on molecule-level representations. Moreover, the absence of 3D structural information increases the challenge for deep neural networks to effectively model molecules and reactions.

To address these issues, we extend the Molecular Graph into a Reaction Graph (RG). To incorporate reaction modeling, we introduce a reaction edge. This edge links nodes representing the same atom in reactants and products based on atomic mapping, enabling deep neural networks to capture changes in chemical reactions. Additionally, to incorporate 3D spatial structure modeling into RG, we develop a simple yet effective method that is rotationally and translationally invariant. This method utilizes two chemical bond edges and a proposed angular edge to implicitly convey bond angle information by forming stable triangles within molecular

graphs. The two bond edges serve as adjacent edges, while the angular edge acts as the diagonal edge. In summary, we extend Molecular Graph to the following Reaction Graph,

$$\mathcal{G} = (\boldsymbol{V}, \boldsymbol{E}, \boldsymbol{L}). \tag{1}$$

In the Reaction Graph, we introduce a new edge attribute, specifically the edge length $\boldsymbol{L} \in \mathbb{R}^{N \times N}$, to represent the 3D structure. We use $l_{ij}$ to denote the length between the $i$-th node and the $j$-node. Additionally, the edge types are expanded to seven categories, i.e., $\boldsymbol{E} \in \{0, 1, 2, 3, 4, 5, 6\}^{N \times N}$, with each number corresponding to a specific type of edge:

> 0 : no edge,  1 : single bond,  2 : double bond,
> 3 : triple bond,  4 : aromatic bond,
> 5 : reaction edge,  6 : angular edge.

If there is no edge between nodes $i$ and $j$, or if the edge type is a reaction edge, the length $l_{ij}$ is defined as 0.

**Edge Embedding.** We use the radial basis function (RBF) kernel to embed the edge length,

$$\boldsymbol{l}_{ij} = \exp(-\boldsymbol{\sigma}(l_{ij} \cdot \boldsymbol{1} - \boldsymbol{\mu})^2), \tag{2}$$

where $\boldsymbol{\sigma}$ and $\boldsymbol{\mu}$ are learnable parameters that transform the scalar edge length into vector representations.

**Vertex-Edge Integration.** To merge the vertex and edge into a unified representation, we follow MPNN (Gilmer et al., 2017) and convert the edge information, including type and length, into a linear projection, which is then applied to the vertex representation as follows,

$$\mathcal{M}_{ij} = \text{Reshape}(\boldsymbol{W}_v \cdot [\boldsymbol{l}_{ij}; \boldsymbol{e}_{ij}]), \tag{3}$$

$$\boldsymbol{v}_i^{t+1} = \boldsymbol{v}_i^t + \sum_{j \in \mathbb{N}_i} \mathcal{M}_{ij} \cdot \boldsymbol{v}_j^t, \quad \boldsymbol{v}_i' = \boldsymbol{v}_i^{T_1}, \tag{4}$$

where $[\cdot; \cdot]$ denotes concatenation, $\boldsymbol{e}_{ij}$ is the one-hot vector of the edge type for edge $ij$, $\boldsymbol{W}_v$ is the learnable parameters for vertex-edge integration, the Reshape$(\cdot)$ function reshapes a vector to a matrix, $\mathbb{N}_i$ denotes the set of neighbors of the $i$-th node, $\boldsymbol{v}_j$ represents the representation of the $j$-th vertex, and $T_1$ denotes the total number of iterations. In this way, the vertex representation $\boldsymbol{v}_i'$ becomes edge-related and is able to collect related information from its neighbors.

**Attention-based Aggregation.** To capture the global representation of a Reaction Graph, inspired by Set2Set (Vinyals et al., 2016), we employ an attention-based aggregation method with an LSTM. Specifically, at each iteration of the LSTM, we use the hidden state $\boldsymbol{h}$, initially set to $\boldsymbol{0}$ (the same initialization applies to the cell state $\boldsymbol{c}_0 = \boldsymbol{0}$), to query over all vertices with a softmax-based attention mechanism and collect the most informative clues from these vertices, as

described below:

$$\boldsymbol{h}^{t+1}, \boldsymbol{c}^{t+1} = \text{LSTM}(\boldsymbol{q}^{t+1}; \boldsymbol{h}^t, \boldsymbol{c}^t), \tag{5}$$

$$\alpha_i^t = \frac{\exp(\boldsymbol{v}_i' \cdot \boldsymbol{h}^t)}{\sum_{j=1}^N \exp(\boldsymbol{v}_j' \cdot \boldsymbol{h}^t)}, \quad \boldsymbol{q}^{t+1} = \sum_{i=1}^N \alpha_i^t \times \boldsymbol{v}_i'. \tag{6}$$

where $\alpha_i^t$ represents the attention weight of atom $i$ at the $t$-th iteration. After the $T_2$ iteration, the model outputs the reaction global representation vector $\boldsymbol{r}$, where $\boldsymbol{r} = \boldsymbol{W}_r \cdot [\boldsymbol{q}^{T_2}; \boldsymbol{h}^{T_2-1}] + \boldsymbol{b}_r$, with $\boldsymbol{W}_r$ and $\boldsymbol{b}_r$ being the learnable weight matrix and bias, respectively.

**Implementation Details.** As shown in Fig. 1, to construct RG, we first use RXNMapper to predict the atomic mapping, and then employ MMFF94 to calculate atom coordinates. Our method traverses all the angles in molecular graphs to construct angular edges and use the atomic mapping to construct reaction edges, resulting in the final RG.

As shown in Fig. 2, when applying RG to reaction condition prediction, we use an iterative output technique (Gao et al., 2018) to support beam search. Moreover, we employ a two-stage training strategy. Following the joint training in the first stage, the parameters of the neural network are frozen, and the output module's parameters are reinitialized. Then in the second stage, the output module is trained separately.

For reaction yield prediction, due to the high noise in yield data, we follow Kwon et al. (2022b) and simultaneously output the mean $y$ and variance $\sigma^2$ of the predicted yield. When the model encounters noise during training, it can increase the predicted variance to keep the output mean relatively stable, thus enhancing training stability. In implementation, we utilize a Multilayer Perceptron (MLP) with Monte Carlo Dropout (Gal & Ghahramani, 2016) technique.

Lastly, the reaction classification module uses a standard three-layer MLP for output.

## 3. Experiments

### 3.1. The roles of Reaction and 3D Information

#### 3.1.1. THE EFFECT OF REACTION INFORMATION

**Attention Weights Visualization.** To illustrate the advantages of Reaction Graph (RG) compared to Molecular Graph (MG), we train a condition prediction model on USPTO[1] and visualize the attention weights $\alpha_i$ of reactions. Attention weights can display the model's focus on different parts of molecules, especially reaction centers, revealing the model's understanding of the reaction mechanism. As shown in Fig. 3, we take 3-Amino-5-bromobenzoic acid ($C_7H_6BrNO_2$) and its two related reactions as examples.

---

[1] https://figshare.com/articles/dataset/Chemical_reactions_from _US_patents_1976-Sep2016_/5104873

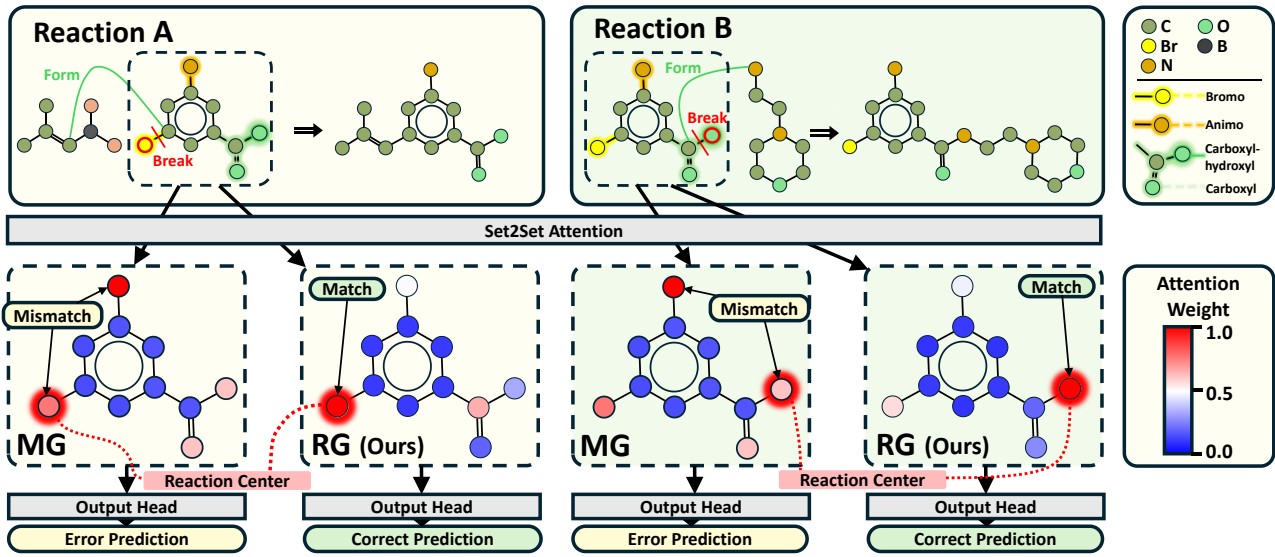

*Figure 3.* Visualization of attention weights and prediction results for two reactions involving the bromo and carboxyl-hydroxyl groups in $C_7H_6BrNO_2$. The colors of the atoms in the upper diagrams correspond to the types of atoms, while the colors of the atoms in the lower diagrams correspond to the sizes of the atomic attention weights. The model using the Molecular Graph (MG) focuses on atoms that are less relevant to the reaction, thus leading to prediction errors. In contrast, the model equipped with the Reaction Graph (RG) accurately concentrates on the reaction center, and produces the correct prediction results. For more results, please refer to Sec. H.1.

The $C_7H_6BrNO_2$ features three active functional groups: bromo, amino, and carboxyl-hydroxyl. In reaction $A$, the bromo group acts as the reaction center, while the carboxyl-hydroxyl group serves this role in reaction $B$.

As depicted in Fig. 3. In both reactions, the MG-based model focuses more on the non-reactive amino group and insufficiently on reaction centers, resulting in prediction errors. In contrast, the RG-empowered model pays correct attention to reaction centers and provides reasonable reaction conditions. The experiment results validate our hypothesis: MG, which represents reactants and products independently, struggles to capture atom and bond transformations during the reaction process, while RG helps the model accurately locate the reaction center and extract relevant features of reaction changes. More results can be found in Sec. H.1.

**Leaving Group Identification.** We design the Leaving Group (LvG) identification task to further validate the effectiveness of RG. LvG refers to the atomic group that is present in the reactants and detaches from the products during a reaction, which is closely related to the reaction mechanism (Wang et al., 2023c). LvG identification is a node-level multi-class classification task, where the node label specifies whether an atom belongs to a LvG and its type. This requires the model to focus not only on the features of the molecule itself but also on reaction-related features.

Both models based on MG and RG are trained on the LvG dataset extracted from USPTO. The evaluation metrics include accuracy (ACC), confusion entropy (CEN), the multi-

*Table 1.* Leaving group (LvG) identification results of Molecular Graph (MG) and Reaction Graph (RG) representations, with overall and LvG atom-specific evaluation.

| Rep. | Overall | | | | LvG Atom-Specific | | | |
|---|---|---|---|---|---|---|---|---|
| | ACC↑ | CEN↓ | MCC↑ | F1↑ | ACC↑ | CEN↓ | MCC↑ | F1↑ |
| MG | 0.950 | 0.036 | 0.549 | 0.365 | 0.448 | 0.201 | 0.519 | 0.404 |
| RG (ours) | **0.997** | **0.002** | **0.973** | **0.904** | **0.947** | **0.031** | **0.945** | **0.903** |

class Matthews Correlation Coefficient (MCC), and the Macro F1 Score (F1). CEN assesses the misclassification level, while MCC and F1 measure accuracy accounting for the imbalance of sample categories. We report relevant metrics for all atoms and LvG atoms, separately.

As shown in Tab.1, RG outperforms MG on all metrics. Compared to MG, RG improves the overall ACC, MCC and F1 by 4.7%, 42.4% and 53.9%, respectively. For LvG atoms, RG achieves an ACC of 94.7%, which is twice that of the MG. The advantage of RG on LvG identification demonstrates its ability to understand the reaction mechanism. More comparisons and visualizations are in Sec. H.2.

### 3.1.2. THE EFFECT OF 3D INFORMATION

**Settings.** In this section, we explore the effects of incorporating various 3D information in RG. Specifically, we investigate: (1) no 3D information, (2) only bond edge length, (3) bond edge length and bond angle, as well as (4) bond edge length and angular edge length. We conduct experiments on the USPTO-Condition dataset to evaluate the accuracy.

*Table 2.* Top-$k$ accuracy of reaction condition prediction on the USPTO-Condition and Pistachio-Condition datasets. (*) indicates that the result is sourced from Wang et al. (2023b).

| Method | USPTO-Condition | | | | | Pistachio-Condition | | | | |
|---|---|---|---|---|---|---|---|---|---|---|
| | Top-1↑ | Top-3↑ | Top-5↑ | Top-10↑ | Top-15↑ | Top-1↑ | Top-3↑ | Top-5↑ | Top-10↑ | Top-15↑ |
| CRM (Gao et al., 2018) | 0.260* | 0.377* | 0.421* | 0.461* | 0.472* | 0.330 | 0.469 | 0.510 | 0.548 | 0.554 |
| Parrot (Wang et al., 2023b) | 0.269* | 0.404* | 0.451* | 0.491* | 0.503* | 0.350 | 0.532 | 0.588 | 0.626 | 0.630 |
| AR-GCN (Maser et al., 2021) | 0.146* | 0.237* | 0.273* | 0.312* | 0.326* | - | - | - | - | - |
| CIMG (Zhang et al., 2022) | 0.184* | 0.271* | 0.303* | 0.339* | 0.353* | - | - | - | - | - |
| D-MPNN (Heid & Green, 2021) | 0.198 | 0.300 | 0.334 | 0.378 | 0.392 | 0.259 | 0.342 | 0.378 | 0.442 | 0.469 |
| Rxn Hypergraph (Tavakoli et al., 2022) | 0.213 | 0.308 | 0.345 | 0.381 | 0.393 | 0.288 | 0.367 | 0.412 | 0.464 | 0.485 |
| Reaction Graph (ours) | **0.325** | **0.434** | **0.472** | **0.506** | **0.518** | **0.392** | **0.557** | **0.604** | **0.638** | **0.643** |

*Table 3.* Influence of different types of 3D information on the USPTO-Condition dataset. Experimental groups include no 3D information, bond edge length, bond edge length and bond angle, as well as bond edge length and angular edge length.

| 3D Information | Accuracy |
|---|---|
| Without 3D Information | 0.3133 |
| Bond Edge Length | 0.3165 |
| Bond Edge Length + Bond Angle | 0.3179 |
| Bond Edge Length + Angular Edge Length | **0.3246** |

To assess computational efficiency, we further design the following experiments. Inspired by bin-packing (Cormen et al., 2022), we select 16 sets of chemical reactions from USPTO-Condition, ensuring that each set contains the same number of atoms, with quantities ranging from 1500 to 4500. Subsequently, the 16 sets of reactions are input into the condition prediction model, and the average runtime is measured to indicate computational efficiency.

**Accuracy Evaluation.** According to Tab. 3, 3D information effectively improves the model's performance. Specifically, the RG equipped with bond edge length and angular edge length achieves the best performance. Angular edge length is more effective than directly using bond angle. This is because the angular edge length is integrated into the GNN as part of the graph structure. The geometric consistency helps to more accurately maintain the spatial relationship of the molecule. In contrast, the bond angle needs to be treated separately from the bond edge length, which may distort the original geometric continuity and integrity of the molecule.

**Efficiency Evaluation.** The results in Fig. 4 suggest that incorporating bond length brings almost no extra computational overhead. Besides, compared to bond angle, using angular edge length can significantly reduce the inference time. Moreover, according to the curve steepness, the time cost associated with using bond angle rises more significantly as the number of atoms increases. Hence, when integrating 3D molecular information into RG, we ultimately employ bond edge length and angular edge length, enhancing accuracy while maintaining efficiency.

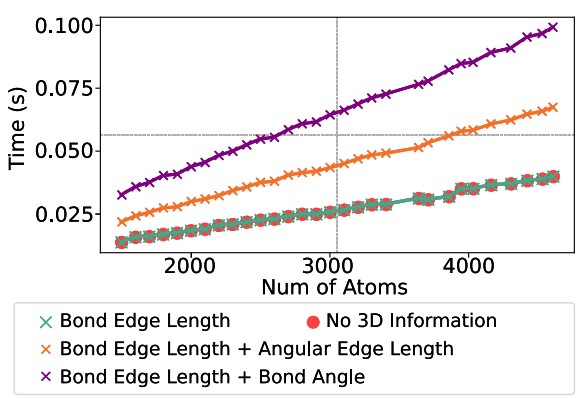

*Figure 4.* Influence of different methods of 3D structure modeling on running time. Compared to using bond angles, the proposed angular edge method effectively reduces inference time.

### 3.2. Reaction-related Tasks

#### 3.2.1. REACTION CONDITION PREDICTION

**Dataset.** The USPTO-Condition dataset is derived from Parrot (Wang et al., 2023b), comprising over 680K samples, divided into 80% for training, 10% for validation, and 10% for testing. Besides, we construct Pistachio-Condition from the Pistachio database by thorough cleaning and filtering. It includes over 560K samples, with a training, validation, and testing split of 8:1:1.

**Evaluation Metrics.** Following Gao et al. (2018); Wang et al. (2023b), we use the top-$k$ accuracy to evaluate the condition prediction performance.

**Comparison Methods.** CRM (Gao et al., 2018) utilizes molecular fingerprints. Parrot (Wang et al., 2023b) employs SMILES. AR-GCN (Maser et al., 2021) and CIMG (Zhang et al., 2022) use MG. D-MPNN (Heid & Green, 2021) leverages the condensed graph of reactions (CGR) (Varnek et al., 2005), and Rxn Hypergraph (Tavakoli et al., 2022) employs its own designed graph representation.

**Results.** The performance comparisons are reported in Tab. 2. Our method outperforms all the comparison meth-

*Table 4.* Regression accuracy ($R^2$ ↑) for reaction yield prediction on the Buchwald-Hartwig (B-H), Suzuki-Miyaura (S-M), Gram, and Subgram datasets. B-H-1, B-H-2, B-H-3 and B-H-4 are more challenging splits of the B-H dataset. (*) indicates the results are reported from the original paper.

| Method | Representation | B-H | B-H-1 | B-H-2 | B-H-3 | B-H-4 | S-M | Gram | Subgram |
|---|---|---|---|---|---|---|---|---|---|
| DRFP* (Probst et al., 2022) | Fingerprint | 0.95 | 0.81 | 0.83 | 0.71 | 0.49 | 0.85 | **0.130** | 0.197 |
| Yield-Bert* (Schwaller et al., 2021b) | SMILES | 0.95 | **0.84** | 0.84 | 0.75 | 0.49 | 0.82 | 0.117 | 0.195 |
| Egret* (Yin et al., 2024) | SMILES | 0.94 | **0.84** | **0.88** | 0.65 | 0.54 | 0.85 | 0.128 | 0.206 |
| UGNN (Kwon et al., 2022b) | Molecular Graph | **0.97*** | 0.74* | **0.88*** | 0.72* | 0.50* | **0.89*** | 0.117 | 0.190 |
| D-MPNN (Heid & Green, 2021) | CGR | 0.94 | 0.80 | 0.82 | 0.73 | 0.55 | 0.85 | 0.125 | 0.202 |
| Rxn Hypergraph (Tavakoli et al., 2022) | Rxn Hypergraph | 0.96 | 0.81 | 0.83 | 0.71 | 0.56 | 0.85 | 0.118 | 0.196 |
| Reaction Graph (ours) | Reaction Graph | **0.97** | 0.80 | **0.88** | **0.76** | **0.68** | **0.89** | 0.129 | **0.216** |

*Table 5.* Influence of reaction information (Reaction Edge) and 3D structure on the prediction of chemical reaction conditions.

| Dataset | Reaction Edge | 3D Structure | ACC↑ |
|---|---|---|---|
| USPTO Condition | ✗ | ✗ | 0.3050 |
| | ✗ | ✓ | 0.3090 |
| | ✓ | ✗ | 0.3133 |
| | ✓ | ✓ | **0.3246** |
| Pistachio Condition | ✗ | ✗ | 0.3806 |
| | ✗ | ✓ | 0.3819 |
| | ✓ | ✗ | 0.3852 |
| | ✓ | ✓ | **0.3915** |

*Table 6.* Likelihood $(y - y')^2/\sigma^2$ and log variance $\log \sigma^2$ metrics on the Gram and Subgram datasets, where likelihood reflects the consistency between predicted variance and regression error, and log variance reflects the size of variance. Within these methods, only UGNN has variance output.

| Methods | Likelihood↓ | | Log Variance↓ | |
|---|---|---|---|---|
| | Gram | Subgram | Gram | Subgram |
| UGNN (Kwon et al., 2022b) | **1.02** | 1.20 | 6.94 | 7.36 |
| Reaction Graph (ours) | 1.06 | **1.18** | **5.86** | **6.14** |

ods on both datasets, demonstrating the superiority of RG on reaction feature modeling. On USPTO-Condition, compared to domain models with 1D and 2D representations, our method improves the top-1 accuracy by 17.2% and 76.6%, respectively. Compared with graph-based methods, RG improves the top-$k$ accuracy by an average of 39.0%. On Pistachio-Condition, RG also demonstrates its advantage by surpassing other methods by 3.4%-18.8%.

**Ablation Study.** The reaction information and 3D structure are key components of RG. We evaluate their respective effects on modeling chemical reactions. Results in Tab. 5 reveal that both the reaction information and 3D structure in RG effectively enhance model performance. Specifically, on USPTO-Condition, the utilization of reaction information and 3D structure brings average performance improvements of 3.9% and 2.5%, respectively; while in Pistachio, the improvements are 1.9% and 1.0%. Moreover, the reaction information and 3D information are complementary, and their combination results in improvements of 6.4% and 2.9% on USPTO-Condition and Pistachio-Condition, respectively.

### 3.2.2. REACTION YIELD PREDICTION

**Dataset.** Buchwald-Hartwig (B-H) (Ahneman et al., 2018) involves six molecules as reactants, with products comprised of a single molecule. B-H is used to create B-H-1 to B-H-4 through different train-test splits, with increasing challenges due to distribution differences. The molecule number involved in each reaction varies in Suzuki-Miyaura (S-M) (Perera et al., 2018). USPTO-Yield (Schwaller et al., 2021b) is divided into Gram and Subgram. We also notice that in the small-scale B-H and S-M datasets, there are only dozens of different molecular types, some of which are reagents; meanwhile, the USPTO-Yield dataset contains a significant amount of noise. This makes it difficult for the model to capture the relatively complex and variable 3D information, preventing it from learning the correct 3D priors. Therefore, we only test the role of reaction information in the yield prediction task.

**Evaluation Metrics.** The proposed method simultaneously outputs the mean $y$ and variance $\sigma^2$ of the predicted yield. Following Schwaller et al. (2021b); Kwon et al. (2022b), we use the $R^2$ score to evaluate the accuracy of the output mean. We additionally introduce likelihood $(y - y')^2/\sigma^2$ and log variance $\log \sigma^2$ from negative log-likelihood (Lakshminarayanan et al., 2017) to evaluate the output variance, where $y'$ is the ground truth.

**Comparison Methods.** DRFP (Probst et al., 2022) utilizes reaction fingerprint. Yield-Bert (Schwaller et al., 2021b) and Egret (Yin et al., 2024) are based on SMILES. UGNN (Kwon et al., 2022b) employs MG and simultaneously predicts yield and uncertainty. D-MPNN (Heid & Green, 2021) employs CGR, while Rxn Hypergraph (Tavakoli et al., 2022) uses its uniquely designed graph representation.

**Results.** As shown in Tab. 4, compared to other advanced methods, RG achieves the highest accuracy on six out of eight yield prediction datasets. Especially on the more challenging B-H-4 and Subgram datasets, RG achieves improvements of 21.4% and 4.9%, respectively. Besides, according to Tab. 6, both methods provide uncertainty that accurately

*Table 7.* Reaction classification results on the USPTO-TPL and Pistachio-Condition datasets. Evaluation metrics include accuracy (ACC), confusion entropy (CEN), Matthews Correlation Coefficient (MCC) and Macro F1 (F1). (*) indicates that the result is sourced from the original paper.

| Method | USPTO-TPL | | | Pistachio-Type | | |
|---|---|---|---|---|---|---|
| | ACC↑ | CEN↓ | MCC↑ | ACC↑ | CEN↓ | MCC↑ |
| DRFP | 0.977* | 0.011* | 0.977* | 0.899 | 0.149 | 0.890 |
| RXNFP | 0.989* | 0.006* | 0.989* | 0.948 | 0.078 | 0.944 |
| T5Chem | 0.995* | 0.003* | 0.995* | 0.976 | 0.041 | 0.974 |
| D-MPNN | 0.997 | 0.001 | 0.997 | 0.982 | 0.033 | 0.980 |
| Rxn Hypergraph | 0.954 | 0.024 | 0.953 | 0.911 | 0.129 | 0.903 |
| Reaction Graph (ours) | **0.999** | **0.001** | **0.999** | **0.987** | **0.024** | **0.986** |

*Table 8.* Influence of the proposed reaction information (Reaction Info) and 3D structure (3D Stru) modeling methods on the USPTO-Condition and Pistachio-Condition datasets, using ACC, CEN and MCC as classification metrics.

| Reaction Info | 3D Stru | USPTO-TPL | | | Pistachio-Type | | |
|---|---|---|---|---|---|---|---|
| | | ACC↑ | CEN↓ | MCC↑ | ACC↑ | CEN↓ | MCC↑ |
| ✗ | ✗ | 0.9921 | 0.0037 | 0.9921 | 0.9658 | 0.0559 | 0.9627 |
| ✗ | ✓ | 0.9955 | 0.0021 | 0.9955 | 0.9669 | 0.0538 | 0.9640 |
| ✓ | ✗ | 0.9978 | 0.0010 | 0.9977 | 0.9862 | 0.0262 | 0.9850 |
| ✓ | ✓ | **0.9991** | **0.0004** | **0.9991** | **0.9873** | **0.0242** | **0.9862** |

reflects the actual error levels, while RG further reduces prediction uncertainty by 12.3% on Gram and 13.3% on Subgram. However, the quality and complexity of the Gram and Subgram datasets restrict further performance improvement in existing yield prediction methods.

### 3.2.3. REACTION CLASSIFICATION

**Dataset.** The USPTO-TPL is from Schwaller et al. (2021a), with labels generated by 1000 reaction templates, making it relatively simple. We construct the more challenging Pistachio-Type dataset from Pistachio, with labels generated by NameRXN[2] based on rules.

**Evaluation Metrics.** Similar to Schwaller et al. (2021a); Lu & Zhang (2022), we use accuracy (ACC), confusion entropy (CEN), Matthews Correlation Coefficient (MCC) and Macro F1 (F1) to evaluate the performance. CEN assesses misclassifications to quantify the uncertainty of predictions, while MCC and F1 provide a more comprehensive measure of classification accuracy.

**Comparison Methods.** DRFP (Probst et al., 2022) uses reaction fingerprint. RXNFP (Schwaller et al., 2021a) and T5Chem (Lu & Zhang, 2022) are based on SMILES. D-MPNN (Heid & Green, 2021) employs CGR, while Rxn Hypergraph (Tavakoli et al., 2022) relies on its own graph representations.

---

[2]https://www.nextmovesoftware.com/namerxn.html

**Results.** According to Tab. 7, RG surpasses advanced models on both USPTO-TPL and Pistachio-Type, demonstrating the effectiveness of the proposed designs. Compared to the state-of-the-art T5Chem, RG reduces the classification error by 66.6% and achieves nearly 100% accuracy on USPTO-TPL. The superior performance on USPTO-TPL is due to the limited number of reaction templates, which simplifies the classification task. RG's precise identification of the reaction center (detailed in Sec. 3.1.1) enhances template discrimination capability, bringing further performance improvements. On the complex Pistachio-Type dataset, RG exceeds the best performance by 1.2% on MCC and 1.1% on F1, highlighting its superiority in modeling reactions.

**Ablation Study.** We investigate the influence of reaction information and 3D structure in RG on the reaction classification task. As shown in Tab. 8, integrating reaction information reduces classification error by an average of 77% on USPTO-TPL and 54.1% on Pistachio-Type. On the other hand, 3D structure can also enhance the accuracy across both datasets. The results suggest that reaction information and 3D structures mutually enhance each other, improving the understanding of reaction mechanisms.

## 4. Conclusion

In this paper, we propose a unified 3D Reaction Graph (RG) for chemical reaction modeling. Unlike existing methods, the RG is equipped with enhanced capabilities for modeling reaction changes and 3D structures. We conduct extensive experiments across various tasks and datasets, demonstrating RG's effectiveness in understanding chemical reactions. Furthermore, since it is independent of any specific GNN architecture, the RG representation may show increased potential as the underlying network backbone is improved.

**Limitations and Future Work.** Like most data-driven methods, the quality of data significantly impacts the performance of our method. Specifically, inaccuracies in the 3D coordinates of atoms can lead to inferior results. Thus, developing an advanced method for 3D prediction could further enhance our approach, which can be investigated in the future.

## Acknowledgements

This work was supported by the National Science and Technology Major Project (2023ZD0120803), the National Natural Science Foundation of China (62472381) and the Earth System Big Data Platform of the School of Earth Sciences, Zhejiang University.

## Impact Statement

This paper aims at enhancing deep representation learning in chemistry. The proposed reaction representation has the potential to accelerate drug design and material development, saving substantial amounts of money and time. Although there may be concerns of misuse, such as manufacturing illegal drugs, we strongly believe that the advantages of our method far outweigh the minuscule risk of misuse.

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

# Reaction Graph: Toward Modeling Chemical Reactions with 3D Structures

## Appendix

## Contents

## A. Uniqueness of 3D Reaction Modeling

### A.1. Differences between Reaction Graph and Retrosynthesis

Reaction Graph can predict conditions and yields for Retrosynthesis. Retrosynthesis utilizes atom mapping and 3D information, which also inspires the design of Reaction Graph. However, there are key differences between Reaction Graph and Retrosynthesis (Laabid et al., 2024) methods:

1. Reaction Graph uses the entire reaction as input. Retrosynthesis uses only the product as input. Therefore, the task settings are different.

2. Reaction Graph is design to predict transformation invariant reaction properties, so it focuses on being invariant to rotation, translation and even bond torsion. Retrosynthesis methods are designed to generate reactant structure, so they focus on geometric completeness. When geometrically complete, a method will not be invariant to bond torsion. Therefore, the design focuses are different.

3. Reaction Graph takes atom mapping as an input, aiming to enable message passing between reactants and products, and provide reaction change information. Retrosynthesis methods output atom mapping to avoid directly outputting the reactants, thereby reducing difficulty. Therefore, the roles of atom mapping are different.

4. Reaction Graph models the relationship between reaction 3D structure and reaction properties, while the Retrosynthesis methods model the relationship between the product 3D structures and reactant 3D structure. Therefore, the functions of 3D information are different.

### A.2. Differences between Reaction Graph and Protein Docking/Binding

The 3D interaction modeling in Protein Docking provides inspiration for Reaction Graph. However, there are key differences between Reaction Graph and Protein Docking (Ganea et al., 2021)/Binding (Stärk et al., 2022):

1. Reaction Graph is independent of specific 3D scene, thus it emphasizes invariance. In contrast, Protein Docking/Binding predicts the transformation/structure of the ligand in a specific 3D scene, thus it emphasizes equivariance and geometric completeness. Therefore, they have different concerns.

2. Reaction Graph models the interaction between reactants and products. Protein Docking/Binding models the interaction between the ligand and receptor, which is the interaction between reactant and reactant. Therefore, Therefore, their interaction types are different.

3. Reactants and products are the same set of atoms, but exist in different space and different time. Ligand and receptor are different sets of atoms, but exist in the same space and same time. As a result, Reaction Graph models interaction by atomic mapping, while Protein Docking/Binding models interaction by spatial relationship. Therefore, their interaction modelings are different.

## B. Related Works

### B.1. Molecular Representation

Research interest in molecular representation learning is growing due to its potential in various biochemical tasks like virtual screening and inverse design. To enhance the expressive capabilities of molecular representations, efforts are focused on developing network architectures and training strategies suited to different modalities of molecular input. 1D molecular fingerprint (Morgan, 1965; Durant et al., 2002; Rogers & Hahn, 2010) and SMILES string (Weininger, 1988; O'Boyle & Dalke, 2018; Krenn et al., 2020) are typically processed by language models (Jaeger et al., 2018; Wang et al., 2019; Chithrananda et al., 2020) to extract chemical properties. GNN-based methods (Duvenaud et al., 2015; Kearnes et al., 2016; Xiong et al., 2019) are commonly used to model 2D molecular graphs, which intuitively simulate the relationships between atoms (nodes) and bonds (edges). Recently, the integration of high-dimensional geometric information, including molecular point clouds and 3D molecular graphs (Schütt et al., 2017a; Gasteiger et al., 2020; Atz et al., 2021; Fang et al., 2022; Zhou et al., 2023; Han et al., 2024), has effectively assisted in understanding complex molecular structures.

## B.2. Reaction Representation

Representing chemical reactions is crucial for scientific discovery. A well-designed reaction representation can facilitate the development of various tasks, such as reaction classification (Ghiandoni et al., 2019; Schwaller et al., 2019; Lu & Zhang, 2022), condition recommendation (Gao et al., 2018; Maser et al., 2021; Kwon et al., 2022a; Wang et al., 2023b), and yield prediction (Schwaller et al., 2021b; Kwon et al., 2022b; Yin et al., 2024). To represent chemical reactions, researchers have developed novel fingerprint (Schneider et al., 2015; Probst et al., 2022), graph representation (Varnek et al., 2005; Tavakoli et al., 2022), and deep learning-based methods (Schwaller et al., 2021a; Hou & Dong, 2023). Recently, some studies have introduced strategies such as multi-modal integration (Chen et al., 2024; Zhang et al., 2024) and pre-training (Wen et al., 2022; Shi et al., 2024), providing new insights for constructing reaction representations. However, the reaction representation methods do not pay as much attention to 3D spatial information as molecular representation does. Furthermore, current approaches generally represent reactants and products separately, overlooking the modeling of chemical changes during the reaction process.

## B.3. Invariant Neural Network

To enhance model's understanding of 3D molecular structure in reaction, we integrate 3D information into RG in an invariant manner. Invariance ensures that the prediction results remain unaffected by the rotation and translation of molecules. This reduces redundant information, allowing the model to focus on reaction-related features.

Invariant neural network utilizes transformation-invariant features, such as distance and angle (Han et al., 2024). Initially, DTNN (Schütt et al., 2017b) utilizes distances between atoms with Gaussian basis embedding. Later, SchNet (Schütt et al., 2017a) improves performance by using learnable radial basis function (RBF) kernel embeddings. DimeNet (Gasteiger et al., 2020) is the first to introduce angle features in quantum chemistry property prediction tasks, and GemNet (Gasteiger et al., 2021) further enhances this by incorporating dihedral (torsion) angles. Subsequently, ComENet (Wang et al., 2022) and SphereNet (Liu et al., 2022) introduce new torsion angle representations to reduce computational costs. Additionally, models like ClofNet (Du et al., 2022) and LEFTNet (Du et al., 2024) use a local coordinate system to guarantee invariance.

**Difference.** While these efforts are dedicated to providing more comprehensive 3D features, we try to take a step back and explore more **suitable 3D features**. To be detailed, the properties of chemical reactions (e.g. condition, yield, type) are invariant to different conformations. Therefore, the 3D features in RG should also be invariant to different conformations. In different conformations, the bond lengths and bond angles exhibit minimal variation, thus they are used in RG. However, the torsion angles and pairwise distances may change dramatically, thus they are excluded to further reduce redundant information. We also demonstrate the effectiveness of our design in Sec. H.8.1.

## B.4. Reaction Condition Prediction

Reaction conditions are pivotal for chemical synthesis and drug design. Suitable reaction conditions can significantly accelerate reactions, increase yields, and save raw materials. Condition prediction is an essential step in computer-aided synthesis planning (CASP). Given the representation of a reaction, the model is designed to predict conditions like catalysts, solvents and temperature.

In the early stages, machine learning models used for predicting reaction conditions are primarily based on knowledge graph reasoning (Segler & Waller, 2017), database similarity searches (Lin et al., 2016), and expert systems (Marcou et al., 2015). With (Gao et al., 2018) and others pioneering the use of neural networks trained on large datasets for the prediction of reaction conditions, increasing attention has been given to the potential of deep learning in this task. Researchers attempt to improve upon the inputs to the network. (Afonina et al., 2021) uses ISIDA fingerprints as descriptors for reactions, (Walker et al., 2019) uses MACCS keys as inputs, and (Chen & Li, 2024) uses the difference between the product and reactant Morgan fingerprints as input. Other researchers focus on enhancements in network architecture. (Ryou et al., 2020), (Maser et al., 2021), and (Kwon et al., 2022a) use GNNs for feature extraction, while (Andronov et al., 2023) and (Wang et al., 2023b) attempt to use Transformers to extract features of reactions. Additionally, (Kwon et al., 2022a) and (Karpovich et al., 2023), aiming to address the one-to-many relationship between reactions and reaction conditions, opt to use Variational Autoencoders (VAEs) (Kingma & Welling, 2014) for reaction condition generation.

## B.5. Reaction Yield Prediction

Yield typically indicates the percentage of reactant molecules converted into the desired product, and is one of the main concern of synthetic route planning. High-yield synthetic routes can reduce costs and improve product purity. To enhance the yield of the synthetic routes planning result, an accurate yield prediction model is vital.

Similar to condition prediction tasks, early approaches utilize DFT (Ahneman et al., 2018) for yield prediction. These methods are generally applied to specific types of chemical reactions. With the advent of Transformers, efforts to achieve yield prediction using language models like BERT emerge, exemplified by models such as YieldBERT (Schwaller et al., 2021b) and Egret (Yin et al., 2024). Additionally, there are methods employing GNNs (Kwon et al., 2022b; Li et al., 2023; Sato et al., 2024). Recently, researchers have begun exploring Bayesian Neural Networks (BNNs) (Gal & Ghahramani, 2016; Kendall & Gal, 2017), which output both prediction and confidence (Kwon et al., 2022b; Chen et al., 2024). Some studies also attempt to integrate reaction conditions or multimodal information to improve practical applicability and model accuracy (Yin et al., 2024; Chen et al., 2024), while the potential of pre-training remains to be explored (Shi et al., 2024).

## B.6. Reaction Classification

Reaction type is closely related to the reaction mechanism. Classification of reaction type is beneficial for reaction understanding, and can be used for data augmentation and preprocessing. Additionally, reaction classification task evaluates model's comprehension of reaction mechanisms, contributing to performance analysis.

Earlier, people use rule-based methods to classify chemical reactions (Ghiandoni et al., 2019). These methods are accurate and have strong interpretability, but they require a huge amount of labor. In recent years, with the rise of machine learning, data-driven methods begin to emerge (Schwaller et al., 2019). RXNFP (Schwaller et al., 2021a) employs Bert to extract implicit vector representations of reactions from SMILES strings, achieving excellent clustering results for reaction types. DRFP (Probst et al., 2022) combines circular fingerprints and hash-based fingerprint generation methods to efficiently represent a chemical reaction in an interpretable manner. T5Chem (Lu & Zhang, 2022) utilizes a specially designed multi-task decoder for reaction type classification, enhancing the model's understanding of reaction mechanisms. Furthermore, works like Egret (Yin et al., 2024) use reaction type classifiers trained on the Pistachio dataset for data analysis, aiming to evaluate the model's yield prediction capabilities.

# C. Molecule and Reaction Representations

In deep learning tasks, the representations of molecules and reactions are used to describe their chemical features. In terms of modality, these representations can be divided into 1D strings and fingerprints, 2D graphs, and 3D point clouds. Tab. 9 lists examples of various representation forms for aspirin, and Tab. 10 lists the representations of chemical reaction for synthesizing aspirin.

## C.1. One-Dimensional

### C.1.1. STRING

String representations are typically composed of a sequence of characters that express molecular structure. They follow specific grammars that can be parsed by computer programs, and is generally human-readable. As a result, string representations are widely used in chemical databases. Due to their similarity to natural language, Transformers and RNNs are often employed to extract features from string representations. For these models, the main challenge lies in the many-to-one relationship between strings and molecules. This hinders models from learning molecular structure patterns. To mitigate this problem, canonical algorithms and data augmentation are generally employed.

**SMILES.** (Weininger, 1988) expression is the most commonly used string representation in deep learning chemistry tasks. It is composed of symbols representing atoms or groups, symbols for bonds, labels for rings, case rules for indicating aromaticity, and symbols for chirality. These symbols work together to represent the atomic composition and topological structure of a molecule.

The SMILES representation of aspirin, as shown in Tab. 9, uses atomic symbols to represent the atoms in the molecule. For example, the symbol for carbon is $C$, for oxygen is $O$, and bromine, which does not appear in the example, has the symbol $Br$. Hydrogen atoms are typically omitted. For chemical bonds, a single bond is represented by a dash (-), which is usually

*Table 9.* Example of different types of molecular representations of aspirin, including string-based representations such as SMILES, SELFIES, DeepSMILES, fingerprint-based representations like ECFP and MACCS, molecular graph and improvements, as well as 3D molecular point cloud (graph).

| Type | Representation |
|---|---|
| SMILES | $CC(=O)Oc1ccccc1C(=O)O$ |
| ECFP | [ 1 1 1 0 1 1 0 1 1 0 1 1 0 1 0 1 1 1 1 0 0 0 1 1 0 1 0 0 1 1 0 0 ] |
| Molecular Graph |  |
| Point Cloud |  |

omitted, a double bond is represented by an equals sign (=), and a triple bond is represented by a hash sign (#). It's important to note the presence of parentheses in the structure, which indicate a branch; for instance, the expression $C(=O)$ signifies a branch with an oxygen atom connected to the carbon by a double bond. Additionally, we observe the structure $c1ccccc1$, where the numbers 1 at both ends indicate that the corresponding carbon atoms are connected by a bond to form a ring. The lowercase letter indicates that this is an aromatic ring. Furthermore, SMILES may include functional groups or atoms with valence states enclosed in brackets, such as $[O-]$ and $[C@H]$. Here, $[O-]$ denotes an oxygen atom with a negative charge, and the (@) symbol typically indicates a chiral center, with different chiralities distinguished by $[C@H]$ and $[C@@H]$. (.) symbol are used to separate the SMILES expressions of different molecules. For chemical reactions, the SMILES notation generally follows the format of $reactants > reagents > products$, as shown in Tab. 10, which does not include reagents.

Mainstream Python libraries such as RDKit[3] and OpenBabel[4] can parse or generate SMILES expressions for molecules or chemical reactions, and efficiently calculate various chemical properties.

**SMARTS.** (Daylight Chemical Information Systems, 2007) is an extension of SMILES used to describe patterns or substructures within molecular structures. It allows the use of wildcards and logical operators to represent more complex chemical queries. For chemical reactions, SMARTS strings can include atomic mapping information. For example, in the aspirin synthesis reaction shown in Tab. 10, symbol $[C : 2]$ can be found in both reactants and products. This indicates that they are the same atom. With atomic mapping information, reaction mechanisms can be easily interpret. Tools like RXNMapper[5] can be utilized to predict atomic mapping.

### C.1.2. FINGERPRINT

Another 1D representation is fingerprint. They typically manifest as binary vectors, where each bit indicates the presence or absence of a substructure or chemical property. These vectors are generally of fixed length, making them easily manageable by most neural network architectures, such as MLPs.

Fingerprints can enhance model's performance on specific tasks by selecting task-relevant features; however, this relies on careful manual design and has limited generalization ability. Additionally, fingerprints have one-to-many relationship with

---

[3]https://www.rdkit.org/

[4]https://openbabel.org/index.html

[5]https://github.com/rxn4chemistry/rxnmapper

*Table 10.* Example of different types of reaction representations, including string-based SMILES and SMARTS, fingerprint-based DRFP and ISIDA, graph representations Molecular Graph, CGR, Rxn Hypergraph and Reaction Graph, as well as 3D reaction point cloud.

| Type | Representation |
| --- | --- |
| SMILES | $c1ccc(c(c1)C(=O)O)O.CC(=O)OC(=O)C$ $>> CC(=O)Oc1ccccc1C(=O)O$ |
| SMARTS | $CC(=O)O[C:2]([CH3:1])=[O:3].[OH:4][c:5]1[cH:6]$ $[cH:7][cH:8][cH:9][c:10]1[C:11](=[O:12])[OH:13]$ $>> [CH3:1][C:2](=[O:3])[O:4][c:5]1[cH:6][cH:7]$ $[cH:8][cH:9][c:10]1[C:11](=[O:12])[OH:13]$ |
| DRFP | $[\,0\ 1\ 1\ 1\ 1\ 0\ 1\ 0\ 0\ 0\ 0\ 0\ 1\ 1\ 1\ 1\ 1\ 0\ 1\ 1\ 0\ 1\ 1\ 0\ 1\ 1\ 0\ 1\ 0\ 0\ 0\ 0\,]$ |
| Molecular Graph |  |
| CGR |  |
| Rxn Hypergraph |  |
| Reaction Graph (Omit Angular Edges and 3D) |  |

molecules or reactions.

**ECFP.** (Rogers & Hahn, 2010) generates fingerprints by traversing the atoms of a molecule and their surrounding environments, capturing the topological structure information of the molecule. The ECFP generation algorithm starts by assigning an initial identifier to each atom based on its features. Then, it iteratively combine the feature of each atom with its neighbors. The features of the $n$-th iteration contains the topological information of a radius of $n+1$ around the atom. Ultimately, these atomic features are combined into a fixed-length bit vector, as shown in Tab. 9.

ECFP is generally designed for molecules, and the ECFP of a reaction is generally a concatenation or difference of the

reactant's and product's ECFP.

ECFP can be generated using popular chemical toolkits like RDKit (referred as Morgan Fingerprint). It is efficient to construct and easy to input into various machine learning models. However, the expressive ability of the ECFP is limited and depends on its length (where the ECFP in condition prediction task can reach the length of up to $2^{14}$ bits).

**DRFP.** (Probst et al., 2022) shares similarities with ECFP. DRFP first extracts a list of circular molecular n-grams from the reactants and products, then calculates the symmetric difference of the n-gram lists. For each n-gram in the resulting symmetric difference list, a descriptor is computed. Finally, the set of descriptors is converted into a fixed-length vector. This approach is akin to the differentiation method in ECFP, emphasizing the changes that occur before and after the reaction. DRFP can be generated using the Python DRFP[6] library.

### C.2. Two-Dimensional

In 2D graph representations, each node corresponds to an atom, while edges correspond to bonds. Each node and edge has associated attributes to express its chemical features(e.g. atom type and bond type). Specially designed molecular graphs or reaction graphs may incorporate additional nodes and edges. They can express additional information or enhance message passing. Graph representations are usually processed by GNNs or Graph Transformers.

Due to the similarity between graphs and molecules, graph representations possess advantage in expressing molecular properties. It can correspond one-to-one with molecules, aiding neural network in modeling the correct relationships between them. However, 2D molecular graph may struggle to model long-range atomic interaction, and lack 3D structural information. Improving graph representations for more efficient feature extraction is currently a key area of research.

**Molecular Graph (MG).** (Duvenaud et al., 2015) uses graphs to represent molecules, where nodes correspond to atoms and edges correspond to bonds. Both nodes and edges carries chemical information. Such graphs are typically undirected and stored as an edge matrix of size $E \times 2$, or an adjacency matrix of size $V \times V$. Additionally, there are atom and bond feature matrices of size $V \times D_V$ and $E \times D_E$.

As shown in Tab. 9 and 10, the visualization of the aspirin MG reveals that the molecule has 13 nodes and 13 edges, thus $V = 13$ and $E = 13$. In property prediction tasks, hydrogen atoms are typically omitted to reduce complexity. However, in molecular dynamic tasks, hydrogen atoms are generally retained for accurate calculation. Currently, most chemical toolkits support the conversion from SMILES to MGs.

**Condense Graph of Reaction (CGR).** (Varnek et al., 2005) is a graph specifically designed for chemical reactions, as shown in Tab. 10. CGR can be seen as an analogous approach of DRFP in graph domain. It models the reaction as the superposition of molecules in the reaction. Each node represents the mixed features of the same atom in the reactants and products, while each edge represents the mixed features of the same chemical bond in the reactants and products. The mixing of features is typically done by subtraction or concatenation. If a certain atom or edge is absent in either the reactants or products, the feature for the absent side is set to zero.

Compared with MG, CGRs have a smaller size and are computationally efficient, but they may sacrifice the relative independence of the molecules. Algorithms for building CGR are provided in toolkits like CGRTools and Chemprop[7].

**Rxn Hypergraph.** (Tavakoli et al., 2022) is also specifically designed for chemical reactions. As in Tab. 10, all atoms in the molecule are connected to a Mol Hypernode that represents the molecule, while all Mol Hypernodes in the reactants or products are connected to a Rxn Hypernode that represents the entirety of the reactants or products. Furthermore, the Rxn Hypernodes in the reactants are interconnected, and the Rxn Hypernodes in the products are also interconnected, with no connections between the reactants and products.

The Rxn Hypergraph is designed to adapt the RGAT architecture. The hierarchical structure allows model to focus on features with different granularities within a reaction. However, the reactants remain isolated from the products, and the atomic mapping is absent. As a result, it still has limitations in expressing the molecular transformation in reaction.

We couldn't find any existing toolkits that can generate Rxn Hypergraphs. The Rxn Hypergraph used in our experiments is implemented based on the formulation in Tavakoli et al. (2022).

---

[6]https://github.com/reymond-group/drfp
[7]https://github.com/chemprop/chemprop

## C.3. Three-Dimensional

3D molecules or chemical reactions are typically represented using point cloud or 3D molecular graph (Thomas et al., 2018; Fuchs et al., 2020), as shown in Tab. 9 and 10. In point cloud, each node corresponds to an atom and contains both chemical properties and structural information(e.g. coordinate). Molecular point clouds are usually processed using Geometric GNNs or Transformers. They use a preset cutoff distance (Schütt et al., 2017a) for graph convolution or attention.

Since molecules can undergo transformations such as translation and rotation in space, there is a many-to-one relationship between point clouds and molecules. This introduces redundant information for chemical property prediction. Therefore, transformation invariance and equivariance are key focus in 3D molecular modeling. Molecular point clouds are generally used in molecular dynamics or quantum chemical property calculations (Jackson et al., 2021; Batzner et al., 2022). However, they are relatively rare in predicting yields, reaction conditions, and reaction types.

## C.4. Comparison of Graph Representations of Reaction

*Table 11.* Ability comparison of graph representations of reaction. **Itn** represents internal message passing within a molecule, **R to R** represents message passing between reactants, **P to P** represents message passing between products, **R to P** represents message passing between reactants and products, **Mol** and **Rxn** indicate the ability to express an independent molecule and an entire chemical reaction during the message passing stage, and **2D** and **3D** represent the inclusion of 2D and 3D topological structures.

| Representation | Message Passing | | | | Expression | | | |
|---|---|---|---|---|---|---|---|---|
| | Itn | R to R | P to P | R to P | Mol | Rxn | 2D | 3D |
| MG | ✓ | | | | ✓ | | ✓ | |
| Rxn Hypergraph | ✓ | ✓ | ✓ | | ✓ | | ✓ | |
| CGR | ✓ | ✓ | ✓ | ✓ | | ✓ | ✓ | |
| RG (ours) | ✓ | ✓ | ✓ | ✓ | ✓ | ✓ | ✓ | ✓ |

As shown in Tab. 11, we compare the message passing and expressive capabilities of various reaction representations. All representations allow for message passing between atoms within a molecule. Among them, MG lacks edges connecting different molecules. Rxn Hypergraph enables message passing among atoms within the reactants and products; however, it still does not facilitate message passing between reactants and products. CGR models the reaction as a superposition of molecules, however, during message passing, the features of the same atom in reactants and products are mixed in one node, which results in a loss of relative independence between the molecules.

RG connects the same atom in reactants and products by reaction edges. This improvement achieves reaction-level message passing while preserving molecular-level feature extraction. Moreover, RG incorporates 3D information, which provides richer structural prior compared to 2D graphs.

# D. Implementation Details

## D.1. Graph Representations

### D.1.1. REACTION GRAPH

In the implementation, each node and edge of Reaction Graph (RG) contains more information. The attributes of node $i$ can be represented as vector $\boldsymbol{n}_i \in \mathbb{R}^{D_n}$ (distinguished from $\boldsymbol{v}_i$ in the Sec. 2), and the attributes of edge $ij$ can be represented as vector $\boldsymbol{e}_{ij} \in \mathbb{R}^{D_e}$, where $\boldsymbol{v}_i$ and $\boldsymbol{e}_{ij}$ are the concatenations of all attributes of the node and edge, respectively. Tab. 12 provides an example of all available attribute values (which will have slight difference between datasets). Tab. 13 provides detailed dimensions of node attributes and edge attributes for each dataset. Note that the edge attributes here do not include $l_{ij}$, which is stored in a separate matrix $\boldsymbol{L}$ according to our previous definition in Sec. 2.

**Torsion Angle Extension.** Reaction Graph can be easily extended to model torsion angle by adding torsion angular edges. Specifically, for each edge $BC$ in the molecular graph, the algorithm searches for each edge $AB$ and $CD$. If $A \neq C$, $D \neq B$, and $A \neq D$, a torsion angular edge $AD$ is added. In this way, each edge length in tetrahedron $ABCD$ is determined, thereby uniquely defining the tetrahedron. Once the tetrahedron $ABCD$ is defined, the dihedral angle between the planes

*Table 12.* Example of node and edge attributes in Reaction Graph. Each attribute is represented by a one-hot vector, and the final node attribute or edge attribute is the concatenation of all relevant attributes.

| | | |
|---|---|---|
| **Node** | **Atom Type** | C, O, N, F, P, S, Cl, Br, I |
| | **Charge Type** | -4, -3, -2, -1, 0, 1, 2, 3, 4, 5, 6, 7 |
| | **Degree Type** | 0, 1, 2, 3, 4, 5, 6, 7, 8 |
| | **Hybridization Type** | SP, SP2, SP3, SP3D, SP3D2, S |
| | **Num of Hydrogens** | 1, 2, 3, 4, 5, 0 |
| | **Valence Type** | 0, 1, 2, 3, 4, 5, 6, 7, 8, 9, 10, 11, 12, 13, 14 |
| | **Ring Size Type** | 3, 4, 5, 6, 7, 8 |
| **Edge** | **Edge Type** | Bond Edge, Reaction Edge, Angular Edge |
| | **Bond Type** | Single, Double, Triple, Aromatic |
| | **Chirality Type** | Clockwise, Counterclockwise |
| | **Stereochemistry Type** | BondCis, BondTrans |

*Table 13.* The dimensions of node and edge attribute vector for each dataset, where $D_n$ denotes node attribute vector dimensions, and $D_e$ denotes edge attribute vector dimensions.

| **Dataset** | $D_n$ | $D_e$ |
|---|---|---|
| USPTO-Condition | 110 | 13 |
| Pistachio-Condition | 117 | 13 |
| Buchwald-Hartwig | 43 | 10 |
| Suzuki-Miyaura | 49 | 10 |
| USPTO-Yield | 128 | 14 |
| USPTO-TPL | 135 | 14 |
| Pistachio-Type | 127 | 17 |

$ABC$ and $BCD$ is determined, which corresponds to the torsion angle of $BC$. **One-hot Length Embedding.** In our experiments, we also use one-hot length embeddings. We use frequency-based discretization, and divide the edge lengths into 16 bins. The specific bond length distribution and the division method are shown in Fig. 5.

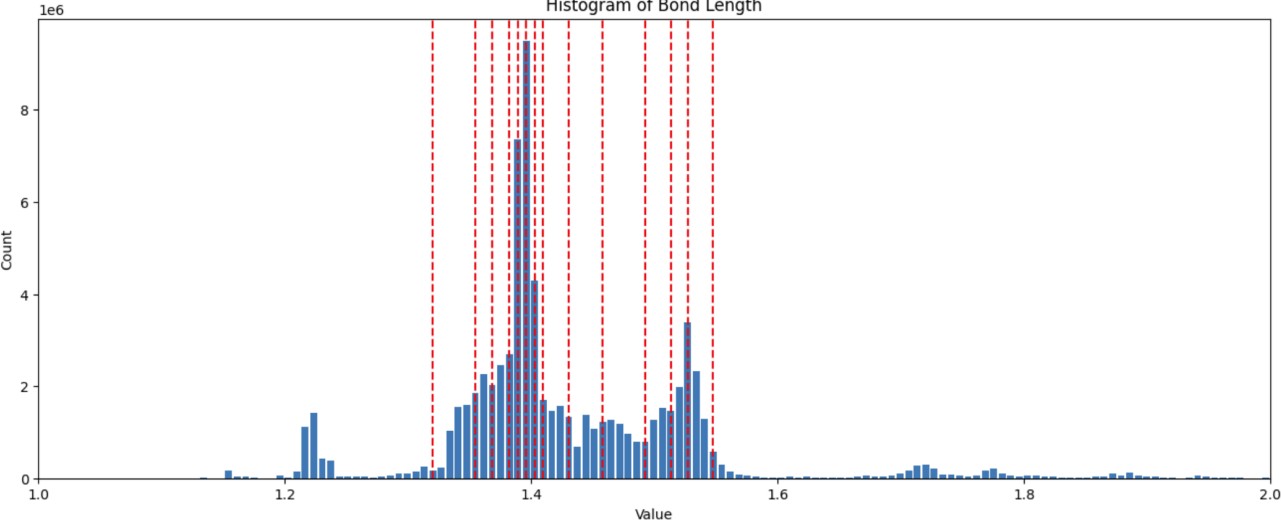

*Figure 5.* Histogram of the bond length distribution of molecular graphs on the USPTO-Condition dataset, along with the division method of the bins.

### D.1.2. RXN HYPERGRAPH

**Interaction Extension.** In Rxn Hypergraph, the interactions between reactants and products can be modeled by adding another hypernode. Specifically, the additional hypernode connects the origin rxn-hypernode of reactants and products.

**3D Extension.** In supplementary experiments, we add 3D bond length information to the Rxn Hypergraph. Different from molecular graph, Rxn Hypergraph contains hypernodes, which do not have actual coordinates. Therefore, the lengths of all edges connected to hypernodes are set to 0. This method ensures invariance. We use a 4-dimensional RBF kernel to embed the edges, and then concatenate the edge length embeddings to the original edge features.

### D.1.3. CGR

**3D Extension.** In supplementary experiments, we add 3D bond length information to the CGR in D-MPNN. Specifically, we implement a 3D Featurizer extension of Chemprop library, and calculate the conformation. The conformation is stored

as coordinates on the nodes. During inference, we calculate the bond lengths using the atomic coordinates, embedding them with a 16-dimensional RBF kernel, and then concatenate the bond length embeddings to the original edge features. Note that in the CGR, edge node corresponds to two atoms, and each edge corresponds to two chemical bonds. Therefore, each node contains two coordinates, and each edge contains two length embeddings.

## D.2. Model Details

### D.2.1. OURS MODEL

**Node Embedding.** We simply use a fully connected (FC) layer to embed the node vector $\boldsymbol{n}_i$, resulting in $\boldsymbol{v}_i \in \mathbb{R}^{D_v}$.

**Edge Embedding.** We first use the RBF kernel to embed the edge length $l_{ij}$ as $\boldsymbol{l}_{ij} \in \mathbb{R}^{D_l}$, and concatenate the edge length embedding to the edge attribute vector to obtain $\boldsymbol{e}'_{ij} = [\boldsymbol{e}_{ij}; \boldsymbol{l}_{ij}]$. Then, $\boldsymbol{e}'_{ij}$ is directly inputted into a FC layer and the output vector is reshaped into matrix $\mathcal{M}_{ij} \in \mathbb{R}^{D_v \times D_v}$. This approach is similar to the method of generating weight matrices in hypernetworks.

**Vertex-Edge Integration.** We refer to the method in Kwon et al. (2022b), using residual connections to enhance training stability and employing GRU to facilitate node attribute updates. Specifically, the node attributes $\boldsymbol{v}_i$ will serve as the memory state of the GRU, while the messages $\boldsymbol{v}_i^t + \sum_{j \in \mathbb{N}_i} \mathcal{M}_{ij} \cdot \boldsymbol{v}_j^t$ aggregated during each MPNN iteration will be treated as the input to the GRU, updating the node attributes instead of directly assignment. The final feature vector of node $i$ is calculated by $\boldsymbol{v}'_i = [\boldsymbol{v}_i^0; \boldsymbol{v}_i^{T_1}]$.

**Attention-based Aggregation.** We use $[\boldsymbol{q}^{t+1}; \boldsymbol{h}^t]$ as the input to the LSTM to obtain $\boldsymbol{h}^{t+1}$. The final reaction representation vector $\boldsymbol{r}$ is obtained by mapping $[\boldsymbol{q}^{T_2}; \boldsymbol{h}^{T_2-1}]$ through a FC layer with PReLU activation.

**Output.** This module is specifically designed for each downstream task. For the condition prediction task, the output module is consistent with CRM.

$$\boldsymbol{z}_1 = f_{c_1}(\boldsymbol{r}; \theta_{c_1}), \tag{7}$$

$$\boldsymbol{z}_i = f_{c_i}(\boldsymbol{r}, \boldsymbol{z}_1, \ldots, \boldsymbol{z}_{i-1}; \theta_{c_i}), \tag{8}$$

where $f_{c_i}$ linearly map each previously predicted condition $\boldsymbol{z}_j$ to embedding vector $\boldsymbol{h}_{z_j} \in \mathbb{R}^{D_z}$, concatenate all $\boldsymbol{z}_j$ with reaction feature $\boldsymbol{r}$ and feed them into a 2-layer FC classifier with ReLU activation to predict the next probability vector $\boldsymbol{z}_i \in \mathbb{R}^{D_{c_i}}$ of the $i$-th condition.

The yield prediction module uses a 3-layer MLP. It incorporates PReLU activation and Dropout layers, and simultaneously outputs the predicted mean $\mu_y$ and logarithmic variance $\log \sigma_y^2$ of yield.

$$\mu_y, \log \sigma_y^2 = f_r(\boldsymbol{h}; \theta_r). \tag{9}$$

The reaction classification head also uses a 3-layer MLP with PReLU activation to output the predicted probability vector of reaction type.

For LvG classification task, we add an FC layer to the last layer of the feature extraction module. It maps the node features into node-level multi-class classification labels.

**Hyperparameter Settings.** For USPTO-Condition, USPTO-TPL, and Pistachio-Type, we set the edge length embedding dimension $D_l$ to 16, with a total edge attribute dimension of $D_l + D_e = 29$. For Pistachio-Condition, $D_l = 1$, with $D_l + D_e = 14$. For the yield prediction task, since 3D information is not used, $D_l = 0$.

For condition prediction and reaction classification, we set the embedding dimension $D_v$ of nodes to 200, the number of MPNN iterations $T_1$ to 3, the number of Set2Set aggregation iterations $T_2$ to 2, the dimension of the GRU memory state to be the same as $D_v$, while the LSTM memory state dimension is set to $2 \times D_v$. The final output reaction feature vector $\boldsymbol{r}$ has a dimension $D_r$ of 4096. For yield prediction task, $D_v$ is set to 64, $D_r$ is set to 1024, and $T_1 = T_2 = 3$.

The hidden layer dimension of the reaction condition output head $f_c$ is 512, and the dimension of the condition embedding $D_z$ is 256. The hidden layer dimension of the reaction yield output head $f_r$ is 512, with a dropout rate of 0.1. The hidden layer dimension of the reaction classification output head is 4096.

The specific hyperparameter tuning methods can be found in Sec. H.

### D.2.2. OTHER MODELS

**Bond Angle Model.** We use the approach of DimeNet (Gasteiger et al., 2020) that integrates bond angles, explicitly providing angular information. In the implementation, we refer to the official DimeNet code[8] , and first construct an edge graph. In the edge graph, nodes represent edges in the molecular graph (MG), and edges represent angles. During each iteration, we first perform message passing in the edge graph to obtain its node features (which is the edge features in MG), and then conduct message passing in MG. Unlike the official implementation targeted at quantum chemistry calculation, we use the method more suitable for property prediction. Following GEM (Fang et al., 2022), we employ 16-dimensional RBF kernel for length and angle embedding[9].

**CRM.** We use the open-source code[10] provided by CRM (Gao et al., 2018) to test the results on the Pistachio-Condition dataset. Since Pistachio-Condition is similar in scale to USPTO-Condition, we refer to the hyperparameter settings in the reproduction code of Parrot (Wang et al., 2023b).

**Parrot.** We use the open-source code[11] provided by Parrot (Wang et al., 2023b). Specifically, since Parrot employs a SMILES tokenizer, and the Pistachio-Condition dataset contains vocabulary that is different from the USPTO-Condition dataset, we couldn't use the officially provided pre-trained parameters. Therefore, we first extract the vocabulary from Pistachio-Condition, then apply the RCM pre-training proposed by Parrot, and finally perform supervised fine-tuning on the Pistachio-Condition dataset.

**AR-GCN & CIMG.** Both methods provide corresponding open-source code[12] [13], but the network structure design of AR-GCN (Maser et al., 2021) only accommodates a specified number of reactants and products, while CIMG (Zhang et al., 2022) does not provide the corresponding training code. Therefore, we only refer to the reproduction results in Parrot (Wang et al., 2023b) for reference and do not train on Pistachio-Condition.

**D-MPNN.** We use the D-MPNN (Heid & Green, 2021) and CGR construction code provided in the Chemprop library. We maintain the default settings for hyperparameters, training strategies, loss functions, and optimizers in all task training.

**Rxn Hypergraph.** The Rxn Hypergraph (Tavakoli et al., 2022) does not provide related open-source code, but the structure of the Rxn Hypergraph is described in detail in the paper. Therefore, we reproduce the construction of the Rxn Hypergraph using the DGL library[14]. Additionally, the Rxn Hypergraph paper conducts experiments using RGAT, but does not specify the exact type of RGAT used. Thus, based on the descriptions in the paper, we employ EGAT provided in DGL, which is a commonly used RGAT.

**UGNN.** We use the open-source code[15] provided by UGNN (Kwon et al., 2022b) to train on the USPTO-Yield dataset. Since the molecules in the USPTO-Yield dataset are different from those in the HTE dataset, we adapt the dataset preprocessing script in the open-source code to handle the USPTO-Yield dataset and adjust the input dimension of the fully connected layer in the input layer accordingly to accommodate the reaction data from USPTO-Yield, while keeping other hyperparameters unchanged.

**DRFP.** We use the DRFP library[16] to generate DRFP fingerprints. According to the description in the original paper, we select 2048-dimensional DRFP and train an MLP with a hidden layer dimension of 1664 and a tanh activation function.

**RXNFP.** We use the RXNFP[17] library to test the performance of Pistachio-Type. Similar to Parrot, we are unable to train using the original token settings of RXNFP. Therefore, we construct a vocabulary for the Pistachio-Type dataset and perform MLM pre-training on RXNFP, followed by fine-tuning for the reaction classification task. The hyperparameter settings for the model architecture remain at their default values.

---

[8]https://github.com/gasteigerjo/dimenet

[9]https://github.com/PaddlePaddle/PaddleHelix/tree/dev/apps/pretrained_compound/ChemRL/GEM

[10]https://github.com/Coughy1991/Reaction_condition_recommendation

[11]https://github.com/wangxr0526/Parrot

[12]https://github.com/slryou41/reaction-gcnn

[13]https://github.com/zbc0315/synprepy

[14]https://www.dgl.ai/

[15]https://github.com/seokhokang/reaction_yield_nn/

[16]https://github.com/reymond-group/drfp

[17]https://github.com/rxn4chemistry/rxnfp

**T5Chem.** We use the official open-source code[18] of T5Chem (Lu & Zhang, 2022) to train on the USPTO-yield, Pistachio-Type, and Suzuki-Miyaura datasets. The official pre-trained parameters provided by T5Chem are based on character tokenization. After checking, we find that T5Chem's vocabulary includes all characters from our datasets. Therefore, we use the pre-trained parameters provided by T5Chem and train on the downstream tasks for each dataset. The hyperparameters remain at default settings.

**UniMol.** We use UniMol as a comparison method in our supplementary experiments. We utilize the official code[19] and released pre-trained parameters of UniMol. Specifically, to enable UniMol to handle reaction task, we use the Transformer-based feature extraction module from UniMol, and maintain consistency with our model in subsequent modules. The hyperparameter settings for the feature extraction module align with the default values provided in the UniMol official code, while the hyperparameters for other modules are consistent with our model.

**Vector Scalarization Model.** We utilize PaiNN to implement vector scalarization. Since PaiNN has not released an official implementation, we refer to a third-party implementation[20], and develop a DGL version. PaiNN does not support scalar edge attributes. To address this problem, we concatenate the scalar edge attributes with edge length embeddings during message computation. Additionally, since the lengths of reaction edges are set to 0, they may lead to division-by-zero error. Therefore, we uniformly increase all edge lengths by 1 before embedding, and only compute vector messages for edges other than the reaction edge. These operations does not affect the model's equivariance. We set the edge length embedding dimension to 20, with other hyperparameter consistent with our model.

**GAT & GIN.** We implement GAT and GIN by utilizing the EGAT and GINE modules from DGL, where EGAT employs 8-head attention, and the GINE module adds an additional BatchNorm1D layer after each iteration to maintain numerical stability. Other hyperparameters, such as iterations, hidden layer dimensions, and subsequent modules, are consistent with our model.

**ReaMVP.** In the appendix, we test the performance of ReaMVP without large-scale yield dataset pre-training. In this way, the yield data used for training is the same as our model and other methods (4k), rather than utilizing a much larger dataset (600k) for training. This better reflects the model's performance on the benchmark, allowing for a fair comparison. Specifically, we obtain the model code and pre-trained parameters from the official repository[21], and keep the default hyperparameter setting. But since the database used by ReaMVP has become proprietary data, we are unable to acquire the dataset and use their pre-training method.

**Set Transformer.** We attempt to use the Set Transformer as our aggregation module. Specifically, we use the Set Transformer Encoder and Decoder implemented by DGL toolkit for feature aggregation. Since the older version of DGL does not include Set Transformer, we directly used the source code from GitHub. We set the hidden dimension to 200, attention head number to 8, number of layer to 2, for both encoder and decoder.

### D.3. Visualization Details

**TMAP.** We implement TMAP dimensionality reduction by using TMAP[22] library in Python.

**UMAP.** We implement UMAP dimensionality reduction by using UMAP[23] library in Python.

**t-SNE.** We implement t-SNE dimensionality reduction by using Scikit-Learn[24] library in Python.

### D.4. Training Details

All the model training is completed on RTX 4090 using CUDA version 11.3, with PyTorch 1.12.1 and DGL 0.9.1.post1 to build the model and training framework. The machine has 512GB RAM and an Intel(R) Xeon(R) Gold 6326 CPU @ 2.90GHz. For the models using PyTorch Geometry, we use PyTorch Geometry 2.5.2[25] for training and reproduction. We

---

[18]https://github.com/HelloJocelynLu/t5chem
[19]https://github.com/deepmodeling/Uni-Mol
[20]https://github.com/nityasagarjena/PaiNN-model
[21]https://github.com/Meteor-han/ReaMVP
[22]https://tmap.gdb.tools/
[23]https://umap-learn.readthedocs.io/en/latest/
[24]https://github.com/scikit-learn/scikit-learn
[25]https://pytorch-geometric.readthedocs.io/en/latest/

use Scikit-Learn[26] and PyCM[27] to compute various evaluation metrics. For the replication of other works, we set up the corresponding environments using the official requirements files or guidelines provided.

**Leaving Group Identification.** We use the average of the cross-entropy loss for all atoms and the cross-entropy loss for leaving group atoms. This makes the model focus more on classifying the leaving group type. The training, testing, and validation sets are distinguished according to the USPTO-Condition in an 8:1:1 ratio. A learning rate of 5e-4 is used, along with a ReduceLROnPlateau training schedule with mode set to $min$, a factor of 0.1, patience of 5, and a minimum learning rate of 1e-8. Training is conducted with a learning rate of 5e-4, a weight decay of 1e-10, and the Adam optimizer with beta values of [0.9, 0.999]. We use a batch size of 32, and set the accumulation steps to 4 (equivalent to a batch size of 128). The training lasts for 50 epochs with early stopping applied. We choose 666 as the random seed and take the best evaluation epoch as the result. The total training time takes approximately 6 hours.

**Reaction Condition Prediction.** We use an improved cross entropy loss to optimize the condition prediction model:

$$\mathcal{L}_{\theta,\mathcal{G}} = -\sum_{i=1}^{N_1}\sum_{j=1}^{N_{c_i}} w_{ij} \cdot c_{ij}^{'} \log \frac{exp(z_{ij})}{\sum_{k=1}^{N_{c_i}} exp(z_{ik})}, \tag{10}$$

$$c_{ij}^{'} = \lambda_i c_{ij} + (1-\lambda_i)/D_{c_i}, \tag{11}$$

where $w_{ij}$ is the weight of class $j$ of $i$-th type of reaction condition, $\lambda_i$ is the label smoothing factor (Szegedy et al., 2016), $z_i$ is the predicted result vector for the $i$-th reaction condition output by the model, while $c_i$ is the one-hot encoding of the ground truth for the reaction condition. We use a two-stage training strategy to train our reaction condition model, with a learning rate of 5e-4, weight decay of 1e-10, and the Adam optimizer with beta values of [0.9, 0.999] for each stage. We employ a ReduceLROnPlateau training schedule with mode set to $min$, a factor of 0.1, patience of 5, and a minimum learning rate of 1e-8. For Pistachio-Condition, we split the dataset into training, validation, and test sets in an 8:1:1 ratio. In the first stage, we train for 50 epochs, using $\lambda$ of 0.9, 0.8, 0.8, 0.7, and 0.7 for catalyst1, solvent1, solvent2, reagent1, and reagent2, respectively. For the $None$ category of solvent2 and reagent2, we set the learning rate weight $w$ to 0.1. And for USPTO-Condition dataset, we set all $\lambda$ and $w$ to 1. After completing the first stage of training, we reset and initialize the weights of the output module using a normal distribution with a mean of 0 and a standard deviation of 0.1, and freeze the feature extraction and aggregation modules for the second stage of training, which lasted for 50 epochs. We use a batch size of 32 with 4 accumulation steps (equivalent to a batch size of 128). We choose 666 as the random seed and take the best evaluation epoch as the result, with a total training duration of approximately 2 days. The training times of other models and representations on USPTO-Condition are roughly the same as the ratio of the training times we presented in Fig. 7.

**Reaction Yield Prediction.** We use the training method of BNNs (Gal & Ghahramani, 2016; Kendall & Gal, 2017) combined with L2 regularization to train the yield model:

$$\mathcal{L}_{\theta,\mathcal{G}} = (1-\gamma)*(y-\mu_y)^2 + \gamma*(\frac{(y-\mu_y)^2}{\sigma_y^2} + \log\sigma_y^2) + \lambda\sum_{M}^{M\in\theta}\|M\|_2^2, \tag{12}$$

where $y \in [0,1]$ is the ground truth yield corresponding to $\mathcal{G}$ in the dataset, $\gamma$ and $\lambda$ are two regulatory factor, $\mu_y$ and $\sigma_y$ are the model output mean and variance, and $\theta$ is the set of model parameters. The first term makes $\mu_y$ close to the ground truth yield, and the second loss is used to train $\sigma_y$ to reflect prediction uncertainty. The last term is L2 regularization.

For the HTE dataset, our training framework remains consistent with UGNN, using a learning rate of 1e-3, weight decay of 1e-5, and the Adam optimizer with beta values of [0.9, 0.999]. For the hyperparameters of the model, we set $\gamma$ to 1e-1 and $\lambda$ to 1e-5. We train for a total of 500 epochs, shuffle the dataset at the beginning of each epoch, and employ the MultiStepLR training strategy with milestones set to [400, 450] and gamma set to 0.1. The training batch size is 32, with an accumulation step of 4 (equivalent to a batch size of 128). Training on each dataset takes approximately 20 minutes.

For the USPTO-Yield dataset, we train for 30 epochs (starts to overfit around the 20th epoch each time), modifying the MultiStepLR milestones to [20, 25]. We set $\gamma$ to 1e-1 and $\lambda$ to 0. We randomly split off 10% of the training set as a

---

[26]https://github.com/scikit-learn/scikit-learn
[27]https://github.com/sepandhaghighi/pycm

validation set and again choose 10 random seeds to take the best results. Training for Gram takes about 45 minute, while Subgram takes approximately 1.5 hours.

**Reaction Classification.** For the USPTO-TPL and Pistachio-Type datasets, we use the same training framework, employing cross entropy loss function and the Adam optimizer with learning rate of 5e-5, weight decay of 1e-10, and beta values of [0.9, 0.999]. We utilize ReduceLROnPlateau training scheduler with mode set to $min$, a factor of 0.1, patience of 5, and a minimum learning rate of 1e-8. We use a batch size of 32 with 4 accumulation steps (equivalent to a batch size of 128) and train for 100 epochs. We randomly split off 10% of the training set as a validation set and find that the dataset quality is relatively high, allowing us to achieve good performance without careful hyperparameter tuning and early stopping. Therefore, we take the final epoch as the result. We choose 123 as the random seed. The training for Pistachio-Type takes 60 hours, while USPTO-TPL takes 40 hours.

# E. Algorithm and Pipeline Details

## E.1. Reaction Graph Construction

**SMILES Formatting.** First, given a SMILES expression, the reaction expression takes the form: *A*>*B*>*C D*, where *A* and *C* represent the reactants and products, respectively. Part *B* typically represents the reagents. In the task of reaction condition prediction, *B* is one of the prediction targets, so we convert part *B* into our reagent classification labels. While in the tasks of predicting reaction types and yields, such reagents will be treated as additional prior. Therefore, in these tasks, we concatenate part *B* with part *A* using a (.) symbol, treating it as *A*. *D* contains additional information about the reaction, which is not require during the construction of RG. At this point, the SMILES of the chemical reaction simplifies to *A*>>*C*.

**Reaction Validation.** Second, we use the RDKit to parse reactants *A* and products *C*. If parsing succeed, *A* and *C* are considered valid. Next, we predict the atomic mapping of the reaction by RXNMapper and validate reaction formula by checking if the atoms with mapping in the reactants and products correspond one-to-one. Although this cannot guarantee that all reactions passing the tests are valid, it significantly ensures the quality of our reaction data.

**Molecular Graph Construction.** For each molecule in the reactants and products, we compute the attributes of each atom and bond by RDKit, as the example in Tab. 12, and use DGL to construct graph-format data. Each atom corresponds to a node, while each bond corresponds to an edge in the graph. Note that this is slightly differs from the RG defined earlier in Sec. 2. This simplification of RG is made for the sake of ease of writing and understanding, and the actual nodes and edges in RG contain more information. In addition to the aforementioned attributes of the bond, the edge also has a flag to indicate the type of edge (as there will also be reaction edges and angular edges). After generation, MGs of all molecules will be integrated into a single graph for easier manipulation.

**Atomic Coordinate Calculation.** We first attempt to initialize the conformation using ETKDG, and then optimize the conformation using MMFF94; if it fails, we switch to UFF (Casewit et al., 1992). If it still fails, we use the 2D coordinates of MG as a substitute. This ensures that most molecules contain 3D information, while the remaining has 2D coordinates as an approximation. We use RDKit and OpenBabel in this process to calculate conformations, aiming to cover as many molecules as possible.

**Angular Edge Construction.** We add angular edges to the graph to construct triangles to convey bond angle informations. Specifically, for each MG, we will find all edge pairs of the form $a, b$ and $b, c$ by traversing the adjacency list, and check if there is already an edge connecting $a, c$. If not, we will construct an angular edge connecting $a, c$ to implicitly convey the angle information of the angle $\angle abc$.

**Reaction Edges Construction.** Finally, we will construct the reaction edges. For all atoms with indices $m$ in the reactants and $n$ in products with the same mapping label, we will construct a reaction edge to connect them.

**Torsion Angular Edge Construction.** Torsion angular edge is an optional extension of Reaction Graph. In the molecular graph, we traverse all triplets $(ab, bc, cd)$ formed by the edges $ab$, $bc$, and $cd$, where $a$, $b$, $c$, and $d$ are distinct nodes. We then check if there is already a bond edge or angular edge $ad$ present. If not, we add a torsional angular edge $ad$ to implicitly convey the torsional angle of bond $bc$. Additionally, leveraging the advantages of the reaction graph, we can also conduct targeted designs, such as adding the torsional angular edge $ad$ only for edges $bc$ that are single bonds, to simplify calculations. This will be explored in future work.

---

**Algorithm 1** Train Reaction Condition Prediction Model

---

1: **Input:** Reaction Graph $\mathcal{G}$, ground truth labels $\boldsymbol{c}_{gt} = [\boldsymbol{c}_1, \boldsymbol{c}_2, \ldots, \boldsymbol{c}_5]$, label smoothing coefficients $\lambda_1, \ldots, \lambda_5$, learning rate $\alpha$, model parameters $\theta$, feature extraction and aggregation module $f$, output modules $g_1, g_2, \ldots, g_5$
2: **Output:** trained model parameters $\theta'$
3: **Constant:** dimension of reaction condition labels $D_1, \ldots, D_5$
4:
5: **procedure** TrainConditionModelOneIter($\mathcal{G}, \boldsymbol{c}_{gt}, \alpha, \theta, \lambda_1, \ldots, \lambda_5$)
6:     **let** $\boldsymbol{r} \leftarrow f(\mathcal{G}; \theta)$
7:     **let** $\hat{\boldsymbol{c}} = [\hat{\boldsymbol{c}}_1, \hat{\boldsymbol{c}}_2, \ldots, \hat{\boldsymbol{c}}_5]$ be zero vector like $\boldsymbol{c}$
8:     **let** $loss \leftarrow 0$
9:     **for** $i$ **from** $1$ **to** $5$ **do**
10:         **let** $\boldsymbol{s}_i \leftarrow \lambda_i \boldsymbol{c}_i + (1 - \lambda_i)/D_i$
11:         **let** $\hat{\boldsymbol{c}}_i \leftarrow g_i(\boldsymbol{r}, \boldsymbol{c}_1, \ldots, \boldsymbol{c}_{i-1}; \theta)$
12:         $loss \leftarrow loss + \text{CrossEntropy}(\boldsymbol{s}_i, \hat{\boldsymbol{c}}_i)$
13:     **end for**
14:     **let** $\theta' \leftarrow \theta - \alpha \cdot \nabla_\theta loss$
15:     **return** $\theta'$
16: **end procedure**

---

## E.2. Reaction Condition Prediction

Our model autoregressively predict five types of reaction conditions, which are catalyst, solvent1, solvent2, reagent1 and reagent2. As shown in Alg. 1, during training, the model predicts next condition based on the reaction representation vector and the ground truth labels of the previous conditions. While during inference, the input is the reaction representation vector and all the previous predicted conditions.

We also utilize beam search during inference. As shown in Alg. 2, before inference, we define how many candidate labels to generate for each reaction condition $(n_1, n_2, n_3, n_4, n_5)$. When inferring the $i$-th reaction condition, we input the reaction representation vector and all combinations of the previously predicted reaction conditions into the current classification head. The classification head produces $n_1 \times n_2 \times \cdots \times n_{i-1}$ probability vector. For each probability vector, we then select the top-$n_i$ as the output labels, forming $n1 \times n_2 \times \cdots \times n_{i-1} \times n_i$ combinations of reaction conditions. Then we continue to predict the next condition.

## E.3. Attention Weights Sample Selection

As mentioned in Sec. 3, we use the attention map of representative molecules to observe the model's understanding of reaction mechanism. Specifically, for these molecules, they have different reaction center atoms in different reactions. We identify reaction center by atomic mapping, and extract representative molecules through an automated script.

## E.4. Leaving Group Extraction

We use a dataset labeled with Leaving Groups (LvGs) in the LvG identification task. To get the LvG label, for each RG in the dataset, we first remove all reaction edges and their connected nodes. This step eliminate all non-LvG atoms, leaving the remaining connected subgraphs as the LvGs. We assign temporary labels to these connected subgraphs, and mark the temporary label on the original RG nodes. Next, we perform graph hashing to each LvG by a randomization parameter GNN. We count all the hash values and filter in LvGs with frequencies above threshold. This results in 204 LvG labels, while other LvGs are uniformly assigned a $Other$ label. Finally, we build a mapping between temporary labels and LvG labels, and update the labels in RG. For atoms that do not belong to LvGs, they are assigned $None$ label.

## E.5. Bin-Packing

When testing the model's computational efficiency, we first construct a series of reaction bins. Then we test model's average runtime on these bins as a measure of efficiency. Each bin contains the same number of (nodes) atoms. To construct these bins, we use a greedy algorithm as shown in Alg. 3.

---

**Algorithm 2** Reaction Condition Beam Search Inference

---

1: **Input:** Reaction Graph $\mathcal{G}$, candidate labels $(n_1, n_2, n_3, n_4, n_5)$, model parameters $\theta$, feature extraction and aggregation module $f$, output modules $g_1, g_2, \ldots, g_5$
2: **Output:** Predicted reaction condition combinations $\boldsymbol{C} \in \mathbb{R}^{(n_1 n_2 \ldots n_{i-1}) \times 5}$
3:
4: **procedure** ConditionBeamSearch($\mathcal{G}, \boldsymbol{r}, (n_1, n_2, n_3, n_4, n_5), \theta$)
5:     **let** $\boldsymbol{r} \leftarrow f(\mathcal{G}; \theta)$
6:     **let** $\boldsymbol{c}_1 \leftarrow g_i(\boldsymbol{r}; \theta)$
7:     **let** $c_{11}, \ldots, c_{1n_1}$ be top $n_1$ of $\boldsymbol{c}_1$
8:     **let** $\mathcal{C}$ be $[[c_{11}], \ldots, [c_{1n_1}]]$
9:     **for** $i$ **from** 2 **to** 5 **do**
10:         **let** $\mathcal{C}_{new}$ be a empty list
11:         **for** $j$ **from** 0 **to** $\mathcal{C}.length - 1$ **do**
12:             **let** $\boldsymbol{c}_i \leftarrow g_i(\boldsymbol{r}, \mathcal{C}[j]; \theta)$
13:             **let** $c_{i1}, \ldots, c_{in_i}$ be top $n_i$ of $\boldsymbol{c}_i$
14:             **for** $k$ **from** 1 **to** $n_i$ **do**
15:                 **let** $\boldsymbol{c}_{new} \leftarrow Concatenate([\mathcal{C}[j], [c_{ik}]])$
16:                 **append** $\boldsymbol{c}_{new}$ **to** $\mathcal{C}_{new}$
17:             **end for**
18:         **end for**
19:         $\mathcal{C} \leftarrow \mathcal{C}_{new}$
20:     **end for**
21:     **let** $\boldsymbol{C} \leftarrow$ Stack($\mathcal{C}$)
22:     **return** $\boldsymbol{C}$
23: **end procedure**

---

**Algorithm 3** Reaction Bin-Packing

---

1: **Input:** Number of bins $N$ and capacity of each bin $C$, number of atoms of reactions $\mathcal{A} = [a_1, a_2, \ldots, a_M]$
2: **Output:** Indices of reactions in each bin $\mathcal{B} = [\mathcal{L}_1, \mathcal{L}_2, \ldots, \mathcal{L}_N]$
3:
4: **procedure** ReactionBinPacking($N, C, \mathcal{A}$)
5:     **shuffle** $\mathcal{A}$
6:     Initialize $\mathcal{B}$ as a list of lists of length $N$
7:     Initialize $\mathcal{W}$ as a zero list of length $N$
8:     **for** each reaction $a_i$ **in** $\mathcal{A}$ **do**
9:         **let** $d_{min} \leftarrow 0$
10:         **let** $k \leftarrow 0$
11:         **for** $j \leftarrow 1$ **to** $N$ **do**
12:             $d \leftarrow C - (\mathcal{W}[j] + a_i)$
13:             **if** $d < d_{min}$ and $d > 0$ **do**
14:                 $d_{min} \leftarrow d$
15:                 $k \leftarrow j$
16:             **end if**
17:         **end for**
18:         **if** $k > 0$ **do**
19:             **push** $i$ into $\mathcal{B}[k]$
20:             $\mathcal{W}[k] \leftarrow \mathcal{W}[k] + a_i$
21:         **end if**
22:     **end for**
23:     **return** $\mathcal{B}$
24: **end procedure**

---

# F. Complexity Analysis

## F.1. Theoretical Analysis

Assume that the messages are calculated by passing the atom features through a FC layer. We first define:

$V$: Number of atoms; $M$: Number of molecules; $E$: Number of chemical bonds;

$D$: Maximum degree of atomic nodes; $L$: Length of the molecular SMILES string;

$H$: Hidden feature dimension of atoms; $A$: Number of atoms in a graph node;

$N$: Number of iterations for the GNN/Transformer; $P$: Number of message passing in a GNN iteration.

- **Molecular Grpah.** Since each edge needs to calculate the message once in each GNN interaction, and the complexity of fully connected layer is $O(H^2)$, each node in molecular graph represents one atom, the complexity of the original Molecular Graph is $O(ENH^2)$.

- **Reaction Graph (ours).** Reaction Graph introduces two types of edges: reaction edges and angular edges.

  - Reaction Edge: The number of reaction edges $E_{reaction}$ cannot exceed the number of product atoms $V_{product}$, and the number of product atoms cannot exceed half of the total atoms, so $E_{reaction} = V_{product} \leq V/2$. Therefore, the computational cost introduced by the reaction edges is $O(V/2 \cdot NH^2) = O(VNH^2)$.
  - Angular Edge: Since a bond angle is formed between every two neighboring edges of each atom node, and the number of bond angles is equal to the number of angle edges. Therefore, the number of angle edges $E_{angular} = VD^2$. Therefore, the computational cost introduced by the angular edges is $O(VD^2 \cdot NH^2)$.

  The overall complexity of Reaction Graph is $O[(E + V + VD^2) \cdot NH^2] = O[(E + VD^2) \cdot NH^2]$. When 3D information is not used, the complexity of the Reaction Graph is $O[(E + V) \cdot NH^2]$.

- **Bond Angle Model.** Our proposed angular edges are more efficient than explicitly using bond angles. With the angular edge method, there is only one message passing per round of GNN iteration. In contrast, when angle features are explicitly integrated, each GNN iteration requires $P$ message passing operations. The complexity is $O[(E + VD^2) \cdot PNH^2]$.

- **Rxn Hypergrpah.** Rxn Hypergraph adds edges connecting each atom to the corresponding molecule hypernode, introducing a computational cost of $O(VNH^2)$. The molecule nodes between reactants are connected pairwise, all linking to the reactant hypernode. And it is similar for the products. The cost for these new edges are $O[(M^2 + M) \cdot NH^2] = O(M^2 \cdot NH^2)$. Finally, the cost for Rxn Hypergrpah is $O[(E + V + M^2) \cdot NH^2]$

- **Condense Graph of Reaction.** In Condense Graph of Reaction (CGR), each node represents $A$ atoms from the original graph, and each edge represents $A$ edges from the original graph. Therefore, the number of edges in the CGR becomes $E/A$, and the dimension of the hidden layer for the nodes becomes $AH$. Thus, the time complexity of the CGR is $O[E/A \cdot N \cdot (AH)^2] = O(E \cdot NAH^2)$.

- **SMILES-based Methods.** SMILES-based methods typically use Transformers as the backbone. In Transformers, the time complexity for each component is:

  1. Tokenization: The cost of regex tokenizer is $O(L)$
  2. Embedding: The cost of embedding layer is $O(LH)$
  3. Attention: The size of the attention matrix is $L^2$, so the complexity for attention matrix calculation is $O(L^2H)$
  4. Feed Forward: Each token's embedding is processed through a fully connected layer, so the time complexity is $O(LH^2)$

  And there's $N$ layers of Attention and Feed Forward. The overall time complexity of Transformer is $O[L + LH + N \cdot (LH^2 + L^2H)] = O[N \cdot (LH^2 + L^2H)]$

*Table 14.* Data pre-processing time for different representations. The molecular graph can be viewed as a reaction graph without reaction information and 3D structure information. Time units are milliseconds (ms).

| Representation | Information | | Total Length of Reaction SMILES | | | | | | | | | | | | |
|---|---|---|---|---|---|---|---|---|---|---|---|---|---|---|---|---|
| | Reac. | 3D | 35 | 60 | 85 | 110 | 135 | 160 | 185 | 210 | 235 | 260 | 285 | 310 | 335 | 360 |
| Reaction Graph | ✓ | ✓ | 3.2 | 5.6 | 10.1 | 18.1 | 28.7 | 24.9 | 52.5 | 54.0 | 72.4 | 82.8 | 100.3 | 244.4 | 585.6 | 923.3 |
| w/o Reaction Edge | ✗ | ✓ | 3.6 | 4.4 | 7.2 | 16.8 | 26.0 | 28.8 | 45.9 | 57.5 | 69.3 | 81.3 | 91.9 | 157.8 | 722.1 | 945.9 |
| w/o Angular Edge | ✓ | ✗ | 1.2 | 1.5 | 2.3 | 2.7 | 3.5 | 3.9 | 4.8 | 5.4 | 6.1 | 6.8 | 5.6 | 5.5 | 21.7 | 29.3 |
| Molecular Graph (w/o 3D) | ✗ | ✗ | 0.7 | 1.0 | 1.3 | 1.5 | 2.4 | 2.2 | 3.0 | 3.4 | 4.7 | 4.6 | 4.9 | 4.1 | 15.9 | 19.9 |
| Molecular Graph (w/ 3D) | ✗ | ✓ | 2.9 | 4.1 | 7.3 | 16.5 | 23.5 | 26.8 | 49.5 | 54.7 | 69.7 | 82.4 | 90.5 | 156.4 | 718.7 | 932.1 |

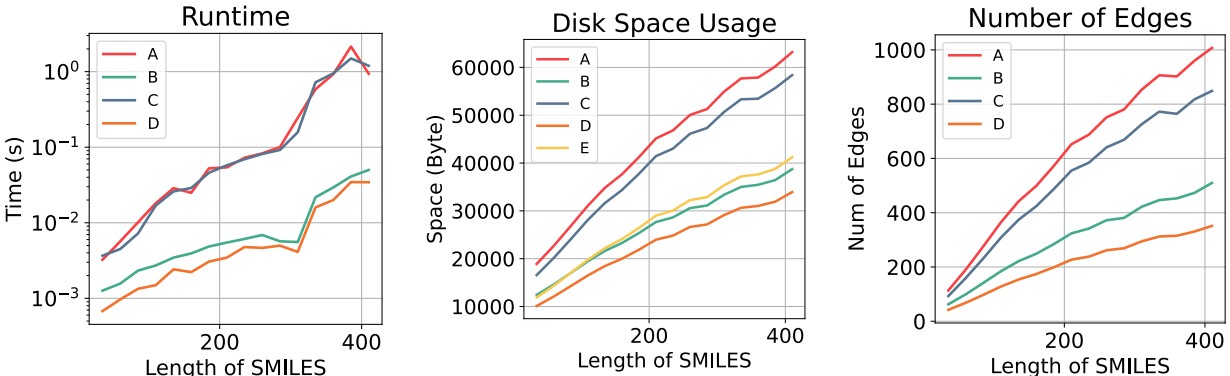

*Figure 6.* Graph construction algorithm metrics of different graph representations, including (A) RG, (B) RG with only reaction edges, (C) RG with only 3D information and angular edges , (D) molecular graph (MG), and (E) MG with 3D information. Due to the overlap between groups (C) and (E) of computation time, and since the number of edges in group (D) is equal to that in group (E), group (E) is omitted in both figures.

## F.2. Experimental Analysis

### F.2.1. CONSTRUCTION COST

We measured the preprocessing cost of the following graph representations, including processing time and disk space:

> (A) RG, (B) RG with only reaction edges, (C) RG with only 3D information and angular edges,
>
> (D) molecular graph (MG), (E) MG with 3D information

As shown in Fig. 6, the construction time for all graphs exhibits a linear increase with the SMILES length. The construction time for RG with angular edges is almost the same as that for 3D MG, so the curve is omitted. In RG, the overhead of computing 3D structures is largest, approximately $10^1 - 10^2$ times that of other parts. For hard disk space, the space occupied by angular edges is largest, while reaction edges and conformation storage is relatively small. This is because the number of angular edges is more than other types of edges, and is about 2-3 times the number of edges in the original MG.

Tab. 14 shows the influence of reaction edges, angular edges, and bond edge length on data pre-processing time in RG, as the total length of reaction SMILES increases. During data pre-processing, the primary time cost comes from 3D conformation calculation. But it is an unavoidable cost for any method that utilizes 3D structure information.

### F.2.2. COMPUTATIONAL COST

**Compared with Graph-based Methods.** We test the computational efficiency of graph-based representations and architectures. To ensure a fair comparison, we use the same GNN architecture for different graph representations. Note that in CGR, one node is the superposition of two atoms, and one edge in the superposition of two bonds. Results are shown in Fig. 7, 8 and Tab. 15.

Compared with other graph representations. Due to incorporating additional reaction information and 3D structures, our method requires slightly more time than molecular graphs and Rxn Hypergraph representations. However, it delivers

*Table 15.* Model inference time for different representations on the same backbone. Time units are milliseconds (ms).

| Representation | Information | | Number of atoms $|V|$ in Reaction Graph | | | | |
|---|---|---|---|---|---|---|---|
| | **Reaction Infomation** | **3D Structure** | **2500** | **3000** | **3500** | **4000** | **4500** |
| Reaction Graph | ✓ | ✓ | 37.28 | 43.77 | 50.57 | 58.03 | 65.45 |
| w/o Reaction Edge | ✗ | ✓ | 35.36 | 40.92 | 48.68 | 53.30 | 58.79 |
| w/o Angular Edge | ✓ | ✗ | 22.51 | 25.96 | 29.82 | 34.21 | 38.77 |
| Molecular Graph | ✗ | ✗ | 18.59 | 21.93 | 25.39 | 28.30 | 31.48 |
| Rxn Hypergraph | ✗ | ✗ | 31.18 | 37.28 | 43.31 | 48.53 | 54.06 |
| Condensed Graph of Reactions | ✓ | ✗ | 44.55 | 52.48 | 60.75 | 66.32 | 75.34 |

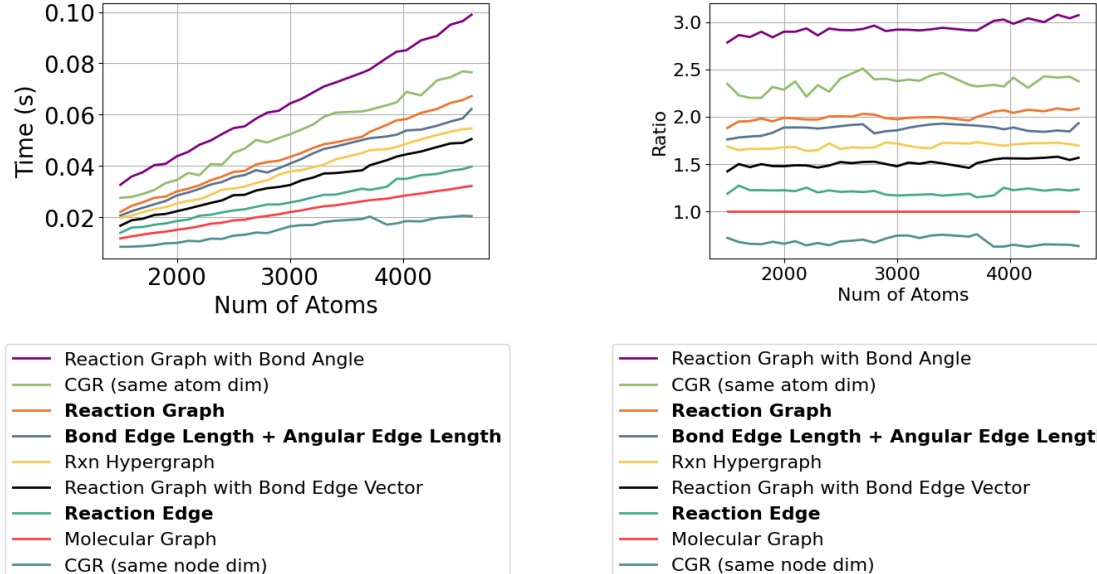

*Figure 7.* Computational time of models and graph representations at different atom counts. Legend order matches line order.

*Figure 8.* Computational time ratios of models and graph representations in relation to molecular graph at different atom counts.

*Table 16.* Inference time comparison for Reaction Graph and SMILES-based methods. According to the result, Reaction Graph requires smaller computation cost.

| Methods | Representation | Time |
|---|---|---|
| T5Chem | SMILES | 3min 19s |
| RXNFP | SMILES | 23min 43s |
| Reaction Graph (ours) | Reaction Graph | **2min 36s** |

significant performance improvement, justifying the additional time cost. The CGR introduces larger computation overhead because each of its nodes is a combination of two atoms, requiring larger hidden layer dimensions in the neural network.

Compared with other 3D informations, angular edge are more efficient than explicitly introducing bond angle, as concluded in Sec. 3. The computational efficiency slightly lags behind bond vector, which only introduces 50% computational overhead. However, bond vector may introduce redundant information (e.g. torsion angle), resulting in suboptimal performance. In contrast, angular edge significantly enhance performance while maintaining moderate computational overhead.

**Compared with Other Methods.** We compare the inference time of Reaction Graph and smiles-based methods. The results are shown in Tab. 16. Reaction Graph, as a graph-based model, exhibits the shortest inference time.

# G. Dataset Preprocessing and Analysis

For each dataset, we test various metrics related to dataset distribution and complexity, including the frequency of each label, the frequency of each atom, the total number of atoms and molecules, the total number of atoms per molecule and per reaction and so on. For clarity, we sort all data categories by frequency in ascending order except for the yield dataset, and use a logarithmic scale. Due to the large number of data points, we use line charts to display the data distribution. In each chart, the x-axis represents the data categories, while the y-axis represents the frequency of that category in the dataset.

### G.1. USPTO-Condition

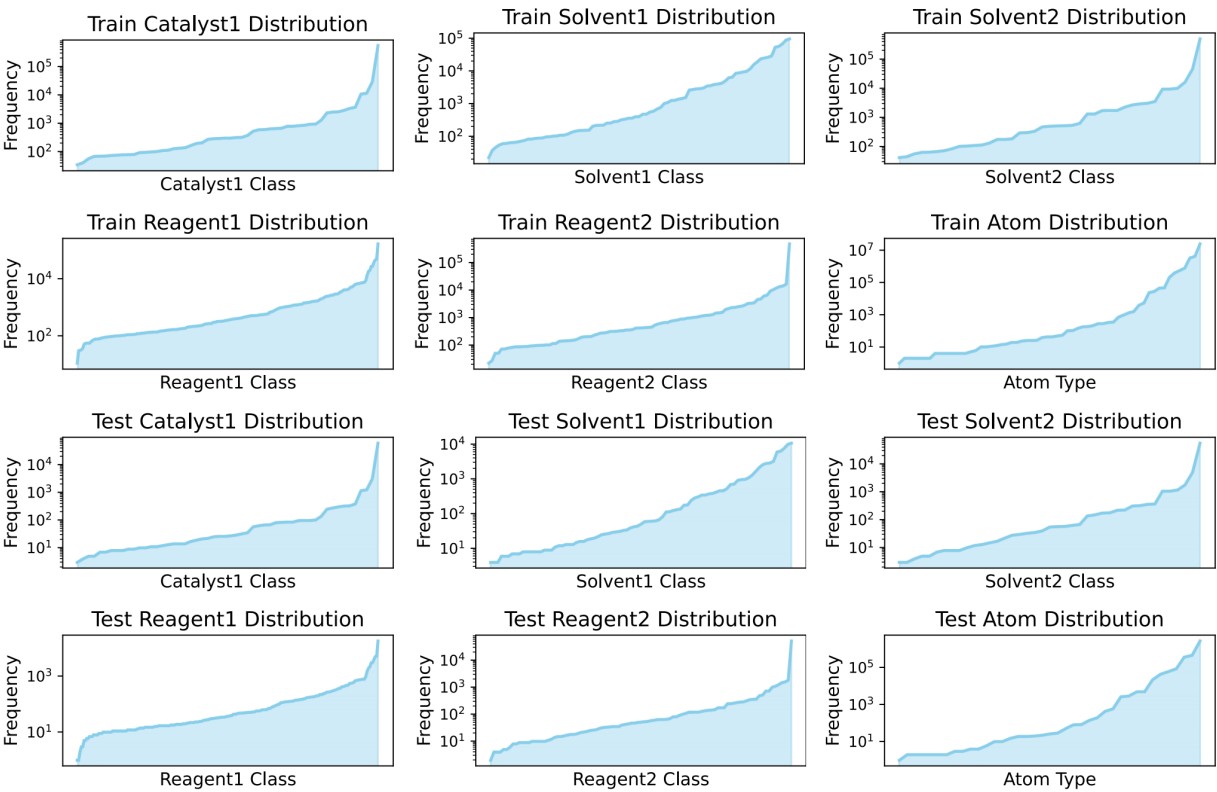

*Figure 9.* Data distributions of USPTO-Condition, including distribution of the number of data points for each category of reaction conditions, as well as the distribution of the occurrence of atom types.

The USPTO-Condition dataset is a collection of reaction condition data from USPTO database. We use scripts provided in Parrot (Wang et al., 2023b) to process the raw dataset. First, for each reaction data in USPTO, we extract all corresponding reaction condition annotations. Then, we use RXNMapper to predict atomic mapping. Molecules in reactants that do not contain any mapping markers are considered reagents. During this process, all invalid SMILES are discarded, and duplicated data rows are dropped. Afterward, we count the frequency of each catalyst, solvent, and reagent, and remove conditions with frequencies below 100. We check reagents with ions and removed non-electrically neutral combinations. Then, to standardize the output format, we delete all data with catalyst $> 1$, solvent $> 2$, reagent $> 2$, and those without reaction conditions. Finally, the dataset is split into training, validation, and test sets. The various metrics of the dataset are shown in Tab. 17, while the distributions of data are shown in Fig. 9.

*Table 17.* Summary of USPTO-Condition, including metrics for measuring the complexity of data points, the overall complexity of the dataset, and the distribution differences between the train/val set and the test set.

| Metric | Train/Val | Test |
|---|---|---|
| Size | 612,666 | 68,075 |
| Mean Length | 96.55 | 98.44 |
| Mean Atoms | 18.20 | 18.53 |
| Mean Molecules | 2.88 | 2.89 |
| Max Length | 868 | 825 |
| Max Atoms | 347 | 280 |
| Atoms | 60 | 39 |
| Mean Atoms per Molecule | 24.22 | 22.92 |
| Max Atoms per Molecule | 176 | 165 |
| Max Molecules | 55 | 24 |
| Molecules | 849,149 | 135,066 |
| Different Molecules (Train/Val and Test) | 787,898 | |
| Num Catalyst 1 | 54 | |
| Num Solvent 1 | 85 | |
| Num Solvent 2 | 41 | |
| Num Reagent 1 | 223 | |
| Num Reagent 2 | 95 | |

Based on the divided USPTO-Condition, we use the Reaction Graph (RG) construction algorithm in Sec. E to build a graph dataset. For molecules that could not generate conformations, we directly discarding them. The number of remaining data samples at each stage is in Tab. 18.

*Table 18.* Changes in the dataset size of USPTO-Condition during the preprocessing process.

| Stage | Num of Data Rows |
|---|---|
| Reaction Data Extraction | 3,130,812 |
| Mapping and duplication Dropping | 1,117,867 |
| Filter Conditions | 680,741 |
| Split Train/Val/Test | 544,591 : 68,075 : 68,075 |
| Graph Dataset Generation | 544,125 : 68,018 : 68,002 |

### G.2. Pistachio-Condition

For the Pistachio database, we use the same approach as the USPTO-Condition dataset. The difference is that most reactions in Pistachio contains mapping information, and there are also atmosphere labels. To address these problems, during the mapping step, we only predict the missing mapping; during the filtering step, we discard all reaction entries with atmosphere labels. Finally, we also split the dataset into 8:1:1 for training, testing, and validation. The various metrics of the dataset are shown in Tab. 19, while the distributions of data are shown in Fig. 10.

Similarly, based on the split dataset, we further generate the RG dataset. And the number of remaining data samples at each stage is as shown in Tab. 20.

The final dataset size is similar to that of USPTO-Condition. However, the reaction conditions in Pistachio-Condition are sparse, and the data samples are more complex. The reason the model achieves higher classification accuracy on Pistachio-Condition is also due to the uneven distribution of labels, where the $None$ class accounts for the majority, which can be observed in the confusion matrix from Sec. H.

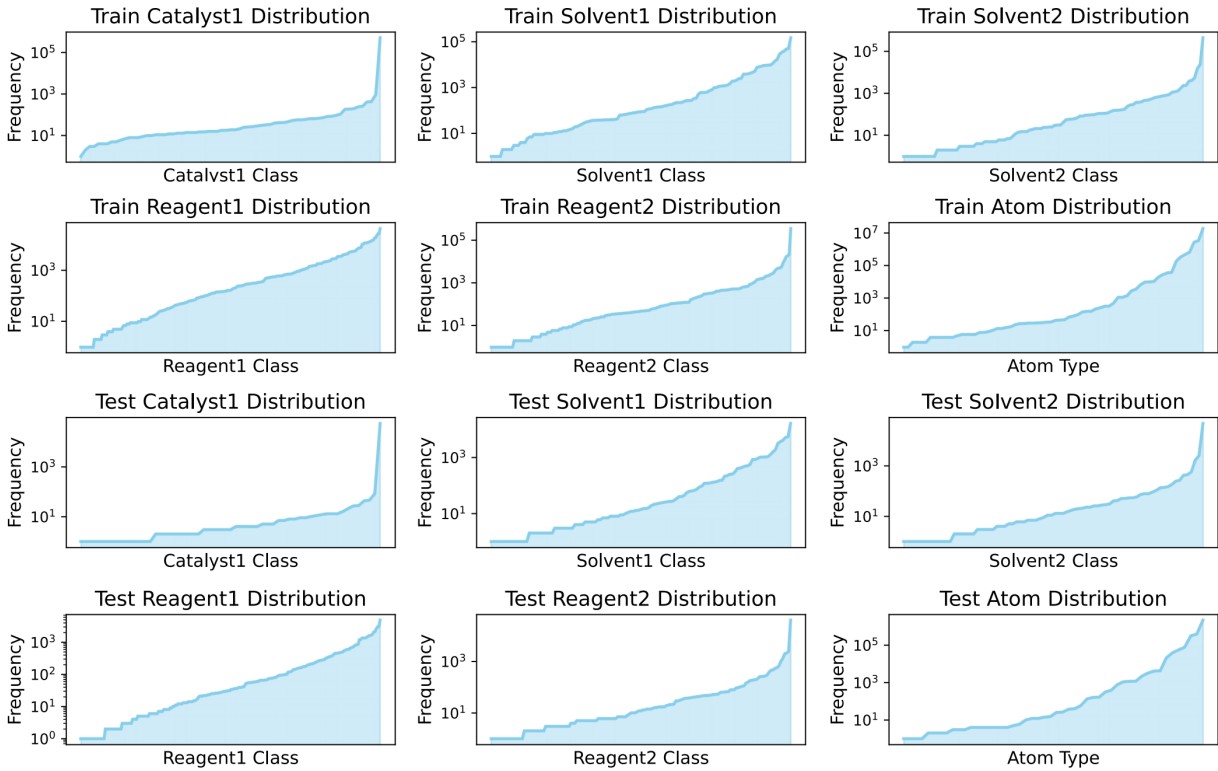

*Figure 10.* Data distributions of Pistachio-Condition, including distribution of the number of data points for each category of reaction conditions, as well as the distribution of atom types.

### G.3. HTE

We use the dataset provided by UGNN (Kwon et al., 2022b), which contains the SMILES expressions of reactions and yield data. Due to the small number of molecules included, it is difficult for models to learn from the limited conformational data. Therefore, we only generate RGs with reaction edges and did not discard any data. Consequently, the final dataset size and metrics are as shown in Tab. 21 and 22.

The number of data samples in the HTE dataset is nearly a thousand times smaller than that in our previous datasets. As all data points belongs to the same reaction, the variety of molecules is relatively limited, and the complexity of the data is relatively low. Previous works also propose using Test datasets to evaluate the model's generalization. Test datasets has greater distribution differences between the training and testing splits. Tab. 23 lists metrics for the Test datasets.

For Test datasets, we can clearly see significant differences between the training and testing sets, which also include different types of molecules. This requires the model to accurately model the relationship between molecules and reaction structures in order to obtain reasonable extrapolation results.

### G.4. USPTO-Yield

We use the USPTO-Yield dataset provided in Yield-Bert (Schwaller et al., 2021b), which is extracted from the USPTO database. It is relatively noisy with a complex distribution. The dataset is processed similarly as described above, with some specific metrics shown in Tab. 24 and 25, and the distribution of data shown in Fig. 11.

The yield distribution in USPTO-Yield is quite uneven, and there is a significant difference in the yield distributions between Gram and Subgram, as noted in (Schwaller et al., 2021b). Additionally, the quality of the yield data is low, and the relationship between yield and molecular structure is non-smooth. These factors increases the task difficulty.

*Table 19.* Summary of Pistachio-Condition dataset, including metrics for measuring the complexity of data points, the overall complexity of the dataset, and the distribution differences between the train/val set and the test set.

| Metric | Train/Val | Test |
|---|---|---|
| Size of Dataset | 506,224 | 56,247 |
| Mean Length per Reaction | 99.66 | 99.93 |
| Mean Atoms per Reaction | 17.44 | 17.48 |
| Atom Types | 68 | 50 |
| Mean Atoms per Molecule | 24.27 | 22.73 |
| Max Atoms of Molecule | 218 | 209 |
| Mean Molecules per Reaction | 3.04 | 3.04 |
| Max Length of Reaction | 988 | 878 |
| Max Atoms of Reaction | 440 | 408 |
| Max Molecules of Reaction | 54 | 18 |
| Molecule Types | 761,357 | 110,771 |
| Different Molecules (Train/Val and Test) | 718,036 | |
| Catalyst1 Types | 69 | |
| Solvent1 Types | 134 | |
| Solvent2 Types | 108 | |
| Reagent1 Types | 267 | |
| Reagent2 Types | 198 | |

*Table 20.* Changes in the dataset size of Pistachio-Condition during the preprocessing process.

| Stage | Num of Data Rows |
|---|---|
| Reaction Data Extraction | 145,035,928 |
| Filter Conditions | 1,981,125 |
| Mapping and duplication Dropping | 562,471 |
| Split Train/Val/Test | 449,977 : 56,247 : 56,247 |
| Graph Dataset Generation | 449,902 : 56,240 : 56,234 |

*Table 21.* Summary of Buchwald-Hartwig on metrics reflecting dataset complexity.

| Metric | Value |
|---|---|
| Size of Dataset | 3,955 |
| Mean Length pre Reaction | 216.65 |
| Mean Atoms pre Reaction | 116.15 |
| Max Length pre Reaction | 300 |
| Max Atoms of Reaction | 146 |
| Molecules pre Reaction | 7 |
| Molecule Types | 51 |
| Atom Types | 10 |
| Mean Atoms per Molecule | 13.20 |
| Max Atoms of Molecule | 46 |

*Table 22.* Summary of Suzuki-Miyaura on various metrics reflecting dataset complexity.

| Metric | Value |
|---|---|
| Size of Dataset | 5,760 |
| Mean Length pre Reaction | 196.925 |
| Mean Atoms pre Reaction | 110 |
| Max Length pre Reaction | 270 |
| Max Atoms of Reaction | 144 |
| Mean Molecules pre Reaction | 12.14 |
| Max Molecules pre Reaction | 15 |
| Molecule Types | 43 |
| Atom Types | 16 |
| Mean Atoms per Molecule | 12.14 |
| Max Atoms of Molecule | 42 |

*Table 23.* Summary of Test datasets of Buchwald-Hartwig, including metrics for measuring the complexity of data points, the overall complexity of the dataset, and the distribution differences between the train/val set and the test set.

| Metric | Test1 | | Test2 | | Test3 | | Test4 | |
|---|---|---|---|---|---|---|---|---|
| | train | test | train | test | train | test | train | test |
| Mean Length per Reaction | 213.23 | 224.62 | 214.61 | 221.41 | 214.53 | 221.60 | 214.55 | 221.54 |
| Mean Atoms per Reaction | 16.40 | 17.05 | 16.56 | 16.67 | 16.53 | 16.74 | 16.48 | 16.85 |
| Max Length of Reaction | 289 | 300 | 300 | 300 | 300 | 300 | 300 | 300 |
| Max Atoms of Reaction | 138 | 146 | 146 | 146 | 146 | 146 | 146 | 146 |
| Molecule Types | 45 | 38 | 46 | 38 | 46 | 38 | 46 | 37 |
| Mean Atoms per Molecule | 13.18 | 14.50 | 13.61 | 13.95 | 13.54 | 14.03 | 13.30 | 14.16 |
| Max Atoms per Molecule | 46 | 46 | 46 | 46 | 46 | 46 | 46 | 46 |
| Different Molecules (Train/Val and Test) | 13 | | 13 | | 13 | | 14 | |
| Molecule per Reaction | 7 | | | | | | | |
| Atom Types | 10 | | | | | | | |
| Size of Dataset | 3955 | | | | | | | |

*Table 24.* Summary of Gram dataset, including metrics for measuring the complexity of data points, the overall complexity of the dataset, and the distribution differences between the train/val set and the test set.

| Metric | Train/Val | Test |
|---|---|---|
| Size of Dataset | 156,565 | 39,137 |
| Mean Length per Reaction | 115.97 | 116.03 |
| Mean Atoms per Reaction | 60.44 | 60.46 |
| Mean Molecules per Reaction | 6.63 | 6.63 |
| Max Length of Reaction | 560 | 493 |
| Max Atoms of Reaction | 257 | 244 |
| Mean Atoms per Molecule | 20.27 | 19.33 |
| Max Atoms of Molecule | 116 | 103 |
| Atom Types | 67 | 57 |
| Max Molecules of Reaction | 59 | 32 |
| Molecule Types | 230,450 | 74,116 |
| Different Molecules (Train/Val and Test) | 197,580 | |

*Table 25.* Summary of Subgram dataset, including metrics for measuring the complexity of data points, the overall complexity of the dataset, and the distribution differences between the train/val set and the test set.

| Metric | Train/Val | Test |
|---|---|---|
| Size of Dataset | 240,326 | 60,075 |
| Mean Length per Reaction | 150.24 | 150.67 |
| Mean Atoms per Reaction | 79.13 | 79.35 |
| Mean Molecules per Reaction | 6.88 | 6.89 |
| Max Length of Reaction | 696 | 641 |
| Max Atoms of Reaction | 352 | 260 |
| Mean Atoms per Molecule | 25.88 | 24.91 |
| Max Atoms of Molecule | 166 | 143 |
| Atom Types | 67 | 57 |
| Max Molecules of Reaction | 38 | 29 |
| Molecule Types | 400,811 | 123,077 |
| Different Molecules (Train/Val and Test) | 353,890 | |

## G.5. USPTO-TPL

We use the USPTO-TPL dataset from RXNFP (Schwaller et al., 2021a). According to Fig. 12 and Tab. 26, the data distribution shares similarity with other datasets extracted from USPTO. However, the labels are extracted based on 1,000 templates. Reaction templates have strong sturctural pattern, which can be easily learnt by the model. Therefore, the difficulty of USPTO-TPL is low.

## G.6. Pistachio-Type

Considering the scale of the dataset, we only extract a portion of the data in Pistachio. First, we simplify the reaction type labels in Pistachio into 13 major categories. Each category contains multiple reaction templates, making it more complex than USPTO-TPL. Then, we discard the last category with too small quantity. From the remaining 12 categories, we select equal amount of data for each category, and then divide it into train/val and test sets.

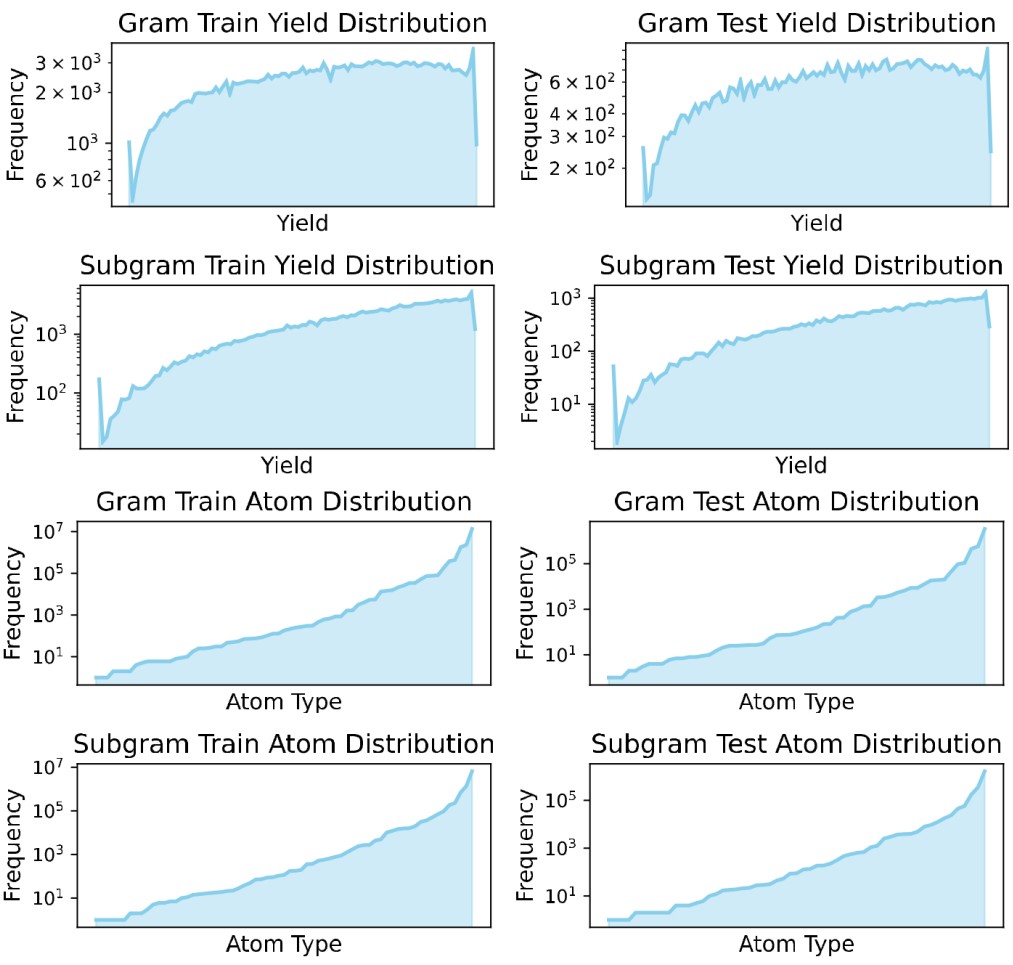

*Figure 11.* Data distributions of USPTO-Yield, including distribution of the number of data points in each bin of yield, and the distribution of atom types.

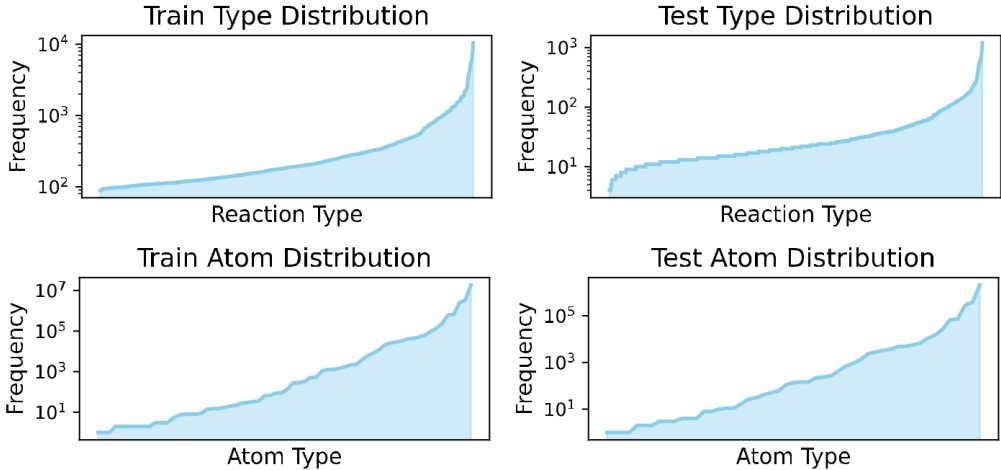

*Figure 12.* Data distributions of USPTO-TPL, including the distribution of the number of data points for various reaction types, as well as the distribution of atom types.

*Table 26.* Summary of USPTO-TPL, including metrics for measuring the complexity of data points, the overall complexity of the dataset, and the distribution differences between the train/val set and the test set.

| Metric | Train/Val | Test |
|---|---|---|
| Size | 400,604 | 44,511 |
| Mean Length | 124.92 | 125.15 |
| Mean Atoms | 11.29 | 11.32 |
| Mean Molecules | 5.91 | 5.91 |
| Max Length | 599 | 493 |
| Max Atoms | 332 | 243 |
| Atoms | 66 | 51 |
| Mean Atoms per Molecule | 24.99 | 23.31 |
| Max Atoms per Molecule | 164 | 99 |
| Max Molecules | 59 | 31 |
| Molecules | 581,458 | 88,219 |
| Different Molecules (Train/Val and Test) | 543,559 | |
| Num Types | 1000 | |

*Table 27.* Summary of Pistachio-Type, including metrics for measuring the complexity of data points, the overall complexity of the dataset, and the distribution differences between the train/val set and the test set.

| Metric | Train/Val | Test |
|---|---|---|
| Size | 656,640 | 72,960 |
| Mean Length | 133.91 | 133.98 |
| Mean Atoms | 13.04 | 13.04 |
| Mean Molecules | 5.36 | 5.35 |
| Max Length | 2706 | 1118 |
| Max Atoms | 926 | 480 |
| Mean Atoms per Molecule | 25.30 | 23.61 |
| Max Atoms per Molecule | 420 | 289 |
| Max Molecules | 64 | 48 |
| Molecules | 1,007,041 | 148,239 |
| Different Molecules (Train/Val and Test) | 946,727 | |
| Num Types | 12 | |

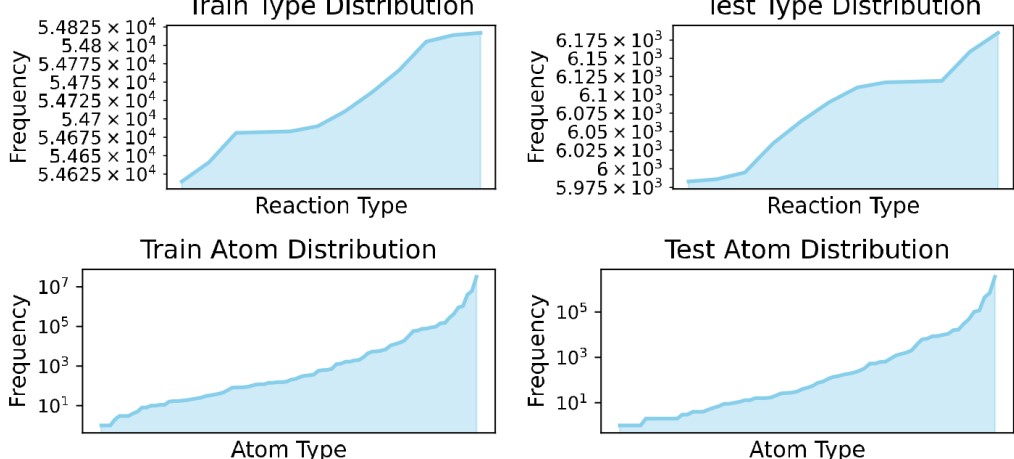

*Figure 13.* Data distributions of Pistachio-Type, including the distribution of the number of data points for various reaction types, as well as the distribution of atom types.

The final Pistachio-Type dataset has a large scale, with data distribution shown in Fig. 13 and dataset metrics shown in Tab. 27. The complexity of Pistachio-Type is significantly higher than USPTO-TPL, and it is closer to the real-world distribution of chemical data.

# H. Supplementary Experiments

## H.1. Attention Weights Analysis

### H.1.1. ATTENTION WEIGHTS VISUALIZATION

We provide the original images of the attention weights drawn by RDKit, and additionally selected 4 groups of observation subjects as supplements. Each group contains a molecule with multiple active functional groups and two associated reactions, with each reaction occurring on different functional groups of the molecule. The results are shown in Fig. 14

The results show that for these five molecules and their respective two reactions, the Reaction Graphs help the model accurately identify the reaction centers while reducing attention to irrelevant atoms. In contrast, the model using molecular graphs exhibits a similar distribution of attention weights for the target molecules in both reactions. It frequently focuses on functional groups that are less relevant to the reaction, leading to errors in prediction.

### H.1.2. FAILURE MODE ANALYSIS

We analyzed the edge cases and failure modes of the reaction center identification results for molecular graphs and Reaction Graph. As shown in Fig. 14, molecular graphs often mislocate the reaction center when multiple functional groups are present, as the model cannot obtain information related to the reaction changes from the molecular graph, making it difficult to determine which functional group is involved in the reaction. In contrast, incorrect localization of the reaction center in Reaction Graph is relatively rare, but still exist. As shown in Fig. 15, the typically reasons are: (1) Errors in the reaction itself; (2) Distraction of attention due to overly complex molecular structures; (3) Encountering rare chemical reactions that do not occur on functional groups.

Although the model using Reaction Graph still encounters failures, we observe that even in the failure cases of Reaction Graph, the model still shows a higher level of attention to the reaction center compared to molecular graphs. This validates the strong adaptability of Reaction Graph to edge cases.

### H.1.3. STATISTICAL VALIDATION

We conduct experiments on dataset scale to quantitatively validate that the Reaction Graph aids in identifying reaction centers.

We use USPTO dataset and use the algorithm described in Sec. E.3 to extract a test set of 1000 molecules, as well as 2000 corresponding reactions. We use the condition prediction model trained on USPTO-Condition to process these reactions.

Attention weights are extracted for reacting atoms and non-reacting atoms from the Set2Set layer for each target molecule. We calculate two metrics to evaluate the model's attention on the reacting atoms. First, we consider the proportion of attention weights for reacting atoms relative to the total. Second, we calculate the average attention weights for reacting atoms and non-reacting atoms, denoted as $a$ and $b$, and computed their ratio $a/b$. The results are shown in Tab. 28

*Table 28.* Influence of reaction information on attention weight results on USPTO.

| Method | Proportion | Ratio $a/b$ |
|---|---|---|
| Molecular Graph | 0.4326 | 1.4542 |
| Reaction Graph | **0.5079** | **2.0390** |

We further test the relationship between the attention weight results and reaction types, calculating the aforementioned metrics for each reaction type in the dataset. The results are shown in the Tab. 29 and 30. The correspondence between reaction type numbers and specific reaction types can be referenced in the Tab. 39.

*Table 29.* The proportion of attention weight for the reacting atoms, across various reaction types.

| Method | 0 | 1 | 2 | 3 | 4 | 5 | 6 | 7 | 8 | 9 | 10 | 11 |
|---|---|---|---|---|---|---|---|---|---|---|---|---|
| Molecular Graph | 0.65 | 0.49 | 0.50 | 0.50 | 0.61 | 0.35 | 0.58 | 0.50 | 0.38 | 0.46 | 0.28 | 0.04 |
| **Reaction Graph** | **0.70** | **0.53** | **0.54** | **0.54** | **0.66** | **0.38** | **0.65** | **0.57** | **0.40** | **0.52** | **0.32** | **0.06** |

*Table 30.* The ratio $a/b$ of attention weight between reacting and non-reacting atoms, across various reaction types.

| Method | 0 | 1 | 2 | 3 | 4 | 5 | 6 | 7 | 8 | 9 | 10 | 11 |
|---|---|---|---|---|---|---|---|---|---|---|---|---|
| Molecular Graph | 4.65 | 1.92 | 2.87 | 3.07 | 2.25 | 0.67 | 4.56 | 2.56 | 0.84 | 2.84 | 0.55 | 0.05 |
| **Reaction Graph** | **6.07** | **2.35** | **3.21** | **3.79** | **2.86** | **0.84** | **6.73** | **3.39** | **0.99** | **3.66** | **0.64** | **0.09** |

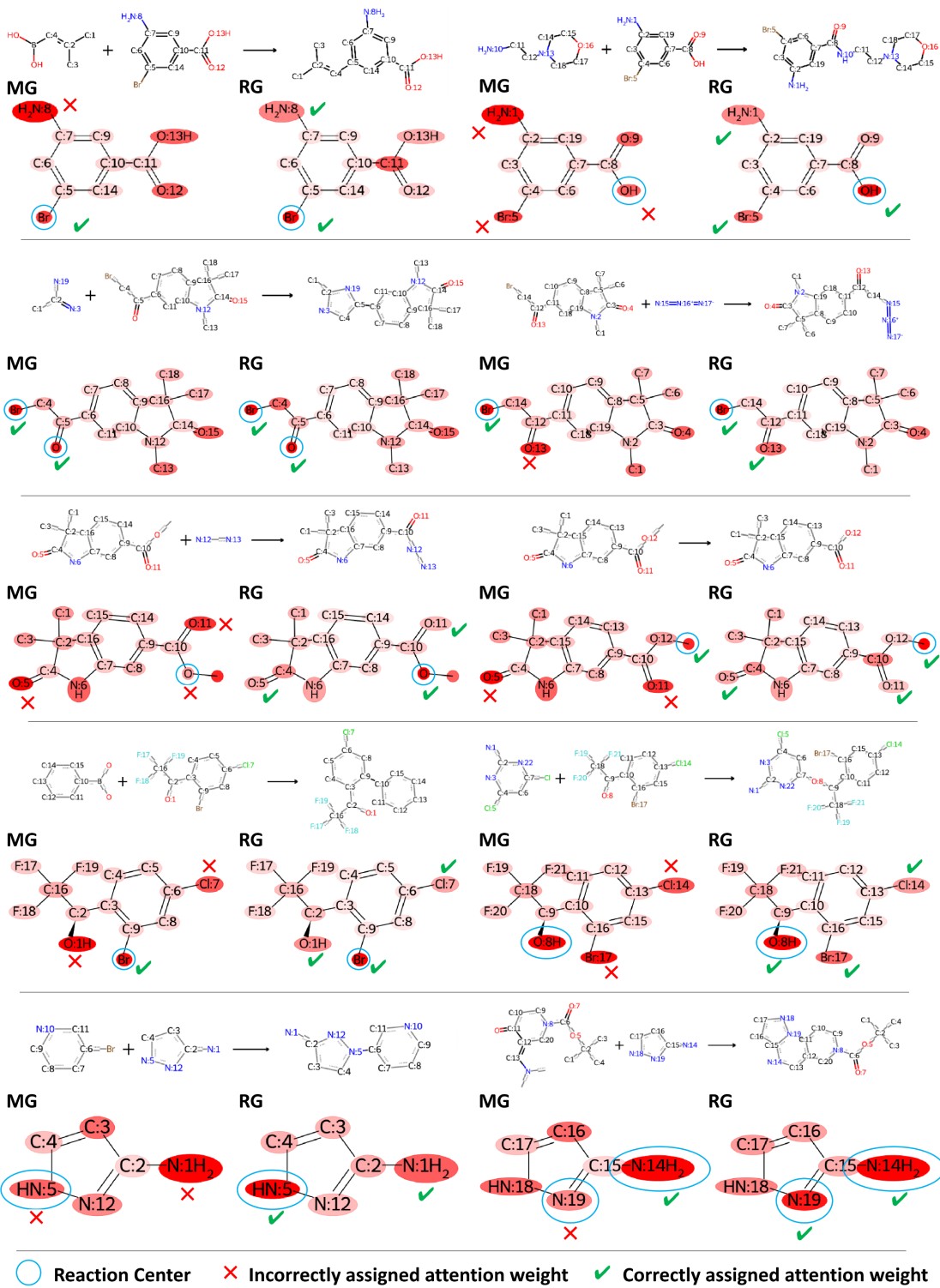

*Figure 14.* Visualization results of attention weights of the reaction condition prediction model on molecular graph and Reaction Graph. The depth of red represents the magnitude of attention weights, with deeper shades indicating larger attention weights.

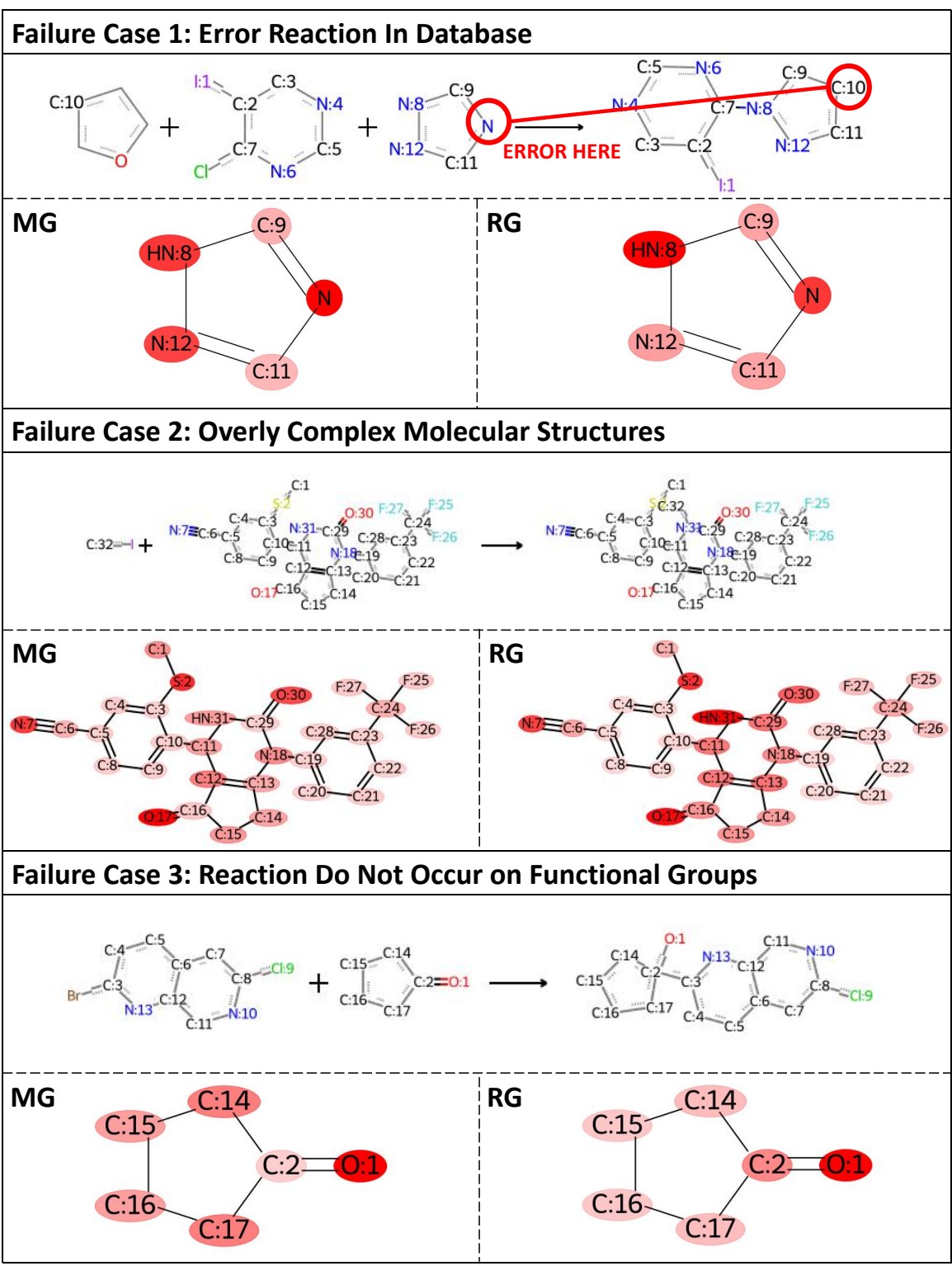

*Figure 15.* Visualization of attention weights for the failure cases. For the Reaction Graph, these cases can be categorized into three types, each being relatively rare. However, even in failure cases, the Reaction Graph (RG) still outperforms the Molecular Graph (MG).

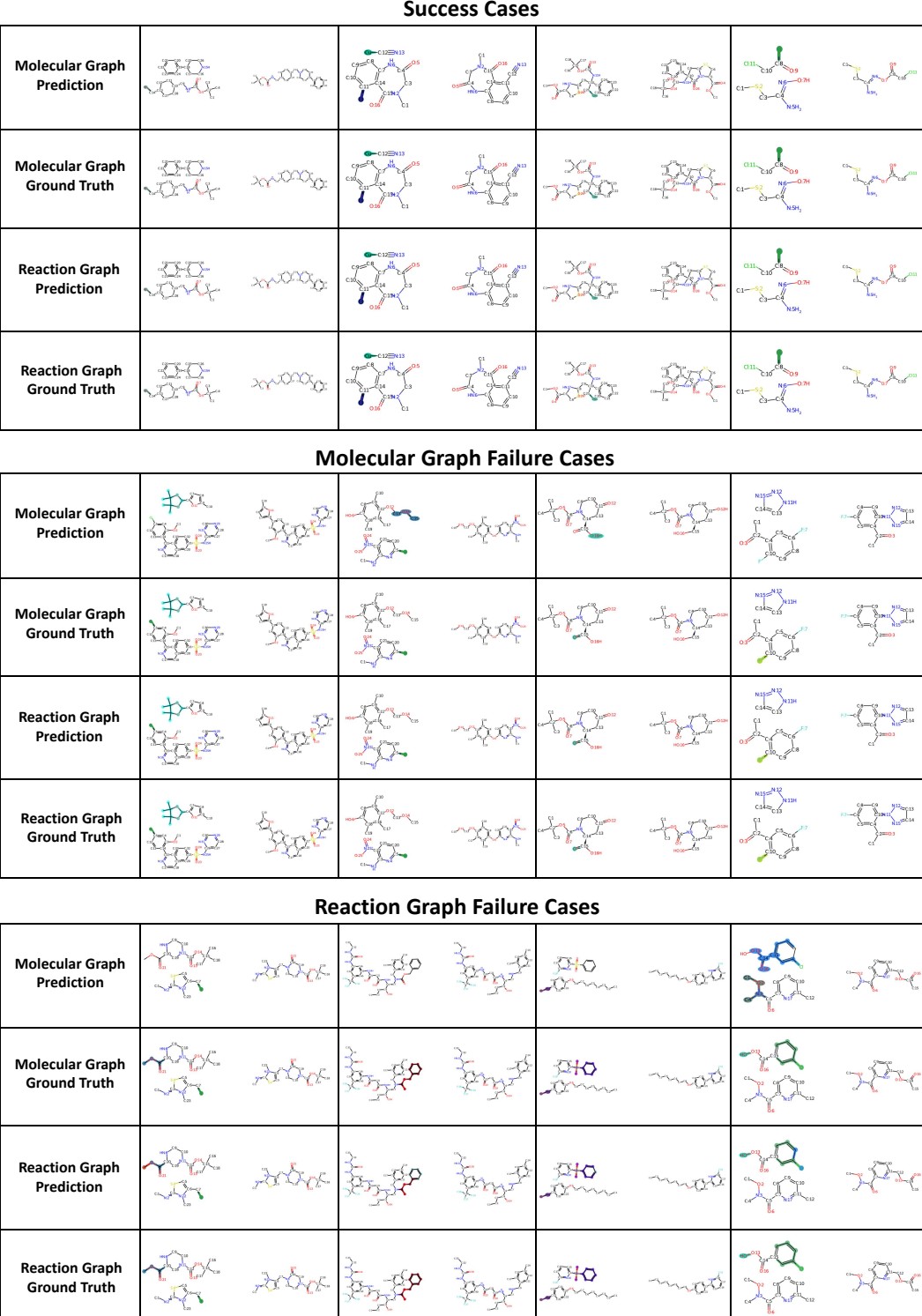

*Figure 16.* Visualization of the success and failure cases on leaving group identification. Molecular graphs often exhibit errors in positioning the leaving group when the molecule has multiple functional groups, while errors in Reaction Graph are relatively rare, typically occurring due to incorrect classification of certain atoms.

## H.2. Leaving Group Identification Analysis

### H.2.1. ERROR ANALYSIS OF MISCLASSIFIED CASES

We conduct a detailed analysis of the failure cases in the leaving group recognition task for molecular graphs and Reaction Graph, and we attempt to identify the failure modes.

We visualize the leaving group identification results of molecular graph and Reaction Graph. As shown in Fig. 16, molecular graph often misidentify leaving group positions, hindering accurate classification. In contrast, Reaction Graph position leaving groups correctly at most time, with minor errors mainly in specific type classification. This shows that Reaction Graph effectively helps the model to focus on features related to the reaction.

### H.2.2. ADDITIONAL BASELINE METHODS

We conduct extra experiment with D-MPNN and Rxn Hypergraph, and the results are shown in Tab. 31.

*Table 31.* Performance of different baseline methods on the Leaving Group Identification task.

| Method | Overall | | | LvG | | |
|---|---|---|---|---|---|---|
| | ACC | CEN | MCC | ACC | CEN | MCC |
| Molecular Graph | 0.950 | 0.036 | 0.549 | 0.448 | 0.201 | 0.519 |
| Rxn Hypergraph | 0.969 | 0.026 | 0.743 | 0.679 | 0.150 | 0.699 |
| D-MPNN | 0.993 | 0.003 | 0.949 | 0.902 | 0.051 | 0.899 |
| Reaction Graph | **0.997** | **0.002** | **0.973** | **0.947** | **0.031** | **0.945** |

*Table 32.* Results for statistical significance analysis of Reaction Graph on leaving group identification in USPTO, using $t$-test method from the mean and standard deviations of multiple trials with different random seeds. The baseline method is Molecular Graph.

| Metrics | Overall | | LvG-Specified | |
|---|---|---|---|---|
| | ACC | MCC | ACC | MCC |
| $-logP$ | 13.47 | 20.19 | 20.75 | 20.28 |

According to the results, Reaction Graph achieved the best performance in leaving group identification compared to other baselines.

### H.2.3. DETAILED RESULTS

**Standard Deviations.** Due to the large size of USPTO, and for the convenience of t-test, we conduct four random tests, which is closest to the square of the t-value (1.96) for a 95% confidence interval. We calculate the results of training under four different random seeds, and then compute the mean and standard deviation. The results are shown in Tab. 33.

*Table 33.* Results for leaving group identification of Molecular Graph and Reaction Graph on USPTO, including mean and standard deviation from multiple trials with different random seeds.

| Method | Overall | | | LvG-Specified | | |
|---|---|---|---|---|---|---|
| | ACC | CEN | MCC | ACC | CEN | MCC |
| Molecular Graph | 0.950±0.001 | 0.037±0.001 | 0.538±0.006 | 0.423±0.004 | 0.209±0.002 | 0.497±0.005 |
| Reaction Graph | **0.996±0.001** | **0.002±0.001** | **0.971±0.001** | **0.944±0.001** | **0.035±0.001** | **0.942±0.001** |

**Statistical Significance Tests.** Statistical significance testing is essential for determining whether the Reaction Graph demonstrates an improvement over the baseline method. For each task, we selected the optimal baseline and employed the $t$-test to assess statistical significance. The formula used is $t = \sqrt{n} \cdot \frac{x-b}{s}$, where $t$ represents the t-statistic, indicating the size of the difference relative to the variation in the sample data; $n$ is the sample size, which in this case is 4; $x$ is the mean performance of the Reaction Graph; $b$ is the baseline performance; and $s$ is the standard deviation of the performance measurements. By calculating the $t$-statistic, we can evaluate whether the observed improvement in the Reaction Graph is statistically significant compared to the baseline method.

The significance level P corresponding to the $t$-value can be found in the lookup table[28], and we use $-\log P$ to make the results easier to observe; the larger the value, the higher the probability that the Reaction Graph shows an improvement over the original methods.

We choose Molecular Graph as the baseline method to calculate the $t$-values, and the results are shown in Tab. 32.

---

[28]https://en.wikipedia.org/wiki/Student%27s_t-distribution

**Cross-validation Results.** Cross-validation can reduce the bias introduced by the unevenly distributed chemical reaction dataset, ensuring the model's generalization. We test the model's performance using K-fold cross-validation. This means splitting the dataset into K parts, using one part for test and the remaining for train each time. Considering the training cost and the reuse of experiment results, we divide the dataset into 9 folds (excluding the original test set) and conduct three random tests. Including the validation results from the original splits, there are a total of four tests. We take the average as the result of K-fold cross-validation. The results are shown in Tab. 34.

*Table 34.* Cross-validation results for leaving group identification of Molecular Graph and Reaction Graph on USPTO, including mean and standard deviation from multiple trials with different train/test splits.

| Method | Overall | | | LvG-Specified | | |
|---|---|---|---|---|---|---|
| | ACC | CEN | MCC | ACC | CEN | MCC |
| Molecular Graph | 0.948±0.004 | 0.038±0.003 | 0.535±0.015 | 0.418±0.010 | 0.203±0.001 | 0.495±0.008 |
| Reaction Graph | **0.995±0.002** | **0.004±0.001** | **0.955±0.012** | **0.915±0.022** | **0.050±0.011** | **0.912±0.022** |

**Confidence Intervals.** We utilize the normal distribution method to calculate the confidence interval (CI) for our model using the formula $CI = \bar{x} \pm z \cdot \frac{s}{\sqrt{n}}$, where $\bar{x}$ represents the sample mean, $z$ is the z-score corresponding to the desired confidence level, $s$ is the standard deviation of the sample, and $n$ is the sample size. For a 95% confidence level, the corresponding z-score is $z = 1.96$. We use the results from Tab. 33 to calculate the 95% confidence interval. The results are shown in Tab. 35.

*Table 35.* Confidence interval results ([Min,Max]) for leaving group identification of Molecular Graph and Reaction Graph on USPTO, calculated using normal distribution method from mean and standard deviations of multiple trials with different random seeds.

| Method | Type | Overall | | | LvG-Specified | | |
|---|---|---|---|---|---|---|---|
| | | ACC | CEN | MCC | ACC | CEN | MCC |
| Molecular Graph | Min | 0.949 | 0.036 | 0.532 | 0.419 | 0.207 | 0.492 |
| | Max | 0.951 | 0.038 | 0.544 | 0.427 | 0.211 | 0.502 |
| Reaction Graph | Min | **0.995** | **0.001** | **0.970** | **0.943** | **0.034** | **0.941** |
| | Max | **0.997** | **0.003** | **0.972** | **0.945** | **0.036** | **0.943** |

## H.3. Hyperparameter Selection.

We test several hyperparameters that have the greatest impact on performance, specifically the hidden layer dimension $D_v$ of MPNN and the number of iterations $T_1$, as well as the number of iterations $T_2$ for Set2Set. We conduct experiments on the USPTO-Condition dataset.

*Table 36.* Top-$k$ accuracies on USPTO-Condition under different hyperparameter combinations, including hidden layer dimensions and iterations of GNN and pooling. Each has 3 candidate values.

| Hidden Dim $D_v$ | MPNN Iters $T_1$ | Pooling Iters $T_2$ | Top-1↑ | Top-3↑ | Top-5↑ | Top-10↑ | Top-15↑ |
|---|---|---|---|---|---|---|---|
| 200 | 3 | 2 | **0.325** | **0.434** | **0.472** | 0.506 | 0.518 |
| 50 | 3 | 2 | 0.307 | 0.419 | 0.456 | 0.492 | 0.505 |
| 100 | 3 | 2 | 0.315 | 0.427 | 0.464 | 0.500 | 0.512 |
| 200 | 2 | 2 | 0.320 | 0.433 | 0.471 | **0.508** | **0.520** |
| 200 | 4 | 2 | 0.317 | 0.428 | 0.466 | 0.502 | 0.514 |
| 200 | 3 | 1 | 0.319 | 0.430 | 0.467 | 0.502 | 0.514 |
| 200 | 3 | 3 | 0.318 | 0.429 | 0.465 | 0.501 | 0.514 |

According to Tab. 36, the hyperparameter combination we selected achieves optimal overall performance. Further reducing the number of iterations for MPNN may enhance the model's top-15 performance, but it significantly affects the Top-1 performance. We also observe that as the hidden layer dimension increases, the model's performance gradually improves, indicating that our model has scalability. Due to GPU memory limitations, we do not continue to try higher hidden layer dimensions to ensure high training efficiency.

## H.4. Reaction Condition Prediction

### H.4.1. METRIC CALCULATION

In the reaction condition prediction task, the most commonly used metric is the top-$k$ accuracy. Formally speaking, given a ground truth condition $\boldsymbol{c} = [\boldsymbol{c}_{catalyst}, \boldsymbol{c}_{solvent1}, \boldsymbol{c}_{solvent2}, \boldsymbol{c}_{reagent1}, \boldsymbol{c}_{reagent2}]$, to calculate top-$k$ accuracy, we allow the model to generate $k$ sets of labels $\mathbb{C} = \{\hat{\boldsymbol{c}}^1, \hat{\boldsymbol{c}}^2, \dots, \hat{\boldsymbol{c}}^k\}$. Let $a$ represent the correctness of model's prediction on this data sample, and $a_{category}$ represent the correctness of model's prediction on this one label category, where category can be one of catalyst, solvent1, solvent2, reagent1 and reagent2. We have:

$$a = \begin{cases} 1, & \hat{\boldsymbol{c}}^i = \boldsymbol{c}, \exists\, i \in \{0, 1, \dots, k\}, \\ 0, & otherwise, \end{cases} \tag{13}$$

$$a_{category} = \begin{cases} 1, & \hat{\boldsymbol{c}}^i_{category} = \boldsymbol{c}_{category}, \exists\, i \in \{0, 1, \dots, k\}, \\ 0, & otherwise, \end{cases} \tag{14}$$

And assume the correctness of model on data sample $i$ is $a^i$, then the overall top-$k$ accuracy on the whole dataset is $\overline{a} = \frac{\sum_i^{N_d} a^i}{N_d}$, and the top-$k$ accuracy of a label category is $\overline{a}_{category} = \frac{\sum_i^{N_d} a^i_{category}}{N_d}$, where $N_d$ is the number of data samples in the test set.

It is important to note that the overall accuracy calculation aligns with our intuitive understanding, while the accuracy calculation for each label category is not. This is because it is possible that $\hat{\boldsymbol{c}}^i_{category} = \hat{\boldsymbol{c}}^j_{category}$, where $i, j \in \{0, 1, \dots, k\}$, meaning the number of different classification results output by the model is actually less than $k$. Therefore, this evaluation criterion is stricter than our intuitive understanding. In other words, we do not generate $k$ labels for each category separately to calculate the top-$k$ accuracy of it.

### H.4.2. DETAILED RESULTS

**Top-$k$ Accuracy for Each Condition.** Following Gao et al. (2018); Wang et al. (2023b), we test the top-$k$ accuracy for the comparison methods in Sec. 3. According to Tab. 37 and 38, the model using RG surpasses existing methods on the majority of top-$k$ accuracy, further demonstrating the effectiveness of our proposed approach. We also note that Parrot shows performance advantages in certain specific condition categories, indicating the effectiveness and potential of large-scale pre-training.

**Standard Deviations.** We calculated the mean and standard deviation using the results from four different random seeds. The results are shown in Tab. 40.

**Statistical Significance Tests.** We choose Parrot (Wang et al., 2023b) as the baseline method to calculate the $t$-values, and the results are shown in Tab. 41.

**Cross-validation Results.** We conduct K-fold cross-validation using the settings in Sec. H.2.3 on the USPTO-Condition and Pistachio-Condition datasets. The results are shown in Tab. 42.

**Confidence Intervals.** We use the same settings in Sec. H.2.3 on the USPTO-Condition and Pistachio-Condition datasets to calculate confidence intervals. The results are shown in Tab. 43.

**Confusion Matrix Analysis.** We perform a confusion matrix analysis of the model's classification results in Fig. 18. Due to the uneven data distribution in the dataset, we apply row normalization to the confusion matrix values, dividing each row's values by the sum of that row. Based on the results, the model can achieve correct classification results for most categories. However, the columns corresponding to the $None$ category and some frequently occurring categories are similarly dark. This is due to the data distribution of the model. From the label distribution of the dataset shown in Sec. G, the number of certain high-frequency categories is significantly higher than that of other categories, leading the model to favor predicting the labels of these categories. Although the category weights and label smoothing methods can alleviate these problems, improving performance on these sparse samples still relies on the inclusion of high-quality data.

*Table 37.* Detailed results of top-$k$ accuracies of comparison methods on USPTO-Condition, including performance for each reaction condition category.

| Method | Conditions | Top-1↑ | Top-3↑ | Top-5↑ | Top-10↑ | Top-15↑ |
|---|---|---|---|---|---|---|
| CRM | catalyst | 0.9219 | 0.9219 | 0.9219 | 0.9219 | 0.9219 |
| | solvent-1 | 0.5015 | 0.6640 | 0.7055 | 0.7340 | 0.7346 |
| | solvent-2 | **0.8130** | 0.8369 | 0.8461 | 0.8525 | 0.8527 |
| | reagent-1 | 0.4972 | 0.6597 | 0.7402 | 0.8184 | 0.8516 |
| | reagent-2 | 0.7622 | 0.8408 | 0.8664 | 0.8876 | 0.8986 |
| | overall | 0.2596 | 0.3771 | 0.4206 | 0.4612 | 0.4717 |
| AR-GCN | catalyst | 0.9024 | 0.9024 | 0.9024 | 0.9024 | 0.9024 |
| | solvent-1 | 0.4114 | 0.5787 | 0.6295 | 0.6635 | 0.6650 |
| | solvent-2 | 0.8093 | 0.8093 | 0.8093 | 0.8093 | 0.8093 |
| | reagent-1 | 0.4200 | 0.5740 | 0.6667 | 0.7515 | 0.7622 |
| | reagent-2 | 0.7486 | 0.7486 | 0.7486 | 0.7486 | 0.7486 |
| | overall | 0.1460 | 0.2374 | 0.2733 | 0.3121 | 0.3261 |
| CIMG-Condition | catalyst | 0.9146 | 0.9146 | 0.9146 | 0.9146 | 0.9146 |
| | solvent-1 | 0.4218 | 0.6139 | 0.6542 | 0.6780 | 0.6789 |
| | solvent-2 | 0.8110 | 0.8110 | 0.8110 | 0.8110 | 0.8110 |
| | reagent-1 | 0.4351 | 0.5685 | 0.6665 | 0.7462 | 0.7598 |
| | reagent-2 | 0.7574 | 0.7574 | 0.7574 | 0.7574 | 0.7574 |
| | overall | 0.1839 | 0.2714 | 0.3026 | 0.3391 | 0.3525 |
| Parrot | catalyst | 0.9250 | 0.9250 | 0.9250 | 0.9250 | 0.9250 |
| | solvent-1 | 0.5018 | 0.6858 | **0.7311** | **0.7536** | **0.7543** |
| | solvent-2 | 0.8096 | 0.8426 | 0.8521 | 0.8582 | 0.8585 |
| | reagent-1 | 0.5039 | 0.6820 | 0.7629 | **0.8436** | **0.8776** |
| | reagent-2 | 0.7648 | 0.8486 | 0.8774 | 0.8998 | 0.9110 |
| | overall | 0.2691 | 0.4035 | 0.4510 | 0.4914 | 0.5031 |
| D-MPNN | catalyst | 0.9198 | 0.9198 | 0.9198 | 0.9198 | 0.9198 |
| | solvent-1 | 0.4621 | 0.6295 | 0.6583 | 0.7177 | 0.7192 |
| | solvent-2 | 0.8120 | 0.8120 | 0.8120 | 0.8120 | 0.8120 |
| | reagent-1 | 0.4777 | 0.6272 | 0.7449 | 0.8067 | 0.8089 |
| | reagent-2 | **0.7702** | 0.7702 | 0.7702 | 0.7702 | 0.7702 |
| | overall | 0.1977 | 0.3000 | 0.3341 | 0.3780 | 0.3924 |
| Rxn Hypergraph | catalyst | 0.9160 | 0.9160 | 0.9160 | 0.9160 | 0.9160 |
| | solvent-1 | 0.4676 | 0.6309 | 0.6767 | 0.7077 | 0.7095 |
| | solvent-2 | 0.8089 | 0.8089 | 0.8089 | 0.8089 | 0.8089 |
| | reagent-1 | 0.4761 | 0.6246 | 0.7105 | 0.7844 | 0.7937 |
| | reagent-2 | 0.7642 | 0.7642 | 0.7642 | 0.7642 | 0.7642 |
| | overall | 0.2127 | 0.3084 | 0.3447 | 0.3808 | 0.3927 |
| Reaction Graph | catalyst | **0.9316** | **0.9316** | **0.9316** | **0.9316** | **0.9316** |
| | solvent-1 | **0.5429** | **0.6925** | 0.7265 | 0.7475 | 0.7481 |
| | solvent-2 | 0.8075 | **0.8564** | **0.8654** | **0.8723** | **0.8725** |
| | reagent-1 | **0.5343** | **0.6982** | **0.7713** | 0.8420 | 0.8729 |
| | reagent-2 | 0.7630 | **0.8663** | **0.8928** | **0.9119** | **0.9193** |
| | overall | **0.3246** | **0.4343** | **0.4715** | **0.5061** | **0.5181** |

*Table 38.* Detailed results of top-$k$ accuracies on Pistachio-Condition, including performance for each reaction condition category.

| Method | Conditions | Top-1↑ | Top-3↑ | Top-5↑ | Top-10↑ | Top-15↑ |
|---|---|---|---|---|---|---|
| CRM | catalyst | 0.9943 | 0.9943 | 0.9943 | 0.9943 | 0.9943 |
| | solvent-1 | 0.5188 | 0.6954 | 0.7281 | 0.7580 | 0.7584 |
| | solvent-2 | 0.8406 | 0.9004 | 0.9077 | 0.9134 | 0.9135 |
| | reagent-1 | 0.4287 | 0.5990 | 0.6640 | 0.7350 | 0.7643 |
| | reagent-2 | 0.7120 | 0.8326 | 0.8603 | 0.8924 | 0.9055 |
| | overall | 0.3300 | 0.4692 | 0.5098 | 0.5476 | 0.5538 |
| Parrot | catalyst | 0.9951 | 0.9951 | 0.9951 | 0.9951 | 0.9951 |
| | solvent-1 | 0.5084 | 0.7667 | 0.7981 | 0.8064 | 0.8065 |
| | solvent-2 | 0.8417 | 0.9124 | 0.9160 | 0.9167 | 0.9168 |
| | reagent-1 | 0.4315 | 0.6438 | 0.7236 | 0.8057 | 0.8314 |
| | reagent-2 | 0.7341 | 0.8642 | 0.8944 | **0.9225** | **0.9348** |
| | overall | 0.3500 | 0.5323 | 0.5883 | 0.6263 | 0.6301 |
| D-MPNN | catalyst | 0.9940 | 0.9940 | 0.9940 | 0.9940 | 0.9940 |
| | solvent-1 | 0.5015 | 0.6485 | 0.6871 | 0.7830 | 0.7872 |
| | solvent-2 | 0.8614 | 0.8614 | 0.8614 | 0.8614 | 0.8614 |
| | reagent-1 | 0.4061 | 0.5727 | 0.6844 | 0.7461 | 0.7480 |
| | reagent-2 | 0.7491 | 0.7491 | 0.7491 | 0.7491 | 0.7491 |
| | overall | 0.2586 | 0.3422 | 0.3775 | 0.4415 | 0.4693 |
| Rxn Hypergraph | catalyst | 0.9945 | 0.9945 | 0.9945 | 0.9945 | 0.9945 |
| | solvent-1 | 0.5173 | 0.6841 | 0.7466 | 0.7895 | 0.7925 |
| | solvent-2 | **0.8619** | 0.8619 | 0.8619 | 0.8619 | 0.8619 |
| | reagent-1 | 0.4246 | 0.5793 | 0.6608 | 0.7373 | 0.7470 |
| | reagent-2 | **0.7520** | 0.7520 | 0.7520 | 0.7520 | 0.7520 |
| | overall | 0.2881 | 0.3671 | 0.4117 | 0.4636 | 0.4851 |
| Reaction Graph | catalyst | **0.9952** | **0.9952** | **0.9952** | **0.9952** | **0.9952** |
| | solvent-1 | **0.5579** | **0.7791** | **0.8032** | **0.8130** | **0.8130** |
| | solvent-2 | 0.8539 | **0.9214** | **0.9248** | **0.9261** | **0.9261** |
| | reagent-1 | **0.4884** | **0.6767** | **0.7456** | **0.8123** | **0.8384** |
| | reagent-2 | 0.7416 | **0.8733** | **0.8964** | 0.9206 | 0.9319 |
| | overall | **0.3915** | **0.5566** | **0.6039** | **0.6384** | **0.6432** |

*Table 39.* Correspondence between reaction type labels and reaction types. The class here correspond to the label of Pistachio-Type.

| Label | Name |
|---|---|
| 0 | Unrecognized |
| 1 | Heteroatom alkylation and arylation |
| 2 | Acylation and related processes |
| 3 | C-C bond formation |
| 4 | Heterocycle formation |
| 5 | Protections |
| 6 | Deprotections |
| 7 | Reductions |
| 8 | Oxidations |
| 9 | Functional group interconversion (FGI) |
| 10 | Functional group addition (FGA) |
| 11 | Resolutions |

*Table 40.* Results for reaction condition prediction of Reaction Graph on USPTO-Condition and Pistachio-Condition, including mean and standard deviation from multiple trials with different random seeds.

| Dataset | Top-1 | Top-3 | Top-5 | Top-10 | Top-15 |
|---|---|---|---|---|---|
| USPTO-Condition | 0.322±0.002 | 0.432±0.003 | 0.469±0.003 | 0.504±0.003 | 0.516±0.003 |
| Pistachio-Condition | 0.391±0.001 | 0.556±0.001 | 0.602±0.001 | 0.636±0.001 | 0.641±0.001 |

*Table 41.* Results for statistical significance analysis of Reaction Graph on reaction condition prediction in USPTO-Condition and Pistachio-Condition, using $t$-test method from the mean and standard deviations of multiple trials with different random seeds. The baseline method is Parrot. The value in the table is $-logP$.

| Dataset | Top-1 | Top-3 | Top-5 | Top-10 | Top-15 |
|---|---|---|---|---|---|
| USPTO-Condition | 11.81 | 8.69 | 7.38 | 6.43 | 6.43 |
| Pistachio-Condition | 13.12 | 11.52 | 9.90 | 8.90 | 9.18 |

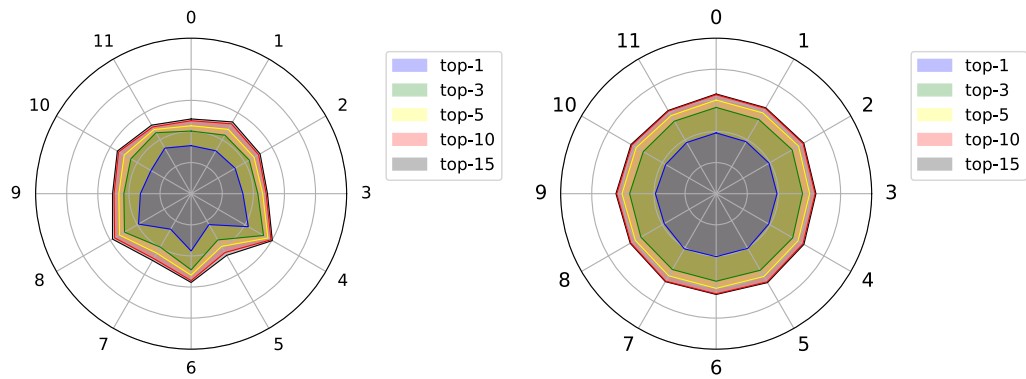

*Figure 17.* Radar chart of condition prediction accuracy under various reaction types. The results show that the USPTO-Condition prediction results have stronger correlation with reaction categories than that of Pistachio-Condition.

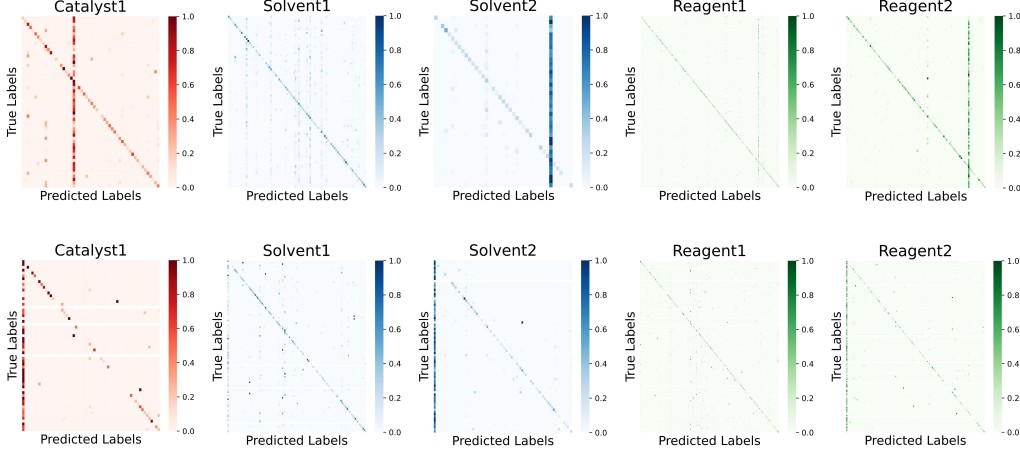

*Figure 18.* Normalized confusion matrices of condition prediction results on USPTO Condition dataset, using the proposed model architecture with Reaction Graph.

*Table 42.* Cross-validation results for reaction condition prediction of Reaction Graph on USPTO-Condition and Pistachio-Condition, including mean and standard deviation from multiple trials with different train/test splits.

| Dataset | Top-1 | Top-3 | Top-5 | Top-10 | Top-15 |
|---|---|---|---|---|---|
| USPTO-Condition | 0.320±0.003 | 0.430±0.002 | 0.467±0.002 | 0.503±0.002 | 0.516±0.002 |
| Pistachio-Condition | 0.391±0.002 | 0.556±0.002 | 0.603±0.001 | 0.636±0.001 | 0.641±0.001 |

*Table 43.* Confidence interval results ([Min,Max]) for reaction condition prediction of Reaction Graph on USPTO-Condition and Pistachio-Condition dataset, calculated using normal distribution method from mean and standard deviations of multiple trials with different random seeds.

| Dataset | Type | Top-1 | Top-3 | Top-5 | Top-10 | Top-15 |
|---|---|---|---|---|---|---|
| USPTO-Condition | Min | 0.320 | 0.429 | 0.466 | 0.501 | 0.513 |
|  | Max | 0.324 | 0.435 | 0.472 | 0.507 | 0.519 |
| Pistachio-Condition | Min | 0.390 | 0.555 | 0.601 | 0.635 | 0.640 |
|  | Max | 0.392 | 0.557 | 0.603 | 0.637 | 0.642 |

*Table 44.* Influence of the output head on model performance. We test the prediction accuracy using different output heads based on Molecular Graph on the USPTO-Condition dataset. The results show that the CRM output head we use is more effective.

| Method | Cond | Top-1↑ | Top-3↑ | Top-5↑ | Top-10↑ | Top-15↑ |
|---|---|---|---|---|---|---|
| MLP Output Head | c1 | 0.922 | 0.922 | 0.922 | 0.922 | 0.922 |
|  | s1 | 0.501 | 0.653 | **0.712** | **0.734** | **0.7347** |
|  | s2 | 0.799 | 0.799 | 0.799 | 0.799 | 0.799 |
|  | r1 | **0.503** | 0.638 | 0.727 | 0.797 | 0.811 |
|  | r2 | **0.763** | 0.763 | 0.763 | 0.763 | 0.763 |
|  | all | 0.249 | 0.305 | 0.318 | 0.387 | 0.422 |
| CRM Output Head | c1 | **0.926** | **0.926** | **0.926** | **0.926** | **0.926** |
|  | s1 | **0.511** | **0.656** | 0.693 | 0.715 | 0.716 |
|  | s2 | **0.803** | **0.853** | **0.863** | **0.869** | **0.869** |
|  | r1 | 0.500 | **0.671** | **0.743** | **0.821** | **0.856** |
|  | r2 | 0.753 | **0.861** | **0.887** | **0.908** | **0.917** |
|  | all | **0.298** | **0.400** | **0.437** | **0.472** | **0.484** |

*Table 45.* Influence of the two-stage training on model performance, using USPTO-Condition. The model is based on Reaction Graph. The results indicate that the proposed two-stage training strategy significantly improves the top-$k$ accuracy.

| Method | Cond | Top-1↑ | Top-3↑ | Top-5↑ | Top-10↑ | Top-15↑ |
|---|---|---|---|---|---|---|
| One-Stage | c1 | 0.928 | 0.928 | 0.928 | 0.928 | 0.928 |
|  | s1 | 0.517 | 0.664 | 0.702 | 0.725 | 0.725 |
|  | s2 | 0.799 | 0.855 | **0.866** | **0.873** | **0.874** |
|  | r1 | 0.508 | 0.685 | 0.758 | 0.833 | 0.869 |
|  | r2 | 0.754 | 0.866 | 0.891 | 0.911 | **0.920** |
|  | all | 0.304 | 0.413 | 0.449 | 0.484 | 0.495 |
| Two-Stage | c1 | **0.932** | **0.932** | 932 | **0.932** | **0.932** |
|  | s1 | **0.543** | **0.693** | **0.727** | **0.748** | **0.748** |
|  | s2 | **0.808** | **0.856** | 0.865 | 0.872 | 0.873 |
|  | r1 | **0.534** | **0.698** | **0.771** | **0.842** | **0.873** |
|  | r2 | **0.763** | **0.866** | **0.893** | **0.912** | 0.919 |
|  | all | **0.325** | **0.434** | **0.472** | **0.506** | **0.518** |

**Reaction Type Analysis.** In addition, we analyze the classification accuracy of different reaction types across the two datasets. The correspondence between the reaction type indices and the names of reaction types is shown in Tab. 39, while the results are illustrated in Fig. 17. The categories in the USPTO-Condition dataset are classified using the NameRXN tool from (Wang et al., 2023b), while the category labels in the Pistachio-Condition dataset are derived from the labels in the Pistachio database.

The accuracy in USPTO-Condition is correlated with the reaction categories, while the correlation is less evident in Pistachio-Condition. This reflects that reaction type distribution has impact on model performance. Meanwhile, when $k$ increases, the growth of top-$k$ accuracy gradually slows down, suggesting that the choice of top-15 is reasonable. Further increasing the value of $k$ does not effectively improve the model's performance and may lead to higher costs for inference.

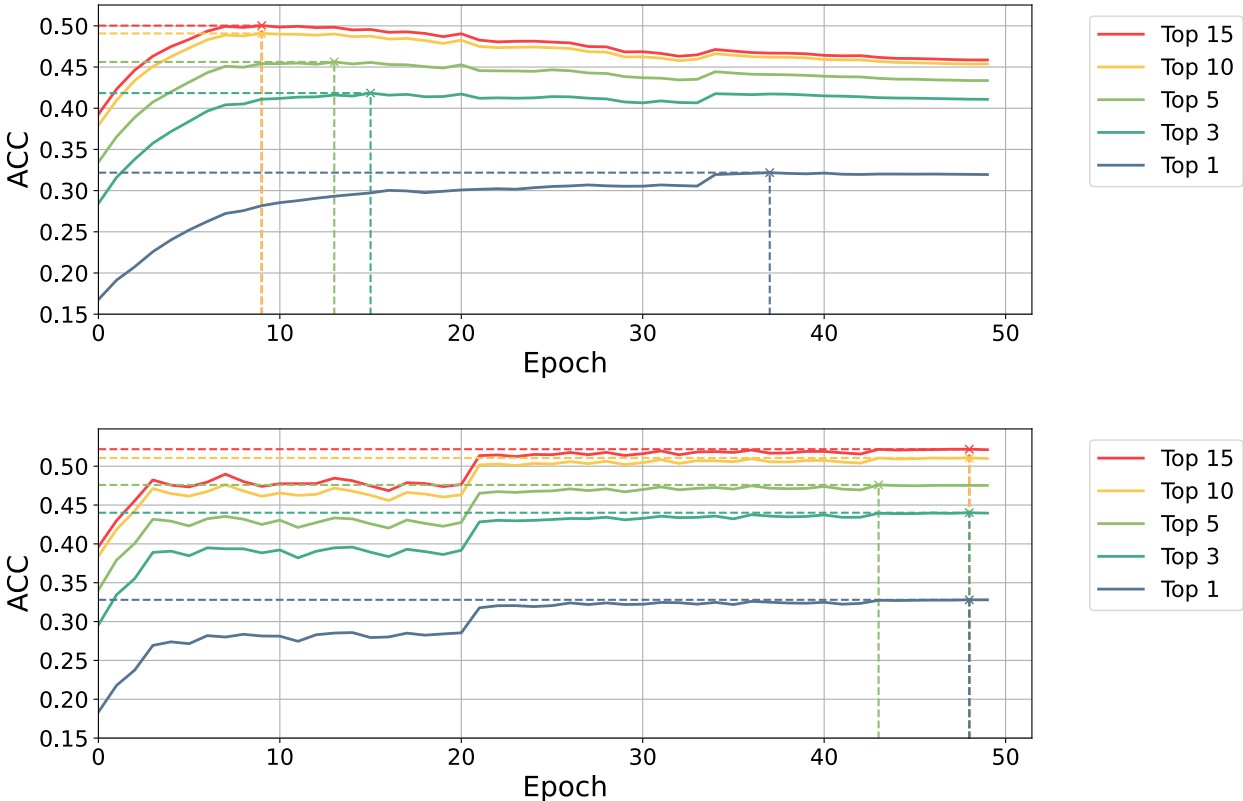

*Figure 19.* Top-$k$ validation accuracies of two-stage training, where the positions of the optimal results are marked with crosses. The upper figure is from the first stage, and the lower figure is from the second stage. The positions where the accuracies change sharply correspond to the points where the scheduler adjusts the learning rate.

### H.4.3. ADDITIONAL ABLATION

**Output Head.** Following Gao et al. (2018), we use the CRM output head. It autoregressively output conditions, and significantly enhances the performance. In contrast, the MLP outputs all conditions simultaneously, which results in a loss of overall accuracy. To demonstrate the effectiveness of our model architecture design, we eliminate additional influencing factors, and test the model performance of the MLP and the CRM output head under the Molecular Graph.

Based on the results shown in Tab. 44, the CRM output head effectively improves the model's overall top-$k$ accuracy. However, for each type of reaction condition, the model using the MLP output head achieves accuracy comparable to that of the CRM. This indicates that, compared to MLP, CRM can fully consider the previous prediction results when provide the next prediction, allowing it to generate more suitable combinations of reaction conditions.

**Training Strategy.** We propose a two-stage training strategy, while also incorporating the technique of label smoothing. We test the effects of these two factors on model performance in one experiment. Specifically, we use a loss function that includes label smoothing and class weight to train the RG model on the USPTO-Condition dataset and observe the changes in validation top-$k$ accuracy.

According to the results in Tab. 19, the proposed two-stage training strategy effectively enhances the model's performance. Without two-stage training, the timing of the best performance differs for each top-$k$ metric. Especially for the top-15 accuracy, it appears overfitted after reaching its peak. By introducing two-stage training, the timing of the best performance for top-1 to top-15 accuracy is consistent.

Comparing the results in Fig. 19 and Tab. 45 reveals that the inclusion of label smoothing results in more imbalance in top-k performance. Therefore, we ultimately do not use label smoothing on USPTO-Condition.

H.4.4. MERGING DATASET

In this paper, we focus on proposing a novel chemical reaction representation, which can enhance model performance within a fixed data volume. Exploring the effect of training with large-scale data is another interesting research topic, which is not the focus of this paper. Nevertheless, we conduct additional experiments to explore whether combining the USPTO and Pistachio datasets can provide valuable insights. Since the reaction category labels between the two datasets are not completely consistent, we select reactions from Pistachio that match the categories in USPTO. We perform joint training using the merged dataset.

*Table 46.* Reaction classification results before and after mixing USPTO with Pistachio, on the test set of original USPTO-Condition and the OOD test set of Pistachio.

| Method | USPTO-Condition Test Set | | | | | Pistachio-OOD Test Set | | | | |
|---|---|---|---|---|---|---|---|---|---|---|
| | Top-1↑ | Top-3↑ | Top-5↑ | Top-10↑ | Top-15↑ | Top-1↑ | Top-3↑ | Top-5↑ | Top-10↑ | Top-15↑ |
| w/o Pistachio | **0.325** | 0.434 | **0.472** | **0.506** | **0.518** | 0.177 | 0.235 | 0.260 | 0.284 | 0.292 |
| w Pistachio | 0.323 | **0.436** | 0.470 | 0.500 | 0.511 | **0.275** | **0.455** | **0.517** | **0.567** | **0.578** |

The results on the USPTO test set are as the left side of Tab. 46. Mixing the two datasets for training does not improve the model's performance on a single dataset. This may result from their significant differences in data distribution. Although USPTO and Pistachio are large in scale, their sparse annotations still cannot cover the diverse chemical space.

We further evaluate the effect of mixed data training on model generalization. Specifically, we extract a subset of data from Pistachio to serve as the test set. This subset is out-of-distribution (OOD), which can reflect the model's generalization performance. The results are as the right side of Tab. 46. Training with mixed data significantly improved performance on the Pistachio-OOD test set, while keeping the performance of model on the original USPTO-Condition dataset, indicating a benefit for generalization.

## H.5. Reaction Yield Prediction

H.5.1. METRIC CALCULATION

In yield prediction task, in addition to the commonly used $R^2$ metric, Mean Absolute Error (MAE) and Root Mean Squared Error (RMSE) are also frequently utilized. Assuming the ground truth yield of $i$-th data sample is $y_i$, the model's output mean is $\hat{\mu}_{y_i}$, then the MAE of the prediction result is $\frac{\sum_i^{N_d} |y_i - \hat{\mu}_{y_i}|}{N_d}$, RMSE is $\sqrt{\frac{\sum_i^{N_d} (y_i - \hat{\mu}_{y_i})^2}{N_d}}$, and $R^2$ is $1 - \frac{\sum_i^{N_d} (y_i - \hat{\mu}_{y_i})^2}{\sum_i^{N_d} (y_i - \overline{y})^2}$.

We also used the Negative Log-Likelihood (NLL) to evaluate the fitting performance of the model that incorporates uncertainty and to assess the reasonableness of its output variance. The specific calculation method for NLL is as follows:

$$NLL = \sum_i^{N_D} \left[ \frac{(y_i - \mu_{y_i})^2}{2\sigma_{y_i}^2} + \frac{1}{2} \log(2\pi\sigma_{y_i}^2) \right], \tag{15}$$

where we call the first term $\frac{(y_i - \mu_{y_i})^2}{2\sigma_{y_i}^2}$ as Calibration, and the second term $\frac{1}{2} \log(2\pi\sigma_{y_i}^2)$ as Tolerance. Additionally, follow (Schwaller et al., 2021b; Kwon et al., 2022b), we also evaluate the standard deviation of the above metrics under ten repetitions of the experiment.

H.5.2. DETAILED RESULTS

**Standard Deviations and Cross Validation Results.** Following Schwaller et al. (2021b), we additionally test the MAE and RMSE, as well as the standard deviation of the results. We also introduce more baselines. Specifically, on HTE datasets, we introduce traditional methods such as DFT and MFF. We also compare the performance of the model using one-hot labels of molecules (Onehot). Additionally, we include the performance of T5Chem, as well as the SOTA model RMVP based on large-scale data pre-training. On USPTO-Yield, we introduce HRP as our extra baselines. Since we cannot determine whether T5Chem uses average or optimal results, we exclude it from comparison.

*Table 47.* Detailed HTE yield prediction results with more comparison methods, using MAE, RMSE and $R^2$ metrics.

| Models | B-H | S-M | Test1 | Test2 | Test3 | Test4 |
|---|---|---|---|---|---|---|
| DFT | - | - | - | - | - | - |
| | - | - | - | - | - | - |
| | 0.92 | - | 0.8 | 0.77 | 0.64 | 0.54 |
| Onehot | - | - | - | - | - | - |
| | - | - | - | - | - | - |
| | 0.89 | - | 0.69 | 0.67 | 0.49 | 0.49 |
| MFF | - | - | - | - | - | - |
| | - | - | - | - | - | - |
| | 0.93 | - | 0.85 | 0.71 | 0.64 | 0.18 |
| Y-B | 3.99±0.15 | 8.13±0.34 | 7.35±0.10 | 7.27±0.72 | 9.13±0.75 | 13.67±1.07 |
| | 6.01±0.27 | 12.07±0.46 | 11.44±0.34 | 11.14±1.27 | 14.28±0.82 | 19.68±1.40 |
| | 0.95±0.01 | 0.82±0.01 | **0.84±0.01** | 0.84±0.03 | 0.75±0.04 | 0.49±0.05 |
| Y-B-A | 3.09±0.12 | 6.60±0.27 | 7.02±0.76 | 6.59±0.33 | 11.05±0.95 | 18.42±0.62 |
| | 4.80±0.26 | 10.52±0.48 | 11.76±1.40 | 9.89±0.74 | 18.04±1.40 | 24.28±0.49 |
| | **0.97±0.01** | 0.86±0.01 | 0.81±0.05 | 0.87±0.02 | 0.59±0.07 | 0.16±0.03 |
| DRFP | 4.03±0.13 | 7.00±0.20 | 8.16±0.07 | 7.69±0.10 | 8.92±0.06 | 12.42±0.07 |
| | 6.08±0.28 | 11.00±0.40 | 11.99±0.10 | 11.26±0.16 | 15.04±0.06 | 18.76±0.11 |
| | 0.95±0.01 | 0.85±0.01 | 0.81±0.01 | 0.83±0.00 | 0.71±0.01 | 0.49±0.00 |
| T5 | - | - | - | - | - | - |
| | - | - | - | - | - | - |
| | 0.97 | 0.86 | 0.81 | 0.91 | 0.79 | 0.63 |
| Egret | 4.47±0.23 | 7.00±0.20 | **6.97±0.47** | 6.31±0.37 | 10.40±0.71 | 12.37±0.83 |
| | 6.61±0.30 | 11.00±0.40 | 11.03±0.64 | 9.41±0.98 | 16.58±1.33 | 17.88±0.99 |
| | 0.94±0.01 | 0.85±0.01 | **0.84±0.01** | 0.88±0.03 | 0.65±0.06 | 0.54±0.06 |
| UGNN | **2.92±0.06** | 6.12±0.22 | 8.08±0.83 | 6.30±0.65 | 8.99±0.31 | 13.19±0.75 |
| | **4.43±0.09** | 9.47±0.46 | 13.75±1.18 | 9.48±1.03 | 14.94±0.62 | 18.77±0.57 |
| | **0.97±0.01** | **0.89±0.01** | 0.74±0.04 | 0.88±0.03 | 0.72±0.02 | 0.50±0.03 |
| RMVP | 5.13±0.23 | 7.37±0.21 | 9.60±0.69 | 8.02±1.23 | 10.74±0.72 | 13.59±1.28 |
| | 7.52±0.49 | 10.79±0.25 | 13.33±0.63 | 11.18±1.54 | 15.38±1.20 | 18.16±1.40 |
| | 0.92±0.01 | 0.85±0.01 | 0.76±0.02 | 0.83±0.05 | 0.70±0.05 | 0.53±0.08 |
| RMVP (Pretrained) | 3.11±0.07 | 6.59±0.20 | 7.28±0.12 | **6.08±0.15** | 8.97±0.49 | **10.61±0.66** |
| | 4.63±0.14 | 10.37±0.42 | **10.77±0.14** | **8.72±0.18** | **12.79±0.77** | **14.62±0.93** |
| | **0.97±0.01** | 0.86±0.01 | **0.84±0.01** | **0.90±0.01** | **0.79±0.03** | **0.69±0.04** |
| D-MPNN | 4.72±0.08 | 7.96±0.21 | 8.20±0.28 | 7.81±0.81 | 9.04±0.26 | 12.25±0.23 |
| | 6.41±0.15 | 10.84±0.44 | 12.09±1.00 | 11.21±1.10 | 14.46±0.60 | 17.76±0.63 |
| | 0.94±0.01 | 0.85±0.01 | 0.80±0.03 | 0.82±0.03 | 0.73±0.02 | 0.55±0.03 |
| Rxn Hypergraph | 3.44±0.04 | 7.83±0.31 | 8.44±0.30 | 7.20±0.52 | 9.01±0.21 | 11.62±0.85 |
| | 5.45±0.08 | 11.06±0.46 | 11.65±0.49 | 11.18±0.60 | 14.98±0.48 | 17.65±0.42 |
| | 0.96±0.01 | 0.85±0.01 | 0.81±0.01 | 0.83±0.02 | 0.71±0.02 | 0.56±0.02 |
| RG | 3.07±0.06 | **6.08±0.26** | 7.84±0.20 | 6.23±0.45 | **8.64±0.35** | 10.81±1.37 |
| | 4.64±0.09 | **9.32±0.47** | 12.05±0.30 | 9.20±0.71 | 13.50±0.61 | 15.05±1.40 |
| | **0.97±0.01** | **0.89±0.01** | 0.80±0.01 | 0.88±0.02 | 0.76±0.02 | 0.68±0.06 |

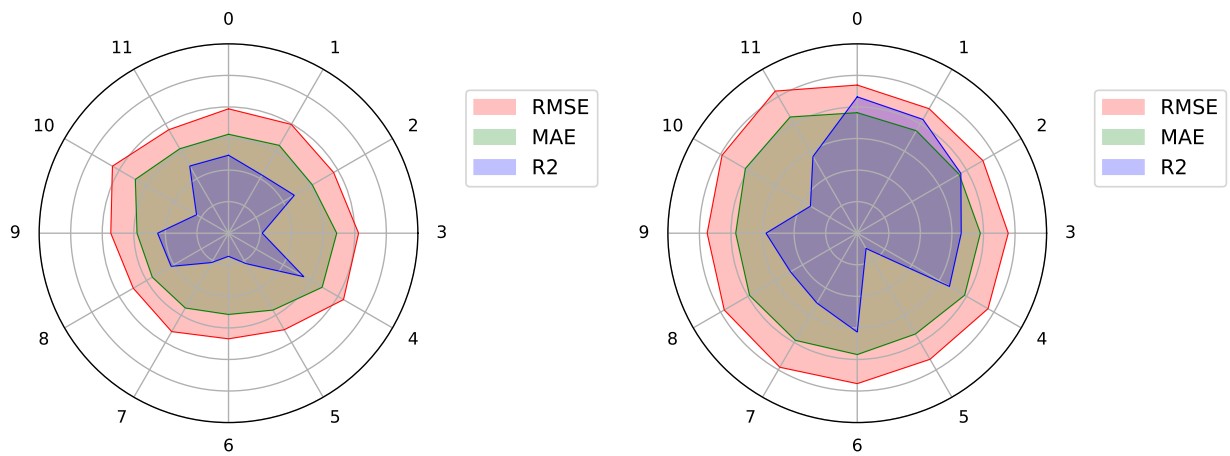

*Figure 20.* Radar chart of yield prediction metrics under various reaction types. The results show a strong correlation between the accuracy of yield prediction and reaction types in the USPTO-Yield.

Based on the results shown in Tab. 47, RG remains highly competitive. Similarly, as shown in Tab. 48, on USPTO-Yield, RG remains at a leading level. Compared with non-pretrained models and methods, whether based on neural networks or theoretical calculations, most of our performance is at the forefront. Compared to RMVP pre-trained on large-scale datasets, we achieve similar $R^2$ scores by using only the original training samples of over two thousand. However, on the more challenging Test4 dataset, our model exhibits significant variance. In ten repeated experiments, RG achieves a maximum $R^2$ score of $0.77$, while the minimum $R^2$ score is only $0.58$. This is due to the small dataset scale and the structure of GNNs. For simpler fingerprint-based models, they have smaller training variance than GNN-based or Transformer-based methods.

*Table 48.* Regression accuracy $(R^2)$ on USPTO-Yield dataset.

| Model | Gram | Subgram |
|---|---|---|
| DRFP | **0.130** | 0.197 |
| Yield-Bert | 0.117 | 0.195 |
| T5Chem | 0.116 | 0.202 |
| Egret | 0.128 | 0.206 |
| UGNN | 0.117 | 0.190 |
| HRP | 0.129 | 0.200 |
| D-MPNN | 0.125 | 0.202 |
| Rxn Hypergraph | 0.118 | 0.196 |
| RG | 0.129 | **0.216** |

*Table 49.* Results for statistical significance analysis of Reaction Graph on reaction yield prediction task. The baseline method is UGNN.

| Metrics | BH | BH1 | BH2 | BH3 | BH4 | SM | Gram | Subgram |
|---|---|---|---|---|---|---|---|---|
| $-logP$ | 0.69 | 8.74 | 0.69 | 5.52 | 6.69 | 0.69 | 3.27 | 7.84 |

*Table 50.* Confidence interval results ([Min,Max]) for reaction yield prediction of Reaction Graph.

| Metrics | Bound | BH | BH1 | BH2 | BH3 | BH4 | SM | Gram | Subgram |
|---|---|---|---|---|---|---|---|---|---|
| $R^2$ | Min | 0.96 | 0.79 | 0.87 | 0.75 | 0.64 | 0.88 | 0.119 | 0.208 |
| | Max | 0.98 | 0.81 | 0.89 | 0.77 | 0.72 | 0.90 | 0.131 | 0.214 |

**Statistical Significance Tests.** We use UGNN (Kwon et al., 2022b) as the baseline method in the $t$-test, and the results are shown in Tab. 49.

**Confidence Intervals.** We use the settings in Sec. H.2.3 and the result in Tab. 47 to calculate confidence intervals. The results are shown in Tab. 50.

**Reaction Type Analysis.** We analyze the regression performance for different reaction types in the USPTO-Yield dataset, as shown in Fig. 20. We classify the data samples in USPTO-Yield using a classifier trained on Pistachio-Type. The results show a strong correlation between regression performance and reaction type, with noticeable differences in the Gram and Subgram datasets. For (5) Protections and (10) FGA reactions, the performance of both models is poor. However, when comparing $R^2$ metrics with RMSE and MAE, we find that the poor $R^2$ for (5) and (10) are due to the complexity of yield distribution. The yield variances of (5) and (10) are significantly greater than that of other reaction types. Similar situations can also be observed for (3) C-C bond formation and (6) Deprotections on the Gram dataset.

## H.6. Reaction Classification

### H.6.1. METRIC CALCULATION

We used ACC, CEN, MCC and F1 Score as evaluation metrics for reaction classification. In implementation, we used the library functions from PyCM to compute these metrics.

Specifically, we refer to the calculation method in Schwaller et al. (2021a). Let $C$ be the number of classes, and class labels are from 1 to $C$. $M \in [0,1]^{C \times C}$ is the confusion matrix where the element $M_{i,j}$ represents the number of instances that belong to the true class $i$ but are predicted to be in class $j$. $M_{i,j}$ can be calculated using the following formula:

$$M_{i,j} = \sum_{k=1}^{N} \delta(y_k, i) \cdot \delta(\hat{y}_k, j),$$

where $N$ is the total number of samples, $y_k$ is the true label of the $k$-th sample, $\hat{y}_k$ is the predicted label of the $k$-th sample, and $\delta(a, b)$ is the Kronecker delta function defined as:

$$\delta(a,b) = \begin{cases} 1 & \text{if } a = b \\ 0 & \text{if } a \neq b \end{cases}$$

From the confusion matrix, the following metrics can be calculated:

- **Accuracy (ACC/Micro F1).** The accuracy is the ratio of the number of correct predictions to the total number of predictions. It is calculated using the following formula:

$$\text{Accuracy} = \frac{\sum_{i=1}^{C} M_{i,i}}{N}$$

- **Macro F1.** The macro F1 score is the unweighted average of the F1 scores of each class. It is calculated using the following formula:

$$\text{Precision}_j = \frac{M_{j,j}}{\sum_{i=1}^{C} M_{i,j}}$$

$$\text{Recall}_j = \frac{M_{j,j}}{\sum_{i=1}^{C} M_{j,i}}$$

$$F1_j = 2 \cdot \frac{\text{Precision}_j \cdot \text{Recall}_j}{\text{Precision}_j + \text{Recall}_j}$$

- **Confusion Entropy (CEN).** The confusion entropy is a measure of the uncertainty in the confusion matrix. It is calculated using the following formula:

$$P_{i,j}^{j} = \frac{M_{i,j}}{\sum_{k=1}^{C} (M_{j,k} + M_{k,j})}, \quad P_{i,j}^{i} = \frac{M_{i,j}}{\sum_{k=1}^{C} (M_{i,k} + M_{k,i})}$$

$$\text{CEN}_j = -\sum_{k=1, k \neq j}^{C} \left( P_{j,k}^{j} \log_{2(C-1)} \left( P_{j,k}^{j} \right) + P_{k,j}^{j} \log_{2(C-1)} \left( P_{k,j}^{j} \right) \right)$$

$$P_j = \frac{\sum_{k=1}^{C} (M_{j,k} + M_{k,j})}{2 \sum_{k,l=1}^{C} M_{k,l}}$$

$$\text{CEN} = \sum_{j=1}^{C} P_j \text{CEN}_j$$

- **Matthews Correlation Coefficient (MCC).** The Matthews correlation coefficient is usually used in binary classification problems. However, it can be extended to multi-class classification problems using the following formula:

$$\text{cov}(X, Y) = \sum_{i,j,k=1}^{C} \left( M_{i,i} M_{k,j} - M_{j,i} M_{i,k} \right)$$

$$\text{cov}(X, X) = \sum_{i=1}^{C} \left[ \left( \sum_{j=1}^{C} M_{j,i} \right) \left( \sum_{k,l=1,k\neq i}^{C} M_{l,k} \right) \right]$$

$$\text{cov}(Y, Y) = \sum_{i=1}^{C} \left[ \left( \sum_{j=1}^{C} M_{i,j} \right) \left( \sum_{k,l=1,k\neq i}^{C} M_{k,l} \right) \right]$$

$$\text{MCC} = \frac{\text{cov}(X, Y)}{\sqrt{\text{cov}(X, X) \times \text{cov}(Y, Y)}}$$

### H.6.2. DETAILED RESULTS

**Additional Baselines.** we additionally introduce HRP, which also tests the performance on USPTO-TPL, for comparison. The result is shown in Tab. 51. Reaction Graph demonstrates superior performance on all metrics.

*Table 51.* USPTO-TPL results.

| Models | ACC | CEN | MCC |
|---|---|---|---|
| DRFP | 0.977 | 0.011 | 0.977 |
| RXNFP | 0.989 | 0.006 | 0.989 |
| T5Chem | 0.995 | 0.003 | 0.995 |
| HRP | 0.991 | 0.005 | 0.990 |
| Rxn Hypergraph | 0.990 | 0.005 | 0.990 |
| D-MPNN | 0.997 | 0.001 | 0.997 |
| **Reaction Graph** | **0.999** | **0.001** | **0.999** |

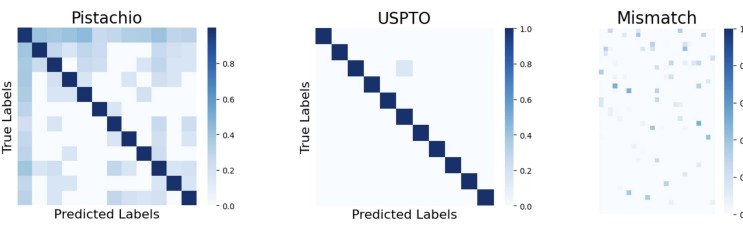

*Figure 21.* Scaled confusion matrices of reaction classification results.

**Standard Deviations.** We calculated the mean and standard deviation using the results from four different random seeds. The results are shown in Tab. 52.

*Table 52.* Results for reaction classification of Reaction Graph on USPTO-TPL and Pistachio-Type dataset, including mean and standard deviation from multiple trials with different random seeds.

| Metrics | USPTO-TPL | | | Pistachio-Type | | |
|---|---|---|---|---|---|---|
| | ACC | CEN | MCC | ACC | CEN | MCC |
| Mean±Std | $0.999 \pm 0.001$ | $0.001 \pm 0.001$ | $0.999 \pm 0.001$ | $0.986 \pm 0.002$ | $0.026 \pm 0.004$ | $0.985 \pm 0.003$ |

**Statistical Significance Tests.** We choose D-MPNN (Heid & Green, 2021) as baseline to calculate the $t$-values. The results are shown in Tab. 53.

*Table 53.* Results for statistical significance analysis of Reaction Graph on reaction classification on USPTO-TPL and Pistachio-Type, using $t$-test method from the mean and standard deviations of multiple trials with different random seeds. The baseline is D-MPNN.

| Metrics | USPTO-TPL | | Pistachio-Type | |
|---|---|---|---|---|
| | ACC | MCC | ACC | MCC |
| $-\log P$ | 4.27 | 4.27 | 4.27 | 3.80 |

**Cross-validation Results.** We conduct cross-validation using the settings in Sec. H.2.3 on USPTO-TPL and Pistachio-Type. The results are shown in Tab. 54.

*Table 54.* Cross-validation results for reaction classification of Reaction Graph on USPTO-TPL and Pistachio-Type dataset, including mean and standard deviation of ACC, MCC and CEN metrics from multiple trials with different train/test splits.

| Metrics | USPTO-TPL | | | Pistachio-Type | | |
|---|---|---|---|---|---|---|
| | ACC | CEN | MCC | ACC | CEN | MCC |
| Mean±Std | $0.999 \pm 0.001$ | $0.001 \pm 0.001$ | $0.999 \pm 0.001$ | $0.987 \pm 0.002$ | $0.025 \pm 0.003$ | $0.986 \pm 0.002$ |

**Confidence Intervals.** We use the same settings in Sec. H.2.3 on the USPTO-TPL and Pistachio-Type to calculate confidence intervals. The results are shown in Tab. 55.

*Table 55.* Confidence interval results ([Min,Max]) for reaction classification of Reaction Graph on USPTO-TPL and Pistachio-Type dataset, calculated using normal distribution method from mean and standard deviations of multiple trials with different random seeds.

| Metrics | Type | USPTO-TPL | | | Pistachio-Type | | |
|---|---|---|---|---|---|---|---|
| | | ACC | CEN | MCC | ACC | CEN | MCC |
| Mean±Std | Min | 0.998 | 0.000 | 0.998 | 0.984 | 0.022 | 0.982 |
| | Max | 1.000 | 0.002 | 1.000 | 0.988 | 0.030 | 0.988 |

**Confusion Matrix Analysis.** For USPTO-TPL, we map the 1,000 templates into 12 reaction types of Pistachio, making it easier to assess the model's performance. We first use the reaction classification model trained on Pistachio-Type to predict the types of chemical reactions in USPTO-TPL. Then, we count the frequency of each Pistachio type corresponding to each USPTO-TPL type, selecting the most frequent as the mapped type.

The results are shown in Fig. 21. The reaction type mapping results are shown in the bitmap on the right of the figure, which contains 1,000 points. Each pixel representing a USPTO-TPL template, with values indicating the proportion of predicted types other than the most frequent type. The misclassification rate for all templates does not exceed 0.5, and most of them have misclassification rates close to 0, demonstrating the validity of our mapping and the generalization of RG.

Since both datasets exhibit high classification accuracy, we perform row normalization and amplification in the confusion matrix, with the amplification function $V' = V^{0.2}$. According to the result, the misclassification rate of $Unknown$ type is significantly higher than other types. This is because the boundaries for $Unknown$ are the most ambiguous.

**Dimensionality Reduction Visualization.** We perform dimensionality reduction visualization on the reaction representation vector $r$. This experiment aim to observe the distribution of chemical reactions, as well as exploring the relationship between $r$ and chemical properties. We use three methods for dimensionality reduction visualization: TMAP, UMAP, and t-SNE.

The result is shown in Fig. 22. Based on the results, we can see a clear correlation between the dimensionality reduction results and reaction types. Additionally, the results indicate a connection between the reaction representation vectors and the number of H donors. This indicates that the reaction representation vectors extracted by the Reaction Graph implicitly contain high-dimensional features of chemical properties.

## H.7. Architecture Module Selection

### H.7.1. EDGE EMBEDDING

We use RBF kernel to calculate edge length embedding. RBF can effectively capture the non-linear relationships between distance and molecular property. This method lifts the scalar edge lengths into a high-dimensional vector that can be more easily utilized by machine learning models, focus more on local structural pattern, and produces smooth mappings which help models to capture variations in continuous data. These advantages make RBF kernel suitable for tasks involving local continuous spatial relationships, such as in molecular structures where edge lengths indicate bond distances.

To demonstrate the efficiency of RBF kernel embedding, we conduct experiments to compare it with different embedding methods, using USPTO-Condition dataset. We use linear projection and discretization embedding as baselines. Specifically, linear projection directly inputs the edge lengths into a fully connected layer to obtain embeddings. Discretization embedding divides the edge lengths into multiple bins to obtain one-hot embeddings. The embeddings are then concatenated with the edge feature vectors. Detailed bin division method is discussed in Sec. D.1.1.

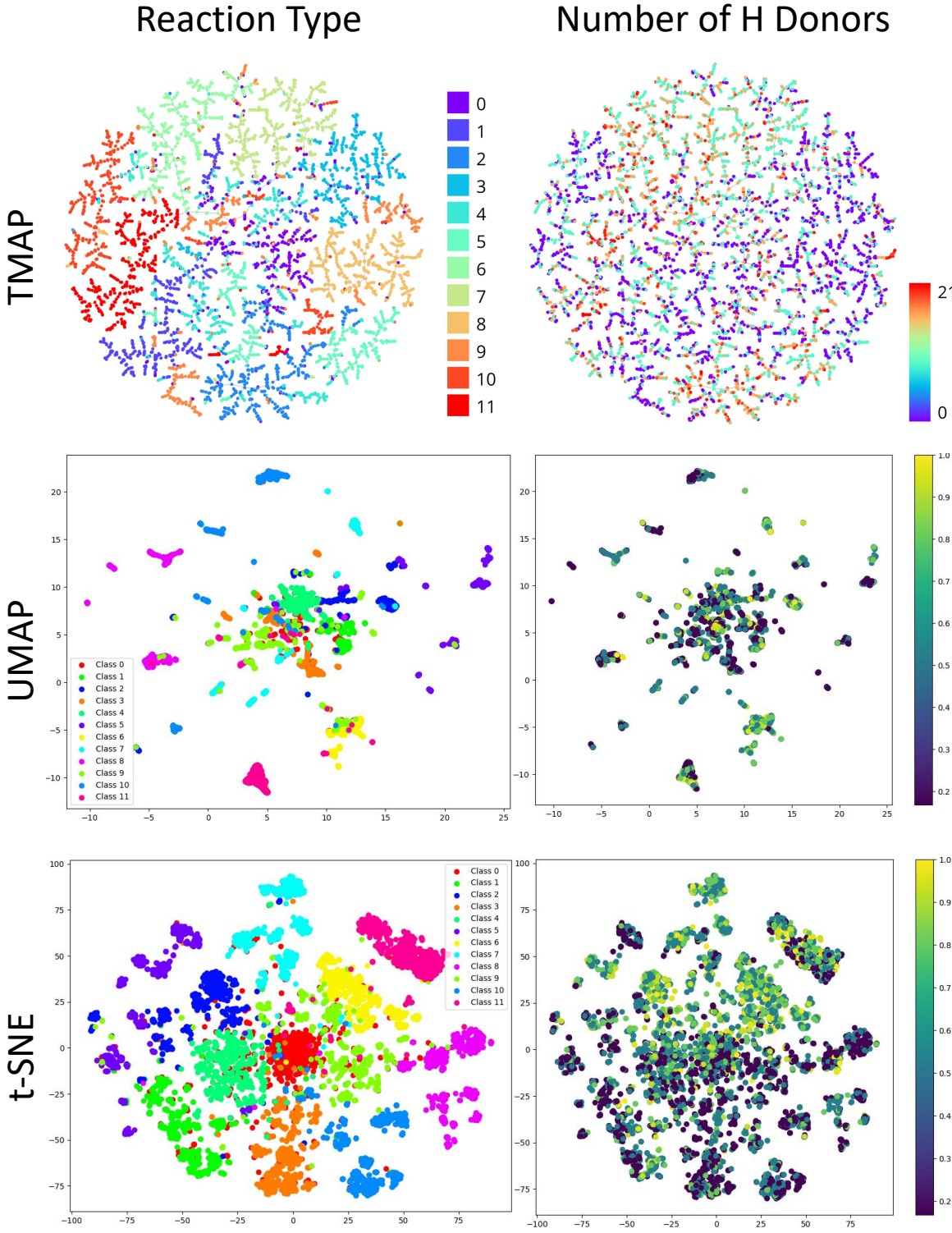

*Figure 22.* The dimensionality reduction visualization results of the reaction representation vectors $r$ extracted from the Reaction Graph. We utilize three unsupervised dimensionality reduction methods: TMAP, UMAP, and t-SNE. The figure shows the relationship between the dimensionality reduction results and reaction types, as well as the number of H donors.

*Table 56.* Influence of different edge length embedding methods on model's performance, using USPTO-Condition dataset.

| Method | Top-1↑ | Top-3↑ | Top-5↑ | Top-10↑ | Top-15↑ |
|---|---|---|---|---|---|
| Linear Projection Embedding | 0.3173 | 0.4303 | 0.4684 | 0.5048 | 0.5164 |
| Discretization Embedding | 0.3101 | 0.4201 | 0.4569 | 0.4926 | 0.5046 |
| RBF kernel Embedding (ours) | **0.3246** | **0.4343** | **0.4715** | **0.5061** | **0.5181** |

As shown in the Tab. 56 , RBF kernel outperforms other embedding methods, demonstrating its efficiency. Directly using linear mapping may cause the distance feature to lose its non-linearity. Meanwhile, using discrete features can easily lead to information loss.

### H.7.2. VERTEX-EDGE INTEGRATION

We conduct experiment to tested different vertex-edge integration methods to demonstrate the efficiency of our chosen approach. The baselines include Bond Vector Message Passing Model following PaiNN (Schütt et al., 2021), the Bond Angle Message Passing Model following DimeNet (Gasteiger et al., 2020), the 3D Graph Transformer following UniMol (Zhou et al., 2023), EGAT (Monninger et al., 2023) and GINE (Hu et al., 2019). Our method follows UGNN (Kwon et al., 2022b) and MPNN (Gilmer et al., 2017)

*Table 57.* Influence of different vertex-edge integration methods on condition prediction task, using USPTO-Condition dataset.

| Method | Top-1↑ | Top-3↑ | Top-5↑ | Top-10↑ | Top-15↑ |
|---|---|---|---|---|---|
| Bond Vector Model | 0.290 | 0.403 | 0.439 | 0.475 | 0.488 |
| Bond Angle Model | 0.318 | 0.429 | 0.466 | 0.502 | 0.514 |
| 3D Graph Transformer | 0.300 | 0.405 | 0.440 | 0.469 | 0.477 |
| EGAT | 0.304 | 0.417 | 0.453 | 0.490 | 0.502 |
| GINE | 0.299 | 0.406 | 0.441 | 0.475 | 0.487 |
| Ours | **0.325** | **0.434** | **0.472** | **0.506** | **0.518** |

The results in Tab. 57 show that our method is superior to other methods, demonstrating its efficiency.

### H.7.3. AGGREGATION

**Influence of Attention Mechanism and LSTM.** Effectively identifying and leveraging the chemical reaction mechanism to understand and reason about reactions is challenging. In our work, we use an attention mechanism to adaptively capture the most important cues for reaction modeling. However, since these cues are not always easy to identify in one time, we employ an LSTM to progressively and interactively discover them. As shown in Fig. 3, the attention-based aggregation module with LSTM accurately locates the reaction center on Reaction Graph.

We conduct experiments to demonstrate the roles of attention and LSTM. Specifically, we compare the performance of our method with the aggregation module without using attention and LSTM. We use USPTO-Condition dataset to evaluate the model's performance. For aggregation module without both attention and LSTM, we use SumPooling. For the attention aggregation module without LSTM, we set the number of iterations for Set2Set to 1, which is equivalent to not using LSTM.

*Table 58.* Influence of attention mechanism and LSTM on model performance, using USPTO-Condition dataset.

| Method | Top-1↑ | Top-3↑ | Top-5↑ | Top-10↑ | Top-15↑ |
|---|---|---|---|---|---|
| w/o Attention & w/o LSTM | 0.3159 | 0.4276 | 0.4642 | 0.4983 | 0.5110 |
| w/ Attention & w/o LSTM | 0.3187 | 0.4303 | 0.4670 | 0.5018 | 0.5136 |
| w/ Attention & w/ LSTM | **0.3246** | **0.4343** | **0.4715** | **0.5061** | **0.5181** |

The result in the table above shows that the attention mechanism and LSTM contributes to the performance.

**Set2Set vs. Set Transformer.** We use Set2Set to capture the global representation of a Reaction Graph by aggregating node features. We compare the ability of Set2Set and Set Transformer (Lee et al., 2019) for capturing the global representation of a Reaction Graph. The detailed implementation of Set Transformer can be found in Sec. D.

*Table 59.* Influence of Set Transformer and Set2Set on condition prediction result and inference time, using USPTO-Condition dataset.

| Method | Top-1↑ | Top-3↑ | Top-5↑ | Top-10↑ | Top-15↑ | Inference Time↓ |
|---|---|---|---|---|---|---|
| Set Transformer | 0.2940 | 0.4079 | 0.4471 | 0.4847 | 0.4968 | 6min 1s |
| Set2Set | **0.3246** | **0.4343** | **0.4715** | **0.5061** | **0.5181** | **2min 36s** |

The results indicate that, when used as an aggregation module, Set2Set surpasses the Set Transformer by 5% to 10% in performance and is also more computationally efficient.

We further analyze the reasons for the unsatisfactory performance of the Set Transformer in the reaction property prediction task. We summarize two points:

1. The set message passing in the Set Transformer disrupts the topological constraints of the graph, leading to the loss of structural information.

2. The seed vector in the Set Transformer is fixed, whereas the adaptive seed vector derived from the graph in Set2Set is more advantageous.

We demonstrate these points through experiments. The result is shown in Tab. 60. Specifically, we gradually reduce the number of layers in the Set Transformer Encoder to observe whether the set message passing has a negative effect on graph information extraction. Additionally, We compare the performance of the Set Transformer Decoder (without adaptive seed vector) and the Set2Set(with adaptive seed cector) to observe whether the adaptive seed vector contributes to the model's performance. We use a subset from USPTO-Condition for training efficiency.

*Table 60.* The influence of set message passing and adaptive seed vector on model's performance. Using a $1/8$ subset of USPTO-Condition for training efficiency.

| Method | Set Message Passing | Adaptive Seed Vector | ACC↑ |
|---|---|---|---|
| Set Transformer | 2 | ✗ | 0.1597 |
| Set Transformer | 1 | ✗ | 0.1634 |
| Set Transformer | 0 | ✗ | 0.1700 |
| Set2Set (ours) | 0 | ✓ | **0.1773** |

According to the results, with the set message passing reduces, the performance gradually improves. This result indicates that set message passing exhibits a negative effect when extracting features from the Reaction Graph. Meanwhile, adaptive seed vector achieves better performance than directly using a fixed seed vector, validating its effectiveness.

## H.8. Graph Representation

### H.8.1. 3D INFORMATION

**Different 3D Representations.** We demonstrate the efficiency of our 3D representation design through experiments. Reaction Graph includes bond length and bond angle information, where bond angle is implicitly conveyed by angular edge. We compare Reaction Graph with methods that use explicit bond angle, pairwise distance, atomic coordinates, equivariant neural networks, and torsion angles.

The method using explicit bond angle is implemented by directional message passing module from DimeNet (Gasteiger et al., 2020). Pairwise distance is implemented by UniMol (Zhou et al., 2023). The atomic coordinate method is implemented by concatenating the atomic XYZ coordinate with the atomic attributes. The equivariant neural network is implemented by replacing the length information in Reaction Graph with vector, and use PaiNN (Schütt et al., 2021) as our vertex-edge integration module. The torsion angle is implemented by extending the angular edge in Reaction Graph to torsion angular edge, and details can be found in Sec. D and Sec. E.

*Table 61.* Influence of different 3D representation methods on Reaction Graph in the task of predicting reaction conditions, using USPTO-Condition dataset.

| Method | Top-1↑ | Top-3↑ | Top-5↑ | Top-10↑ | Top-15↑ |
|---|---|---|---|---|---|
| Without 3D Information | 0.3133 | 0.4248 | 0.4613 | 0.4961 | 0.5094 |
| Bond Length | 0.3165 | 0.4251 | 0.4616 | 0.4971 | 0.5090 |
| Bond Length + Explicit Bond Angle | 0.3179 | 0.4290 | 0.4656 | 0.5018 | 0.5146 |
| Pairwise Distance | 0.2955 | 0.4054 | 0.4397 | 0.4689 | 0.4766 |
| Atom Coordinate | 0.3123 | 0.4243 | 0.4628 | 0.4987 | 0.5111 |
| Equivariant Neural Networks | 0.2899 | 0.4026 | 0.4390 | 0.4749 | 0.4879 |
| Bond Length + Torsion Angular Edge | 0.3022 | 0.4087 | 0.4467 | 0.4821 | 0.4935 |
| Bond Length + Angular Edge (ours) | **0.3246** | **0.4343** | **0.4715** | **0.5061** | **0.5181** |

As shown in the Tab. 61, Bond Length + Angular Edge achieves the best performance. This demonstrates the effectiveness of our approach.

**3D Information on Angular Edge.** We recognize that the role of the angular edge may not only be to provide angular information but also to serve as a shortcut for message passing. Therefore, we conduct an additional ablation experiments.

To analyze the role of the angular edge as a 3D prior, we set it's length to 0. This approach retains the connection of the angular edge to two-hop neighbors, thus also serving as a shortcut. To analyze the effect of offering a shortcut by angular edge, we remove angular edge from Reaction Graph.

*Table 62.* Impact of connection and length (for 3D modeling) on Angular Edge. Angular Edge is mainly used for providing 3D information, instead of providing shortcut for message passing.

| Method | Top-1↑ | Top-3↑ | Top-5↑ | Top-10↑ | Top-15↑ |
|---|---|---|---|---|---|
| w/o connection & w/o length for 3D modeling | 0.3165 | 0.4251 | 0.4616 | 0.4971 | 0.5090 |
| w/ connection & w/o length for 3D modeling | 0.3150 | 0.4259 | 0.4637 | 0.4991 | 0.5107 |
| w/ connection & w/ length for 3D modeling (ours) | **0.3246** | **0.4343** | **0.4715** | **0.5061** | **0.5181** |

The experimental results are shown in the Tab. 62. Based on the results, for angular edge, the role of being a shortcut is far less significant than its provision of 3D priors. This also demonstrates that angular edge effectively provides 3D bond angle information.

**Adding 3D Information to Other Graphs.** To explore whether 3D information can be effective in other methods, we attempt to incorporate bond length information into the D-MPNN and Rxn Hypergraph. The detailed implementation can be found in Sec. D.1.3 and Sec. D.1.2.

We test the reaction condition prediction performance of D-MPNN and Rxn Hypergraph on USPTO-Condition before and after adding 3D information, and the results are shown in Tab. 63.

*Table 63.* Influence of 3D bond length information on D-MPNN and Rxn Hypergraph's performance in reaction condition prediction task, using USPTO-Condition dataset.

| Method | 3D Info. | Top-1↑ | Top-3↑ | Top-5↑ | Top-10↑ | Top-15↑ |
|---|---|---|---|---|---|---|
| D-MPNN | w/o 3D | 0.1977 | 0.3000 | 0.3341 | 0.3780 | 0.3924 |
| D-MPNN | w 3D | **0.2030** | **0.3059** | **0.3410** | **0.3830** | **0.3971** |
| Rxn Hypergraph | w/o 3D | 0.2127 | 0.3084 | 0.3447 | 0.3808 | 0.3927 |
| Rxn Hypergraph | w/ 3D | **0.2149** | **0.3113** | **0.3464** | **0.3825** | **0.3949** |

According to the result, the incorporation of 3D information effectively improved the top-$k$ performance of D-MPNN and Rxn Hypergraph. The average top-$k$ performance is increased by 1.8% and 0.7% for D-MPNN and Rxn Hypergraph relatively, and the Top-1 accuracy is increased by 2.7% for D-MPNN, demonstrating the efficiency of 3D structural priors in reaction property prediction tasks.

**3D Information Accuracy.** The accuracy of 3D information affects model performance. To evaluate the impact of 3D accuracy, instead of using a more accurate method (as computational efficiency of DFT is too low), we use an alternative approach. Specifically, we reduce the 3D accuracy by adding normal noise to original MMFF calculation results. The larger the noise, the lower the accuracy. The results are shown in Tab. 64. The results show that the accuracy of 3D information does affect the performance of the model. As the noise level increases, the model's performance gradually declines.

*Table 64.* The influence of 3D accuracy on model performance in condition prediction task. The noise level reflects the accuracy of 3D information. The smaller the noise, the higher the 3D accuracy. We use 1/8 of the USPTO-Condition dataset.

| Noise Level | Top-1 | Top-3 | Top-5 | Top-10 | Top-15 |
|---|---|---|---|---|---|
| 0.0 | **0.1773** | **0.2749** | **0.3146** | **0.3534** | **0.3670** |
| 0.05 | 0.1738 | 0.2650 | 0.3044 | 0.3443 | 0.3580 |
| 0.1 | 0.1695 | 0.2633 | 0.2990 | 0.3420 | 0.3570 |
| 0.2 | 0.1635 | 0.2643 | 0.2986 | 0.3397 | 0.3556 |
| 0.4 | 0.1518 | 0.2510 | 0.2940 | 0.3346 | 0.3486 |

**Impact of Label Quality on 3D.** We investigate the relationship between label quality and 3D model performance. We use the degree of label sparsity as a measure of label quality. Specifically, we separate Pistachio-Condition into multiple scaffolds of different sizes. The size of scaffold can reflect the degree of label sparsity. Specifically, the test set is consistent across all scaffolds. From the first to the sixth group, the training set size halves each time, reducing from full size to $1/32$ of the original.

*Table 65.* The influence of label sparsity on model performance in condition prediction task. The degree of label sparsity can be reflected by the size of the scaffold.

| Scaffold Ratio | 1 | 1/2 | 1/4 | 1/8 | 1/16 | 1/32 |
|---|---|---|---|---|---|---|
| **w/o 3D** | 0.3852 | 0.3503 | 0.3116 | 0.2716 | 0.2210 | **0.1907** |
| **w 3D** | **0.3915** | **0.3550** | **0.3146** | **0.2768** | **0.2244** | 0.1857 |

The result is shown in Tab. 65. The model's performance decreases with the increase of label sparsity. In the smallest scaffold, the effect of 3D information is affected by label sparsity. Combining the results of 3D accuracy and label sparsity, label quality has greater impact on performance than 3D accuracy. The quality of the labels acts as a bottleneck, limiting the 3D information to further enhance model performance.

### H.8.2. REACTION INFORMATION

**Impact of Reaction Edge.** We attempt to incorporate the proposed reaction edge into other model architectures. Specifically, we test the Bond Vector Model, Bond Angle Model, EGAT and GINE.

*Table 66.* Influence of the proposed reaction information (reaction edge) on the accuracies of different methods, on USPTO-Condition dataset.

| Model Architecture | Reaction Information | Top-1↑ | Top-3↑ | Top-5↑ | Top-10↑ | Top-15↑ |
|---|---|---|---|---|---|---|
| Bond Vector Model (Schütt et al., 2021) | ✗ | 0.282 | 0.393 | 0.432 | 0.468 | 0.481 |
| | ✓ | **0.290** | **0.403** | **0.439** | **0.475** | **0.488** |
| Bond Angle Model (Gasteiger et al., 2020) | ✗ | 0.307 | 0.420 | 0.456 | 0.493 | 0.505 |
| | ✓ | **0.318** | **0.429** | **0.466** | **0.502** | **0.514** |
| EGAT (Monninger et al., 2023) | ✗ | 0.297 | 0.406 | 0.442 | 0.478 | 0.489 |
| | ✓ | **0.304** | **0.417** | **0.453** | **0.490** | **0.502** |
| GINE (Hu et al., 2019) | ✗ | 0.289 | 0.396 | 0.432 | 0.468 | 0.481 |
| | ✓ | **0.299** | **0.406** | **0.441** | **0.475** | **0.487** |

The results in Tab. 66 show that reaction information improves the top-$k$ accuracy of all methods. This demonstrates the broad effectiveness of integrating reaction information.

**Rxn Hypergraph with Reaction Hypernode**

To further demonstrate the effectiveness of reaction information, we incorporate reaction information into other graph representations. Specifically, we can model the interactions between reactants and products by adding another hypernode in the Rxn Hypergraph. Details can be found in Sec. D.

*Table 67.* Influence of an additional Reaction Hypernode in Rxn Hypergraph across various tasks.

| Method | Cond (T1↑) | | Yield (R2↑) | | | | | | | | Type (ACC↑) | |
|---|---|---|---|---|---|---|---|---|---|---|---|---|
| | U-C | P-C | BH | BH1 | BH2 | BH3 | BH4 | SM | Gram | Subgram | U-T | P-T |
| w/o Hypernode | 0.213 | 0.288 | 0.96 | 0.81 | 0.83 | 0.71 | 0.56 | 0.85 | 0.118 | 0.196 | 0.954 | 0.911 |
| with Hypernode | 0.211 | 0.289 | 0.96 | **0.82** | 0.81 | 0.75 | 0.57 | 0.86 | 0.112 | 0.187 | 0.984 | 0.936 |
| Reaction Graph (ours) | **0.324** | **0.392** | **0.97** | 0.80 | **0.88** | **0.76** | **0.68** | **0.89** | **0.129** | **0.216** | **0.999** | **0.987** |

To test the performance of this idea, we conduct experiments on various tasks using the hypernode method. The results are presented in Tab. 67.

According to the result, the hypernode method shows improvement in reaction classification by increasing the accuracy on USPTO-TPL by 3% and that of Pistachio-Type by 2.5%, while it has a slight impact on condition and yield prediction. This may be because reaction classification is relatively intuitive, while condition and yield prediction are complex. To improve the performance of condition and yield prediction, more accurate interaction modeling (e.g., Reaction Graph) is needed. Reaction Graph surpasses the performance of the hypernode method, demonstrating its efficiency in interaction modeling.

## I. Toolkits

We use a series of tools for chemical property calculations, data analysis, and Reaction Rraph construction. We have conducted a brief comparison and summary of these tools, and the results are in Tab. 68.

### I.1. Calculate/Optimize 3D Conformation

ETKDG, UFF, MMFF, and DFT are algorithms for calculating or optimizing 3D conformations. ETKDG and MMFF are accurate for small organic molecules but less effective for larger ones and metal complexes. UFF is more suitable for handling metal complexes. DFT, based on quantum chemistry, provides high precision for various compounds but is inefficient. In this paper, we use large real-world chemical databases, USPTO and Pistachio, with total millions of entries and complex molecular distributions. We find that even with the simplest basis sets, DFT is too time-consuming. Hence, we initialize conformations using ETKDG and then optimize them with MMFF94. For molecules that MMFF94 cannot handle (e.g., involving heavy metals), we use UFF for conformation optimization.

Chemical toolkits such as RDKit and OpenBabel provide implementations of the MMFF and UFF algorithms, with RDKit offering an easy way to initialize conformations using ETKDG. As for calculations like DFT, chemical toolkits such as Psi4 and PySCF provide relevant functionalities. These tools all offer interfaces in Python.

### I.2. Predict Atom Mapping

RXNMapper and NameRXN are tools for atomic mapping. RXNMapper is open-source and based on an unsupervised language model, capable of providing efficient and accurate predictions. NameRXN is a commercial rule-based tool, offering highly reliable results, though some reactions fall outside its rule coverage. After evaluating the mapping performance on the USPTO dataset, we choose RXNMapper because it demonstrates high stability and most of its predictions are accurate.

### I.3. Classify Reaction

NameRXN is also a tool for reaction classification. In this paper, the labels of the Pistachio dataset are annotated using NameRXN.

*Table 68.* Comparison between Data Preprocessing Tools, including functionality, precision, time efficiency, application scope, and limitations.

| Tools | Function | Precision | Efficiency | Application Scope | Limitation |
|---|---|---|---|---|---|
| ETKDG | Calculate/Optimize 3D Conformation | Low | High | Small Organic Molecules | Fail on Some Metal-Complex |
| UFF | Calculate/Optimize 3D Conformation | Relatively Low | High | Universal | Low Accuracy |
| MMFF | Calculate/Optimize 3D Conformation | Medium | High | Small Organic Molecules | Fail on Some Metal-Complex |
| DFT | Calculate/Optimize 3D Conformation | High | Low | Depend on Basis Set | Slow |
| RXNMapper | Predict Atom Mapping | Relatively High | High | Small Molecules | Lack of Rule Constraints |
| NameRXN | Predict Atom Mapping + Classify Reaction | High | High | Limited by Manual Rule | Manual Rule Based |

## J. Terminology List

We explain the methods mentioned in the paper in the Sec. B and Sec. C, and here we provide additional explanations for some remaining concepts.

1. **Graph Neural Network (GNN).** A type of neural network designed to process graph data.

2. **Message Passing (MP).** A mechanism in GNNs where nodes exchange information to update their representations based on neighboring nodes.

3. **Molecule.** A group of atoms bonded together, representing the smallest fundamental unit of a chemical compound.

4. **Edge.** A connection between two nodes in a graph, representing a relationship or interaction.

5. **Node/Vertex.** A fundamental unit in a graph representing an entity, such as an atom in a molecular graph.

6. **Atom.** The basic unit of a chemical element, consisting of protons, neutrons, and electrons.

7. **Bond.** A lasting attraction between atoms that enables the formation of chemical compounds.

8. **Bond Length.** The distance between the nuclei of two bonded atoms in a molecule.

9. **Bond Angle.** The angle formed between three atoms, where the central atom is bonded to the other two.

10. **Torsion Angle.** The torsional angle refers to the angle between two adjacent bonds in a molecule.

11. **Reaction.** A process in which one or more substances are transformed into different substances.

12. **Retrosynthesis.** The process of deconstructing a complex molecule into simpler precursors to design synthetic pathways.

13. **Catalyst.** A substance that accelerates a chemical reaction without being consumed in the process.

14. **Solvent.** A substance, typically a liquid, in which solutes are dissolved to form a solution.

15. **Reagent.** A substance used in a chemical reaction to detect, measure, or produce other substances.

16. **Computational Chemistry.** The use of computer simulations to solve chemical problems and predict molecular behavior.

17. **Atom Mapping.** The correspondence of atoms between reactants and products.

18. **High-throughput Experiment (HTE).** A method that allows rapid testing of multiple conditions or compounds simultaneously to accelerate research.

19. **Conformation.** The 3D shape of a molecule resulting from the rotation around its single bonds.

20. **Bond Edge.** Represents an edge in the reaction graph that corresponds to a chemical bond, distinguished from Reaction Edge and Angular Edge.

21. **Transition State (TS).** The transition state is a high-energy, unstable arrangement of atoms that occurs during a chemical reaction, representing the point at which reactants are transformed into products.

