# OpenReview forum: "Reaction Graph: Towards Reaction-Level Modeling for Chemical Reactions with 3D Structures"
_ICML.cc/2025/Conference — ICML 2025 poster_

### Official Review · Reviewer_fygo · 2025-02-28

**Overall Recommendation:** 3

**Summary:**

Main Contributions:
- Reaction Graph (RG) Representation: The authors introduce Reaction Graph (RG), a novel unified graph representation for chemical reactions that integrates both reactants and products into a cohesive framework. This representation incorporates 3D molecular structures, which are crucial for accurately modeling chemical reactions.
- Incorporation of Reaction Edges: RG includes reaction edges that connect atoms in reactants to their corresponding atoms in products based on atomic mapping. This allows the model to capture changes in molecular structures during reactions.
- 3D Structure Integration: The authors propose a method to incorporate 3D information into RG using bond lengths and angular edges, which implicitly convey bond angles. This approach is rotationally and translationally invariant.
- Empirical Results: The authors demonstrate the effectiveness of RG through extensive experiments on various tasks, including reaction condition prediction, yield prediction, and reaction classification. RG achieves state-of-the-art accuracy across multiple datasets.

Main Findings:
- The proposed Reaction Graph representation significantly outperforms existing methods in modeling chemical reactions.
- Incorporating 3D structural information and reaction edges enhances the model's ability to understand and predict reaction outcomes.
- The model achieves high accuracy in predicting reaction conditions, yields, and types, showcasing its potential for accelerating drug design and material science.

Main Algorithmic/Conceptual Ideas:
- Unified Graph Representation: RG integrates molecular graphs of reactants and products, capturing interatomic relationships pertinent to the reaction process.
- Reaction Edges: These edges enable the model to exchange information between reactants and products during the message-passing phase of GNNs.
- 3D Information: The use of bond lengths and angular edges provides a simple yet effective way to incorporate 3D molecular structures into the graph representation.
- Attention-based Aggregation: The model uses an attention mechanism combined with an LSTM to aggregate node features into a unified reaction feature vector.

**Claims And Evidence:**

Claims:
- Effectiveness of Reaction Graph: The authors claim that Reaction Graph (RG) is more effective than existing methods in modeling chemical reactions.
- Incorporation of 3D Information: The authors assert that incorporating 3D structural information improves the model's performance.
- Reaction Edges: The introduction of reaction edges helps the model capture changes in molecular structures during reactions.

Evidence:
- The authors provide extensive experimental results on various tasks (reaction condition prediction, yield prediction, and reaction classification) across multiple datasets (USPTO-Condition, Pistachio-Condition, USPTO-Yield, USPTO-TPL, and Pistachio-Type).
- The results show that RG outperforms existing methods, achieving higher accuracy in predicting reaction conditions, yields, and types.
- Ablation studies demonstrate the effectiveness of incorporating 3D information and reaction edges.

Problems:
- The claims are generally well-supported by the experimental results. However, the authors could provide more detailed analysis on the impact of different components (e.g., reaction edges, 3D information) on specific types of reactions or datasets.

**Essential References Not Discussed:**

To my knowledge, Yes, they are

**Experimental Designs Or Analyses:**

Validity:
- The experimental designs are sound and follow standard practices in the field.
- The authors provide detailed descriptions of their methods and datasets, allowing for reproducibility.
- The results are consistent across different tasks and datasets, supporting the robustness of the proposed method.

Issues:
- The authors could provide more detailed error analysis to understand the failure cases and potential limitations of the model.
- The impact of different hyperparameters and training strategies could be further explored to provide a more comprehensive understanding of the model's behavior.

**Methods And Evaluation Criteria:**

The evaluation criteria are standard and widely accepted in the field, making the results comparable to other studies.
The datasets used are large-scale and diverse, providing a comprehensive assessment of the model's performance.

**Other Comments Or Suggestions:**

None

**Other Strengths And Weaknesses:**

### Strengths
- **Originality:** The paper introduces Reaction Graph (RG), a novel unified graph representation for chemical reactions that integrates reactants, products, and reaction edges, along with 3D molecular structures.
- **Effectiveness:** RG achieves state-of-the-art performance across multiple tasks and datasets, demonstrating its superior ability to model chemical reactions.
- **Clarity:** The paper is well-structured, with clear methodology, extensive experiments, and detailed results, making it easy to follow and reproduce.

### Weaknesses
- **Data Quality:** The accuracy of 3D coordinates and reaction labels in the datasets may limit the model's performance and generalization.
- **Error Analysis:** A more detailed analysis of failure cases and error patterns could provide deeper insights into the model's limitations.
- **Computational Efficiency:** The model's inference time and computational requirements could be optimized for real-time applications and larger datasets.

**Questions For Authors:**

None

**Relation To Broader Scientific Literature:**

I think that important and relevant chemical reaction prediction works have been discussed.

**Theoretical Claims:**

The paper does not present any theoretical proofs. The contributions are primarily algorithmic and empirical, focusing on the development and evaluation of the Reaction Graph representation.

---

> ### Author Rebuttal · Authors · 2025-03-31
>
> Thank you for acknowledging that our method is **novel**, **crucial**, and **effective**.
> We also appreciate your valuable suggestions for improving this work.
>
> ## Ⅰ. Impact of Reac and 3D Info in RG
>
> ### 1. Impact on Reaction Types
>
> **Tab 1: Impact of Reac and 3D info on 12 reaction types.**
> Reac|3D|0|1|2|3|4|5|6|7|8|9|10|11
> |-|-|-|-|-|-|-|-|-|-|-|-|-|-
> |||0.292|0.292|0.317|0.323|0.396|0.197|0.347|0.244|0.372|0.312|0.259|0.299
> √||0.292|0.306|0.322|0.33|0.403|0.216|0.359|0.246|0.391|0.327|0.277|0.319
> √|√|0.309|0.318|0.327|0.333|0.424|0.229|0.368|0.264|0.391|0.326|0.298|0.338
>
> 3D is more critical for Unknown reactions (type 0), as their mechanisms are diverse and reaction patterns are difficult to capture. For C–C bond formation (type 3) and Oxidation (type 8), the contribution of Reac info is greater.
>
> ### 2. Impact on Specific Datasets
>
> **Tab 2: Impact of Reac and 3D info on different datasets.**
> |Reac|3D|U-C|P-C|U-T|P-T
> |-|-|-|-|-|-
> |||0.305|0.381|0.992|0.966
> √||0.313|0.385|0.998|0.986
> √|√|0.325|0.392|0.999|0.987
>
> Both 3D and Reac info contribute to condition prediction. For reaction classification, which is directly related to the reaction change, Reac info is more effective.
>
> ## Ⅱ. Error Analysis
>
> ### 1. Condition
> The overall error pattern is in Appendix Fig 18. **A**: US20110105766A1 and **B**: US20040204386A1 are representative failure cases.
> - **Case A.** Correct: PdCl₂ (0.05%) Pred: Pd(PPh₃)₄ (1.82%)
>
>     Conditions with low frequencies (0.05%) may be misclassified, but the Pred still exhibit similarity to the ground truth.
>
> - **Case B.** Correct: CuBr₂ (0.01%<) Pred: None (86.81%)
>
>     For extremely rare conditions (0.01%<), model struggles and classifies them as None. Data augmentation or pretraining are promising solutions for future exploration.
>
> ### 2. Yield
>
> Apart from low label quality, the non-smooth relation between structure and yield is the bottleneck of yield prediction.
>
> We find the top-5 closest reactions of **C** using RXNFP. They have similar structures, yet the yield variance is large (0.5–0.94). This distinction hinders model from correctly predicting **D** and **F**.
>
>     C: CCC(C)(C)N.CCN(C)C.Cl.[Cl-]>>CCC(C)(C)NCCN(C)C
>
> **Tab 3: Correct and Pred yields of 5 similar reactions.**
> |Reaction|Correct|Pred
> |-|-|-
> C|0.60|0.50
> D|0.50|0.68
> E|0.64|0.58
> F|0.94|0.64
> G|0.91|0.83
>
> ### 3. Reaction Classification
>
> The overall error pattern is in Appendix Fig 21. **H** and **I** are representative failure cases.
>
> - **Case H.** Correct: FGI; Pred: Unrecognized
>
>         C1=C[C@H]2[C@H]3C=C[C@H](C3)[C@H]2C1>>C1C=CCC=1
>
>     For rare types of reaction changes, the model may misclassify them as Unknown.
>
> - **Case I.** Correct: Oxidations; Pred: FGI
>
>         Cl.N#CC1CCCC=1N.NO>>N#CC1CCCC=1NO
>
>     In this case, the annotated and predicted types are both acceptable, but the Pred seems more reasonable.
>
> ### 4. Other Tasks
> Detailed analysis is in Sec H1 and H2.
>
> ## Ⅲ. Hyperparameter Selection and Training Strategy
>
> ### Hyperparameter Selection
>
> Results show that the current setting is optimal, while the performance may scale with the hidden dim.
>
> **Tab 4: Impact of hyperparameters.**
> |Hid Dim|MPNN Iter|Pool Iter|T1|T15
> |-|-|-|-|-
> 200|3|2|0.325|0.518
> 50|3|2|0.307|0.505
> 100|3|2|0.315|0.512
> 200|2|2|0.320|0.520
> 200|4|2|0.317|0.514
> 200|3|1|0.319|0.514
> 200|3|3|0.318|0.514
>
> ### Training Strategy
>
> According to **Tab 5**  and Appendix Fig. 19, our proposed 2-stage training strategy effectively addresses the negative transfer problem in multi-task learning like condition prediction.
>
> **Tab 5: Impact of two-stage training strategy.**
> |Method|T1|T3|T5|T10|T15
> |-|-|-|-|-|-
> One-Stage|0.30|0.41|0.45|0.48|0.50
> Two-Stage|0.33|0.43|0.47|0.51|0.52
>
> ## Ⅳ. Data Quality
> ### Accuracy of 3D Coords
>
> We control the 3D accuracy by adding gaussian noise to the bond lengths. Results in **Tab 5** show that when noise level is within a certain margin (5%), the performance decrease is minor (0.03).
>
> **Tab 6: Impact of 3D accuracy on 1/8 USPTO-Condition.**
> Noise|0.0|0.05|0.1|0.2|0.4
> |-|-|-|-|-|-
> Accuracy|0.177|0.174|0.170|0.164|0.152
>
> ### Label Quality
>
> Label density is an important indicator of label quality. We investigate its impact on Pistachio-Condition, where the model suffers from sparse condition annotations. Results in **Tab 6** show that label quality is more critical to model performance than 3D accuracy, and is the primary bottleneck.
>
> **Tab 7: Impact of label density on Pistachio-Condition.**
> Label density | 1/1 | 1/2 | 1/4 |1/8|1/16|1/32
> |-|-|-|-|-|-|-
> Accuracy|0.39|0.36|0.32|0.28|0.22|0.19
>
> ## Ⅴ. Computational Time Analysis
>
> As shown in Appendix F2, for most of the reactions, the construct and inference time of RG are **<50ms**, meeting the real-time requirement. The inference time of large dataset (USPTO test set) are **<3min**, demonstrating the efficiency of RG. We are optimistic that, with the advancement of the conformer prediction, the speed of RG construction will further improve.

---

### Official Review · Reviewer_vEQY · 2025-03-12

**Overall Recommendation:** 3

**Summary:**

The paper introduces a new representation learning framework for chemical reactions. Specifically, the authors propose a graph neural network architecture that takes into account a) explicit inter-reactant/product interactions, and b) the three-dimensional structures of reactants and products. The framework is applied to several reaction-related tasks including reaction classification, yield prediction, and condition recommendation, and is shown to outperform SOTA methods on several benchmarks.

**Claims And Evidence:**

The main claim of this work is that previous reaction representations suffer from two significant limitations: a) They do not adequately capture inter-reactant/product interactions, since they treat all molecules separately and then concatenate their features. B) They ignore three-dimensional structure information. These issues are addressed by the development of a 3D-aware reaction graph model. Experimental evidence (performance metrics on several benchmark tasks) clearly supports the claim of improved predictivity by including interaction and 3D information.

**Essential References Not Discussed:**

The introduction of the paper introduces GNNs for molecular property prediction as a relatively recent development. This is misleading, and some foundational works (Scarselli et al. 2009, Duvenaud et al. 2015) are omitted.

**Experimental Designs Or Analyses:**

Experimental evaluations are done on four main tasks: Leaving group identification, reaction classification, reaction condition prediction, and yield prediction. All tasks are well-established in the literature (including datasets and train–test splits). The authors provide systematic benchmarks against SOTA methods for each of these tasks. The experimental design, including the evaluation metrics, is sound. For each task, the authors additionally conduct an ablation study to dissect the contributions of inter-reactant/product interactions and 3D information, respectively.

**Methods And Evaluation Criteria:**

From my perspective, a number of model design choices remain unclear:
* Why does the model employ separate nodes for the atoms in the reactants and products instead of using a unified node, as would be expected given that these atoms are chemically identical? (similar to the idea of condensed reaction graphs)
* The treatment of non-reactant species (e.g. reagents, catalysts, or solvents) is not explained. Are they included in the graph – which would raise concerns about data leakage in the condition prediction task? Are they excluded from the graph – potentially omitting important information for reaction yield prediction?
* 3D structures of molecules are represented using internal coordinates (i.e. bond/edge lengths and bond angles) rather than Cartesian coordinates commonly used in other works. However, to fully represent the 3D structure of a molecule, bond length and bond angle information is insufficient; and dihedral angles / torsional angles would be required. While this is mentioned in the Supplementary Materials, it is unclear why these features were omitted from the model architecture.

**Other Comments Or Suggestions:**

Organization and clarity of the manuscript could be improved at certain points:
* The current presentation of experiments in Section 3 is somewhat confusing, as the discussion shuffles between focusing on model capabilities and specific tasks. As an example, section 3.2 studies the effects of different 3D featurization techniques using the condition prediction task – which is then re-introduced in section 3.3. A more logical organization would enhance readability of this section.
* Figure 1 would benefit from a clearer structure with better defined divisions and sub-headings.
* The labeling of functional groups in Figure 3 does not conform to standard chemical conventions. The term “carboxyl group” refers to the full COOH unit, and the used labels “carboxyl-hydroxyl” and “carboxyl” are nonsensical from a chemical standpoint.
* In Figure 4 and the accompanying discussion, it is necessary to specify what type of timings are discussed (training time? inference time? timing of generating a 3D structure in the first place?)

**Other Strengths And Weaknesses:**

While the technical discussion appears sound, the discussion of chemical concepts could be improved notably. Examples include:
* While the proposed architecture requires molecular 3D structures, the challenges associated with obtaining such structures are largely ignored. In reality, molecules do not have a dingle 3D structure, but a distribution of 3D structures which is dynamic and depends on the environment. This is inherently hard to capture – and the authors use only equilibrium structures for their model. These structures are generated using a cheap force field method rather than a more accurate, but computationally expensive quantum chemical approach. A discussion of these simplifications and tradeoffs would be valuable.
* The introduction on the applications of AI in chemistry could be notably improved (for a starting point see e.g. a recent review by Cheng et al., Faraday Discuss. 2024). As an example, “analysis of retrosynthesis” and “streamlines synthetic pathways” effectively describes the same problem.
* The datasets used, particularly for reaction yield prediction, could be better described to emphasize the specific challenges associated with them. For example, the B–H and S–M datasets originate from focused combinatorial experimentation efforts (meaning high quality, low quantity, low diversity), whereas the USPTO datasets are derived from the patent literature (meaning lower quality, higher quantities, higher diversity).
Along the same lines, some statements are too simplistic in my opinion, and should be more nuanced:
* “study of interactions between molecules, particularly chemical reactions, has been overlooked”
* “The advantage of RG on LvG identification demonstrates its ability to understand the reaction mechanism.” From a chemical perspective, identifying leaving groups based on the reaction equation is a relatively simple pattern recognition task, and should not be confounded with mechanistic understanding.

**Questions For Authors:**

* Given that GNNs are usually considered rather “data-hungry”, what are the data requirements for training a predictive 2D / 3D model? Is it possible to train the model from scratch, even on the small (S–M, B–H) datasets?

**Relation To Broader Scientific Literature:**

As highlighted in the introduction, compared to molecule representations, reaction representations are somewhat underdeveloped. Therefore, the introduction of the 3D-aware model represents a relevant advancement. As discussed in the `Methods and Evaluation Criteria`, some design choices remain unclear, but the improvments in predictivity are significant and noteworthy.

**Theoretical Claims:**

not applicable

---

> ### Author Rebuttal · Authors · 2025-04-01
>
> Thank you for acknowledging that our method "**outperforms SOTA methods**" and that the "**results clearly support the claim**".
> We also appreciate your insightful suggestions for improving this paper.
> ## Ⅰ. Separate vs. Unified Node
> RG separates atoms in reactant and product into two nodes, which can **preserve the uniqueness of molecular structures** and facilitate structural modeling. CGR unifies atoms into a single node, causing their structural features to become entangled during message passing. This may hinder structural understanding.
>
> Results in **Tab 1** show that RG outperforms CGR in condition and yield prediction, which rely on structural modeling.
>
> **Tab 1: Comparison of RG (separate nodes) and CGR (unified nodes).**
> Method|UC|PC|BH|SM|Test|Gram|SubGram|UT|PT
> |-|-|-|-|-|-|-|-|-|-
> CGR|0.20|0.26|0.94|0.85|0.74|0.13|0.20|0.99|0.98
> RG|0.33|0.39|0.97|0.89|0.78|0.13|0.22|0.99|0.99
> ## Ⅱ. Non-reactant Species
> For condition prediction, we exclude non-reactant species from graph and take them as prediction target, avoiding data leakage. For yield prediction, we keep them in the graph as they have a significant impact on the yield.
> ## Ⅲ. Why not Cartesian Coords or Torsional Angles
> Our work aims to predict reaction properties (e.g. conditions), which are **invariant to the specific conformations** of molecules. RG uses bond lengths and bond angles, which vary slightly across different conformations and suit the task. Cartesian Coords and Torsion Angles vary significantly between conformations, introducing redundant information and hindering model learning.
>
> Results in **Tab 2** support this point.
>
> **Tab 2: Comparison of 3D structure representations.**
> Method|T1|T5|T15
> |-|-|-|-
> Cartesian Coords|0.312|0.463|0.511
> Torsion Angles|0.302|0.447|0.494
> Ours|0.325|0.472|0.518
>
> ## Ⅳ. Tradeoffs in 3D Structure Calculation
>
> The reason for using equilibrium structures is the same as in **Ⅲ**.
>
> ETKDG+MMFF provides controllable errors (avg 5%< with DFT) and high efficiency (20ms< for 100 atoms). DFT are unaffordable (>10min for 6 atoms and polynomial growth), as USPTO contains >680k samples and reactions with >300 atoms
>
> Results in **Tab 3** show that controllable errors (5%) in 3D coords have a minor impact (0.003).
>
> **Tab 3: Impact of 3D error on 1/8 USPTO-Condition.**
> Error %|0|5|10|20|40
> |-|-|-|-|-|-
> Accuracy|0.177|0.174|0.170|0.164|0.152
> ## Ⅴ. Challenges of Datasets
> - The key bottlenecks of **USPTO-Yield** lie in the low label quality and missing condition annotations[1]. Although large in scale, the distribution remains sparse due to the large variety of reaction types, making it hard to learn the non-smooth relation between reaction and yield.
> - **B–H/S–M** are come from HTE, which provide dense, high-quality labels but limited data(<10K samples, <50 molecules). Test sets have additives held out from train set, which places high demands on model's generalization on small data scale.
>
> More stats and discussions can be found in Appendix Sec G.
>
> ## Ⅵ. Writing, Content Arrangement, and Figures
> Your suggestions are very helpful for improving the manuscript's quality.
>
> ### Introduction Revision
>
> - **AI4Chem:**
> In the field of chemistry [2], AI enables precise spectral analysis [...] and quantum chemical simulation [...], improves inverse design of molecular structure [...] and retrosynthesis planning [...].
>
> - **GNN:**
> Among various representation methods, molecular graphs [3,4] have proven inherently advantageous for various chemical tasks [...].
> ### Figure Revision
> https://huggingface.co/reactiongraph/Revision.
> ### Section Arrangement
> Sec 3.2 focuses on exploring suitable 3D features, whereas Sec 3.3 focuses on specific task.
>
> But you made well-reasoned point that these two Secs are not logically parallel. Therefore we rearrange them as followed:
> - 3.1 The roles of Reaction and 3D Information
>     - 3.1.1 The Effect of Reaction Information
>     - 3.1.2 The Effect of 3D Information
> - 3.2 Reaction-related Tasks
>     - 3.2.1 Reaction Condition Prediction
>     - 3.2.2 Reaction Yield Prediction
>     - 3.2.3 Reaction Classification
> ## Ⅶ. Data Requirement
> Reac info is effective across different data scales. 3D info requires larger amounts of data.
>
> Models trained from scratch on B–H and S–M shows advantage of reac info (Sec 3.4), and we also conduct experiments on 1/8 of USPTO dataset (see **Tab 4**).
>
> **Tab 4: Results on 1/8 data scale.**
> Method|U-C|U-T
> |-|-|-|
> MG|0.13|0.92
> RG|0.17|0.97
>
> We use different scaffolds on Pistachio to further explore the data requirements of 3D info. **Tab 5** shows that limited data scale hinders the model from learning 3D priors. Results in [5] also leads to a similar conclusion.
>
> **Tab 5: Results of different data scale.**
> Scale|w 3D|w/o 3D
> |-|-|-
> 1|0.392|0.385
> 1/2|0.355|0.350
> 1/8|0.277|0.272
> 1/32|0.186|0.191
>
> [1] Prediction of chemical reaction yields using deep learning
>
> [2-4] correspond to the papers in your comment.
>
> [5] Uni-Mol: A Universal 3D Molecular Representation Learning Framework

---

> > ### Comment · Reviewer_vEQY · 2025-04-05
> >
> > I thank the authors for their responses to my questions and concerns.
> >
> > I would like to clarify one aspect of miscommunication between my review and the authors' rebuttal: A complete 3D molecular conformation is defined either by a full set of Cartesian coordinates or by *internal coordinates* (i.e., the complete set of bond lengths, bond angles, and dihedral angles). I agree with the authors that, in general, variability across conformers increases in the following order: bond lengths < bond angles < dihedral angles. Given this reasoning, the authors' choice to apply a "cutoff" by excluding dihedral angle information from their model appears reasonable. I recommend explicitly including this justification in the manuscript.
> >
> > Overall, I remain convinced by the paper's central idea and continue to lean towards acceptance.

---

> > > ### Author Response · Authors · 2025-04-07
> > >
> > > Dear Reviewer vEQY,
> > >
> > > Thank you for your time and effort in reviewing our work.
> > >
> > > We appreciate your thoughtful clarification regarding the use of internal coordinates. We are glad that our explanation helped clarify the rationale behind excluding dihedral angle information. The manuscript has been revised accordingly to explicitly justify this cutoff.
> > >
> > > Thank you once again for your insightful comments.
> > >
> > > Best regards.

---

### Official Review · Reviewer_S7nT · 2025-03-12

**Overall Recommendation:** 3

**Summary:**

This paper proposes a new graph representation for reaction related tasks, named Reaction Graph (RG). Compared to traditional molecule graph representation, RG introduces a new edge type, ie, reaction edge, which indicates the edges that have been changed during the reaction process. The experimental results show that the proposed RG is effective in various reaction tasks.

**Claims And Evidence:**

1. Unclear claim: "However, this method still separates reactions, which also causes loss of reaction information." How RXN Hypergraph separates reactions and what information is lost remains unclear;

2. In the experiments sections, what specific prediction model is conducted to compare MG and RG representation is ambiguous. Conducting a framework specifically designed for RG on both MG and RG representations may introduce significant bias, as the results could be influenced by the framework's unsuitability for MG representation. For a fair comparison,  the author should conduct a typical used framework on the two representations.

3. Similar to 2, how the 3D information is excluded for comparison is not stated. Will the results of w/o 3D information introduce any bias due to the change of framework?

**Essential References Not Discussed:**

All essential references are covered.

**Experimental Designs Or Analyses:**

How the ablation experiments are conducted is unclear, which may cause some biased results. Does the framework keep consistency in different settings? Will the framework introduce bias to specific settings? Please also check Claims And Evidence part.

**Methods And Evaluation Criteria:**

Yes.

**Other Comments Or Suggestions:**

*  A typo at Line 3330.
* Details about how $l_{ij}$ is calculated.
* Details about how 3D information is included.

**Other Strengths And Weaknesses:**

### Strengths:
1. The proposed reaction graph is intuitive, which can naturally benefit the chemical reaction related tasks.
2. The authors conduct a variety of experiments to show the effectiveness of the proposed framework.
3. The experimental results can demonstrate the priority of proposed framework.

### Weaknesses:
1. Some implementation details about ablations are not clear. Please check Claims And Evidence.
2. The experiments mainly focus on demonstrating the effectiveness of RG, but the modules in the framework lack discussion, eg., Attention-based aggregation, Edge-embedding, and Vertex-edge embedding.
3. The main focus of this paper is not clear. The authors proposed RG and a framework, but how they benefit each other, why they specifically designed a framework for RG, and how effective both modules are, are not clear. The authors should discuss more on each of them, but not only discuss the effectiveness of RG while employing experiments with the overall framework.
4. Some contributions lack discussion, eg, the framework design.

**Questions For Authors:**

Please see above.

**Relation To Broader Scientific Literature:**

The proposed Reaction Graph representation can benefit a broad area in chemical reaction related tasks.

**Theoretical Claims:**

No theoretical claims.

---

> ### Author Rebuttal · Authors · 2025-03-31
>
> Thank you for acknowledging that this paper proposes a "**new graph representation**" and is "**effective in various reaction tasks**."
> We also appreciate your suggestions.
>
> ## Ⅰ. Clarification on Reaction Separation and Information Loss in RXN Hypergraph
> **Seperate Reactions**: In the RXN Hypergraph, reactants and products are represented as separate graphs, rather than integrating the entire reaction into a single connected graph.
>
> **Information Loss**: This representation lacks atom mapping between reactants and products, leading to the loss of key information on bond breaking, bond formation, and atomic reorganization.
>
> **Experimental Verification**: We try to convert the standard non-connected RXN Hypergraph (w/o Reac) into a connected graph (w/ Reac). The improved performance confirms our claim (see **Tab 1**).
>
> **Tab 1: Impact of reaction connectivity.**
> | Method | U-T | P-T |
> |-|-|-|
> w/o Reac | 0.954|0.911
> w/ Reac |0.984|0.936
>
> ## Ⅱ. Results of MG and RG in Typical Framework
>
> We use the typical used RGCN implementation from DGL offical repo. As in **Tab 2**, RG outperforms MG on all the tasks.
>
> **Tab 2: Results of RG and MG on RGCN.**
> Method|1/8 U-C|BH 1-4|1/8 U-T
> |-|-|-|-|
> MG|0.13|0.75|0.92
> RG|0.17|0.78|0.97
>
> We also test MG on its own SOTA framework UniMol. As in **Tab 3**, our method shows advantage in reaction tasks.
>
> **Tab 3: Results of RG and MG on their own SOTA framework.**
> |Method|U-C|BH 1-4|U-T|
> |-|-|-|-|
> |UniMol|0.30|0.62|0.98
> |RG|0.33|0.78|0.99
>
> ## Ⅲ. Details about Ablation
> **Method Overview:** The main contribution of this work is proposing Reaction Graph (RG), a new representation for chemical reactions, characterized by two key features:
> - **Reaction info** are used to model reaction change, which is incorporated through reaction egde;
> - **3D structures** are incorporated through bond lengths and angular edges.
>
> **Ablation Settings:** All ablations are conducted on the **same framework**, with **same hyperparameter setting**. We remove reaction edges for reaction info ablation, and set all edge lengths to 0 for 3D ablation.
>
> ## Ⅳ. Design and Effect of Each Module in the Framework
>
> Our main contribution is **proposing a novel reaction graph representation**. Still, we have carefully considered the framework and conducted extensive experiments. For more details, please refer to Appendix Sec.H.7.
>
> ### 1. Edge Embedding
> We employ typical embedding layer for edge type, and RBF for edge length. RBF lifts edge lengths into high-dim vector by smooth mappings, which help in capturing variations in continuous data. It focus more on local structural pattern, which is suitable for tasks involving local continuous spatial relationship (e.g. molecular structures).
>
> As in **Tab 4**, RBF outperforms linear projection and discretization.
>
> **Tab 4: Comparison of different edge embedding methods.**
> | Method | T1 |T5|T15 |
> |-|-|-|-|
> Linear Projection | 0.317|0.468|0.516|
> Discretization | 0.310|0.457|0.505
> RBF | 0.325|0.472|0.518
>
> ### 2. Vertex-Edge Integration
>
> We map each edge type to a unique learnable message function $f_e(v,e,l_e)$, which computes messages for node feature update. Through this approach, the network can handle bond, reaction and angular edge in a targeted manner, as their semantics are different.
>
> Compared with other vertex-edge integration methods, our method achieves better accuracy (see **Tab 5**).
>
> **Tab 5: Comparison of different vertex-edge integration methods.**
> | Method | T1 |T15 |
> |-|-|-|
> PaiNN | 0.290|0.488
> DimeNet | 0.318|0.514
> Graph Transformer | 0.300|0.477|
> EGAT | 0.304|0.502
> GINE | 0.299|0.487
> Ours |0.325|0.518
>
> ### 3. Aggregation
> To capture RG's global representation, we use an LSTM with attention aggregation. It helps the network focus on the sites related to the reaction change.
> Ablation results validates the effectiveness of each component (See **Tab 6**).
>
> **Tab 6: Ablation results of aggregation module.**
> Attention|LSTM|T1|T5|T15|
> |-|-|-|-|-|
> |||0.3156|0.4642|0.5110
> |√||0.3187|0.4670|0.5136
> |√|√|0.3246|0.4715|0.5181
>
> ## Ⅴ. Calculation of Edge Length $l_{ij}$
>
> First, we input each molecule into RDKit and calculate 3D atomic coords using ETKDG + MMFF.
> The 3D coords of atoms $i$ and $j$ are $p_i$ and $p_j$, respectively.
> The length of edge $e_{ij}$ is calculted as:
> $$
> l_{ij} =
> \begin{cases}
> 0, & \text{$e_{i,j}$ is reaction edge} \\\\
> ||p_i - p_j||_2, & \\text{otherwise}
> \end{cases}
> $$
>
> ## Ⅵ. How 3D Info is Included
> 1. We calculte the edge length $l_{ij}$ (refer to Ⅴ).
> 2. We input $l_{ij}$ to RBF kernel network to get the edge length embedding $\boldsymbol{l}_{ij}$.
> 3. We concat $\boldsymbol{l}ij$ with edge type embedding $\boldsymbol{e}_{ij}$.
> 4. Edge length and type features are combined with vertex features through vertex-edge integration, thereby incorporating 3D info into RG.
>
> ## Ⅶ. Typo Correction
> The typo in line 3330 has been corrected, and the text has been re-proofread. Thank you for your reminder.

---

> > ### Comment · Reviewer_S7nT · 2025-04-03
> >
> > Thanks for the detailed reply and comprehensive experiments. Most of my concerns are addressed. I hope the authors can revise the main text accordingly. I will increase my score to weak accept.

---

> > > ### Author Response · Authors · 2025-04-03
> > >
> > > Dear Reviewer S7nT,
> > >
> > > Thank you for acknowledging the comprehensiveness of our experiments. We appreciate your feedback and will ensure that the revisions to the main text address your concerns thoroughly. We are grateful for your decision to increase your score to a weak accept.
> > >
> > > Best regards.

---

### Official Review · Reviewer_DHMj · 2025-03-15

**Overall Recommendation:** 3

**Summary:**

This paper introduces the Reaction Graph to model the chemical reaction as a graph and capture the molecular transformations during reaction. The reaction edge connects nodes representing the same atom in both reactants and products based on atomic mapping relationships.

Main Results:

1. A condition prediction model on USPTO to show the advantages of the Reaction Graph (RG), demonstrating their model could focus on different parts of molecules, especially reaction centers.

2. The proposed model shows superiority on the Leaving Group (LvG) identification, Reaction Condition Prediction, and Reaction Yield Prediction tasks.


Main algorithmic ideas:

1. The paper proposed a Reaction Graph based on atomic mapping relationships, whose edge connects nodes representing the same atom in both reactants and products.

2. Integrate 3D molecular information into reaction modeling.

**Claims And Evidence:**

1. The experimental results lack comparison with the results of newer papers,  which makes it difficult to evaluate the contribution.

[1] Teasing out missing reactions in genome-scale metabolic networks through hypergraph learning.
[2] Self-supervised contrastive molecular representation learning with a chemical synthesis knowledge graph.
[3] Bridging the gap between chemical reaction pretraining and conditional molecule generation with a unified model

**Essential References Not Discussed:**

The experimental results lack comparison with the results of newer papers,  which makes it difficult to evaluate the contribution.

[1] Teasing out missing reactions in genome-scale metabolic networks through hypergraph learning.
[2] Self-supervised contrastive molecular representation learning with a chemical synthesis knowledge graph.
[3] Bridging the gap between chemical reaction pretraining and conditional molecule generation with a unified model


A paper that considers 3D transition state structures in chemical reactions is  also worth including in the discussion:

[4] Accurate transition state generation with an object-aware equivariant elementary reaction diffusion model

**Experimental Designs Or Analyses:**

1. The method's evaluation lacks comparison with recent research, making it difficult to assess its advantages.


2. It remains unclear whether the entire reaction graph is used as input for calculating 3D coordinates.

3. While the model outperforms baseline methods significantly (0.30 vs 0.21) even without Reaction Edges and 3D structures, there is insufficient ablation analysis to explain which components contribute to this performance. More detailed ablation studies are needed.

4. The impact of using 3D information in other frameworks, such as D-MPNN on MG graphs, has not been explored. Investigating this could provide additional insights into the utility of 3D information across different model

**Methods And Evaluation Criteria:**

1. The proposed method, which models relationships between reactants and products, is pertinent to chemical reaction characterization. It achieves significant results across six datasets, including tasks like reaction center prediction. But, the evaluation lacks comparison with recent studies.

2. It mainly focuses on enhancing the learning of edge information in 3D graph representations. However, it does not deeply consider molecular equivariance, which is popular in 3D molecular learning. While this approach works, it is quite common in graph network learning. There is limited innovation in addressing more complex properties of molecular structures.

**Other Comments Or Suggestions:**

None

**Other Strengths And Weaknesses:**

None

**Questions For Authors:**

1. It mainly focuses on enhancing the learning of edge information in 3D graph representations. However, it does not deeply consider molecular equivariance, which is popular in 3D molecular learning. While this approach works, it is quite common in graph network learning. There is limited innovation in addressing more complex properties of molecular structures

2. The method's evaluation lacks comparison with recent research, making it difficult to assess its advantages.


3. It remains unclear whether the entire reaction graph is used as input for calculating 3D coordinates.

4. While the model outperforms baseline methods significantly (0.30 vs 0.21) even without Reaction Edges and 3D structures, there is insufficient ablation analysis to explain which components contribute to this performance. More detailed ablation studies are needed.

5. The impact of using 3D information in other frameworks, such as D-MPNN on MG graphs, has not been explored. Investigating this could provide additional insights into the utility of 3D information across different model.

**Relation To Broader Scientific Literature:**

1. This paper innovates by constructing reaction graphs to model chemical reactions, diverging from traditional hypergraph-based approaches validated in extensive experiments.

2. Some studies enhance chemical representations through knowledge graphs, integrating broad chemical information for improved performance, unlike the proposed reaction graph method.

3. Incorporating 3D information into the reaction graph framework is uncommon yet adds potential accuracy; however, it raises questions about the necessity and cost-effectiveness due to high generation costs.

**Theoretical Claims:**

The theoretical claims presented in the article are straightforward and do not present any issues.

---

> ### Author Rebuttal · Authors · 2025-03-31
>
> Thank you for acknowledging that our method "**shows superiority on various tasks**."
> We also appreciate your suggestions.
>
> ## Ⅰ. Comparison with Recent Studies
> The performance of our method **outperforms** recent methods ReaKE [2] and UniRXN [3] (see **Tab 1**).
>
> **Tab 1: Comparison with recent baselines. [\*] are reported in [2-3]**
> |Method|U-C|B-H|S-W|Test|U-T
> |-|-|-|-|-|-|
> |ReaKE [2]|0.23|0.89|0.76|0.76*|0.95|
> |UniRXN [3]|0.20|0.94|0.85|0.58|0.92*|
> |Ours|0.33|0.97|0.89|0.78|0.99|
>
> CHESHIRE [1] may **NOT** be directly related to our paper. It focuses on filling missing edges in a Genome-scale Metabolic Graph, while we focus on predicting chemical reaction properties.
>
> ## Ⅱ. Innovation in Using 3D
> Previous methods typically use bond lengths and bond angles.
> However, their differing physical meanings and numerical distributions may cause feature mismatches.
> To address this, we innovatively introduce angular edge, which uses edge length to implicitly represent angle, achieving a unified representation.
>
> ## Ⅲ. Why not Equivariant NNs?
> **Equivariant Networks (EqvNs)** are suitable for tasks that depend on specific conformations, such as energy prediction.
> **Invariant Networks (InvNs)** are suitable for tasks that rely solely on the type of molecule rather than specific conformations, such as predicting reaction conditions and yields.
> This paper focuses on predicting reaction properties, which are invariant to specific conformations, requiring an InvN.
>
> Results in **Tab 2** validate the advantage of InvNs in reaction modeling.
> Hence, our RG uses the invariant features.
>
> **Tab 2: Comparision of EqvN and InvN.**
> |Method|T1|T5|T15|
> |-|-|-|-|
> |EqvN|0.29|0.44|0.49|
> |InvN|0.33|0.47|0.52|
>
> ## Ⅳ. Calculation of 3D Coords
> The entire RG is **NOT** directly used as input for calculating 3D coords.
> Coords of each molecule are calculated individually.
> Specifically, we input each molecule in reaction into RDKit, and then use the ETKDG + MMFF to calculate 3D atomic coords.
> ## Ⅴ. Further Ablation Analysis
> We conduct additional ablations on model components and training strategies.
> Our backbone, equipped with **CRM-H** and trained with a **two-stage strategy**, achieves a top-1 accuracy of 0.3, surpassing the baseline’s 0.2.
> Incorporating RG further boosts the accuracy to 0.33. Details are in Appendix Sec. H.4.3.
>
> **Tab 3: Ablation results.**
> |CRM-H|2 Stage|RG|T1|T5|T10|
> |-|-|-|-|-|-|
> ||||0.25|0.32|0.39|
> |√|||0.30|0.44|0.47|
> |√|√||0.31|0.45|0.49|
> |√|√|√|0.33|0.47|0.51|
>
> ## Ⅵ. Using 3D Info in Other Frameworks
> The results in **Tab 4** show that introducing 3D info into other frameworks leads to accuracy improvements.
>
> **Tab 4: Using 3D Info on other frameworks.**
> |Method|3D|T1|T5|T15|
> |-|-|-|-|-|
> |D-MPNN| |0.1977 |  0.3341 |  0.3924 |
> ||√|0.2030|0.3410|0.3971|
> RHG||0.2127|0.3447|0.3927|
> ||√|0.2149|0.3464|0.3949|
>
> ## Ⅶ. The Necessity of 3D Info
> Several methods have explored incorporating 3D info into molecular modeling.
> Uni-Mol [5] uses large-scale 3D positions, GraphMVP [6] uses bond lengths, and GEM [7] uses bond angles.
> Inspired by these works, we explore 3D in reaction modeling, which improves accuracy (see **Tab 5**).
>
> Although generating 3D conformations is currently costly, many studies [8–9] are actively addressing this issue.
> We are optimistic about the potential of incorporating 3D into the RG framework.
>
> **Table 5: The impact of 3D info.**
> |3D|T1|T3|T5|T10|T15
> |-|-|-|-|-|-|
> ||0.313| 0.425| 0.461| 0.496| 0.509 |
> |√|0.325 |0.434 |0.472 | 0.506 | 0.518 |
> ## Ⅷ. Discussion of Transition State (TS) Structures
>
> 1. TS methods are inspiring to us. However, as discussed in Ⅲ, TS structures are related to specific 3D conformations, whereas reaction properties like conditions are independent of specific conformations. Therefore, their focus in 3D modeling differs.
> 2. We explore TS generation backbones for predicting reaction properties.
> Our RG achieves better performance (see **Tab 6**).
> 3. Methods for calculating TS, such as DFT, are very time-consuming (~10min for 10 atoms[10] and polynomial growth) and unaffordable in dataset like USPTO, which includes 680k reactions with up to 347 atoms. Hence, we didn't use TS structures as inputs for the network.
>
>     **Tab 6: Comparison with TS generation backbones.**
>     |Method|T1|T5|T15|
>     |-|-|-|-|
>     |OARD [4]|0.13|0.20|0.31|
>     |EquiReact [11]|0.12|0.24|0.30|
>     |Ours|0.18|0.31|0.37|
>
> [1]-[4] correspond to those in the comments.
>
> [5] Uni-Mol: A Universal 3D Molecular Representation Learning Framework
>
> [6] Pre-training Molecular Graph Representation with 3D Geometry
>
> [7] Geometry-enhanced molecular representation learning for property prediction
>
> [8] Predicting molecular conformation via dynamic graph score matching
>
> [9] Torsional diffusion for molecular conformer generation
>
> [10] Fast and Automatic Estimation of Transition State Structures Using Tight Binding Quantum Chemical Calculations
>
> [11] 3DReact: Geometric deep learning for chemical reactions

---

> > ### Comment · Reviewer_DHMj · 2025-04-05
> >
> > Thanks for your detailed response and thorough experiments. In light of these results, I've updated my score to 3.
> >
> > I believe incorporating some of the above comments into the revised manuscript will also improve the paper, and hope you do so.

---

> > > ### Author Response · Authors · 2025-04-07
> > >
> > > Dear Reviewer DHMj,
> > >
> > > Thank you very much for your time and effort in reviewing our submission.
> > > The suggested revisions have been incorporated into the main text to further improve the quality of the paper.
> > > We sincerely appreciate your feedback and your decision to raise the score.
> > >
> > > Best regards.

---

### Decision · Program_Chairs · 2025-05-01

**Decision:**

Accept (poster)

**Comment:**

This research introduces the Reaction Graph (RG), a novel graph-based representation designed to model chemical reactions more effectively. Also 3d information is represented as well under this framework. Experiments have been done on a set of benchmarks.

I appreciate the authors’ effort in addressing reviewers concerns especially regarding several ablation studies, which makes the paper more comprehensive. I hope the authors to take these feedback and additional experiments into the revision of the paper, to reflect the improvements done on the quality of this manuscript.